# Piecewise-Stationary Dueling Bandits

**Patrick Kolpaczki**                                                    *patrick.kolpaczki@upb.de*
*Department of Computer Science*
*Paderborn University*

**Eyke Hüllermeier**                                                           *eyke@lmu.de*
*Institute of Informatics, LMU Munich*
*Munich Center for Machine Learning*

**Viktor Bengs**                                                        *viktor.bengs@lmu.de*
*Institute of Informatics, LMU Munich*
*Munich Center for Machine Learning*
*CIFAR Fellow*

**Reviewed on OpenReview:** *https://openreview.net/forum?id=WhEHEDP7ZG*

## Abstract

We study the piecewise-stationary dueling bandits problem with $K$ arms, where the time horizon $T$ consists of $M$ stationary segments, each of which is associated with its own preference matrix. The learner repeatedly selects a pair of arms and observes a binary preference between them as feedback. To minimize the accumulated regret, the learner needs to pick the Condorcet winner of each stationary segment as often as possible, despite preference matrices and segment lengths being unknown. We propose the *Beat the Winner Reset* algorithm and prove a bound on its expected binary weak regret in the stationary case, which tightens the bound of current state-of-art algorithms. We also show a regret bound for the non-stationary case, without requiring knowledge of $M$ or $T$. We further propose and analyze two meta-algorithms, *DETECT* for weak regret and *Monitored Dueling Bandits* for strong regret, both based on a detection-window approach that can incorporate any dueling bandit algorithm as a black-box algorithm. Finally, we prove a worst-case lower bound for expected weak regret in the non-stationary case.

## 1 Introduction

The stochastic *multi-armed armed bandit* (MAB) problem (Thompson, 1933; Robbins, 1952) is an online learning framework in which an agent (learner) chooses repeatedly from a set of $K$ options — called *arms* — in the course of a sequential decision process with (discrete) *time horizon $T$*. Each arm $a_i$ is associated with an unknown *reward distribution* having finite mean and being stationary over the learning process. Choosing arm $a_i$ results in a random *reward* sampled from that distribution. The learner's goal is to choose the *optimal arm*, i.e., the one having highest mean reward, as often as possible, because a suboptimal arm with lower (expected) reward implies a positive *regret* (reward difference). Due to the absence of knowledge about the reward distributions, this task of *cumulative regret minimization* comes with the challenge of tackling the *exploration-exploitation dilemma*: As the learner requires reasonably certain estimates of the arms' mean rewards, it must *explore* by choosing each arm sufficiently often. Otherwise, because rewards are random, it may believe that a suboptimal arm is optimal. On the other side, too much exploration is not good either, because exploration comes at the cost of *exploitation* (choosing the presumably optimal arm), thereby increasing cumulative regret.

In many practical applications, the assumption of stationary reward distributions is likely to be violated. In light of this, the *non-stationary MAB problem* has received increasing interest in the recent past (Hartland

et al., 2006; Liu et al., 2018; Auer et al., 2019). Here, two main types of non-stationarity are distinguished: conceptual *drift*, where the reward distributions change gradually over time, and conceptual *shift*, where the reward distributions change abruptly at a certain point in time, called *changepoint*. Consider music recommendation as an example: It is known that user preferences towards a certain genre can change depending on the time of the year (Pettijohn et al., 2010); while this is an example of drift, a shift might be caused by switching between users sharing a single account. Non-stationarity adds another dimension to the exploration-exploitation dilemma: not only must the learner balance exploration and exploitation to maximize the number of times the optimal arm is chosen, but additionally ensure that previous observations are still valid by conducting ancillary exploration to detect changes in the reward distributions. Algorithms tackling the problem can be divided into *passively adaptive* ones, which devalue older observations and base their strategies on observations made in more recent time steps, e.g. D-UCB (Garivier & Moulines, 2011) and SW-UCB (Garivier & Moulines, 2011), and *actively adaptive* ones trying to detect changepoints through suitable detection mechanisms (and then discarding observations prior to suspected changepoints), e.g. CUSUM-UCB (Liu et al., 2018), PHT-UCB (Liu et al., 2018), and M-UCB (Cao et al., 2019).

Another assumption of the MAB setting that is often difficult to meet in practice is the provision of feedback in the form of precise numerical rewards. In many cases, the learner is only able to observe feedback of a weaker kind, for example qualitative preferences over pairs of arms. This is especially true when the feedback is provided by a human or implicitly derived from human behavior, such as in clinical treatments (Sui & Burdick, 2014), information retrieval (Zoghi et al., 2016), or recommender systems (Hofmann et al., 2013). For example, if a playlist is recommended to a user, one may conjecture that those songs listened to are preferred to those not listened to. In light of this, a variant called *dueling bandits* (Yue & Joachims, 2009) has been proposed, in which the learner chooses a pair of arms in each time step, whereupon feedback in the form of a noisy *preference* is observed. This feedback is governed by an unknown *preference matrix* which determines for each pair of arms $(a_i, a_j)$ the probability that $a_i$ wins against $a_j$, or in other words, is preferred over $a_j$. In order to define the notions of best arm and regret, a common assumption is the existence of a *Condorcet winner* (CW), i.e., an arm that is preferred over all other arms (in the sense of winning with probability $> 1/2$). The *average* or *strong regret* of a chosen pair is then defined as the average of the chosen arms' calibrated probabilities, i.e., the magnitude of the probability that the CW is preferred over an arm. Obviously, this regret is only zero for the pair containing the best arm twice (full commitment). In contrast, the *weak regret* (Yue et al., 2012) is the minimum of the chosen arms' calibrated probabilities, turning zero as soon as any of them is the CW. This is motivated by scenarios where the worse option does not have an impact on the pair's quality. In this case, a learner can conduct exploration and exploitation simultaneously by playing a reference arm — the presumably best one — together with another "exploration arm".

Although the non-stationary variant of the MAB problem has been studied quite intensely in the recent past, the research on non-stationary dueling bandits is still in its infancy. This is somewhat surprising since changes of preferences are not uncommon in applications of dueling bandits.

**Contribution.** In this paper, we contribute to the theoretical understanding of non-stationary dueling bandits, in which preferences change $M - 1$ times at unknown time points. We give a formal problem statement for the *piecewise-stationary dueling bandits problem* in Section 2, with emphasis on stationary segments being separated by changepoints and the Condorcet winner as the best arm. As a first algorithm to tackle the problem, we propose *Beat the Winner Reset* (BtWR) for weak regret and show expected regret bounds of $\mathcal{O}(\frac{K \log K}{\Delta^2})$ in the stationary setting and $\mathcal{O}(\frac{KM}{\Delta^2} \log(K + T))$ in the non-stationary setting (Section 3) under the assumption that the suboptimality gap $\Delta$ is known. BtWR enjoys a tighter regret bound than other state-of-the-art algorithms for the stationary case, and does not need to know the time horizon $T$ or the number of segments $M$ for the non-stationary case. Next, we propose the *Monitored Dueling Bandits* (MDB) meta-algorithm for strong regret based on a detection-window approach, which is parameterized with a black-box dueling bandits algorithm for the stationary setting (Section 4). We bound its expected regret by $\mathcal{O}\left(K\sqrt{MT \log \frac{T}{MK}}\right)$ plus the sum of regret that the black-box algorithm would have incurred when being run on its own for each (stationary) segment. In Section 5, we additionally present with *DETECT* an adaptation of MDB towards weak regret and prove its expected regret to be in $\mathcal{O}(KM \log T)$ plus the sum of regret that the black-box algorithm would have incurred when being run on its own for each

segment. Our regret bounds and the quantities that are required to be known for each bound are summarized in Table 1. Further, we prove a worst-case lower bound of $\Omega(\sqrt{KMT})$ for the expected weak regret of any algorithm for the problem (Section 6). In a comparison with state-of-the-art methods, we provide empirical evidence for our algorithm's superiority.

Table 1: Overview of theoretical regret bounds derived for our proposed algorithms. The second column denotes whether we consider the stationary or the non-stationary case. The last four columns indicate whether the quantity needs to be known for parametrization.

| Algorithm | Stationary | Regret type | Regret bound | $T$ | $M$ | $\Delta$ | $\delta$ / $\delta_*$ |
|---|---|---|---|---|---|---|---|
| **BtWR** | ✓ | bin. weak | $\mathcal{O}\left(\frac{K \log K}{\Delta^2}\right)$ | ✗ | - | ✓ | - |
| **BtWR** | ✗ | bin. weak | $\mathcal{O}\left(\frac{KM \log(K+T)}{\Delta^2}\right)$ | ✗ | ✗ | ✓ | ✗ |
| **MDB** | ✗ | bin. strong | $\mathcal{O}\left(\frac{K\sqrt{MT \log(\delta T/KM)}}{\delta}\right) + R_M^{\bar{S}}(Alg)$ | ✓ | ✓ | ✗[1] | ✓ |
| **DETECT**[2] | ✗ | bin. weak | $\mathcal{O}\left(\frac{MK \log T}{\delta_*^2} + M \log T\right) + R_M^{\bar{W}}(Alg)$ | ✓ | ✗ | ✗[1] | ✓ |

[1] The algorithm itself does not require $\Delta$, but the underlying black-box dueling bandits algorithm might do.
[2] With BtW as the underlying black-box dueling bandits algorithm.

**Related Work.** There is a wealth of work on the non-stationary bandit problem with numerical rewards (Hartland et al., 2006; Kocsis & Szepesvári, 2006; Garivier & Moulines, 2011; Liu et al., 2018; Auer et al., 2019). See Lu et al. (2021) for a good overview of this branch of literature. From a methodological point of view, the work by Cao et al. (2019) is closest to our approaches MDB and DETECT. For the dueling bandits problem (Yue & Joachims, 2009), a variety of algorithms for strong regret based on the Condorcet winner has been proposed: Beat the Mean (Yue & Joachims, 2011), Interleaved Filter (Yue et al., 2012), SAVAGE (Urvoy et al., 2013), RUCB (Zoghi et al., 2014a), RCS (Zoghi et al., 2014b), RMED (Komiyama et al., 2015), MergeRUCB (Zoghi et al., 2015), MergeDTS (Li et al., 2020). Although weak regret has been proposed by Yue et al. (2012), Winner Stays (Chen & Frazier, 2017) and Beat the Winner (Peköz et al., 2020) are the only algorithms specifically tailored to this regret that we are aware of. Remarkably, their expected regret bounds are constant w.r.t. $T$. Bengs et al. (2021) give a survey of dueling bandits.

Non-stationary dueling bandits have been considered by Saha & Gupta (2022); Buening & Saha (2023); Suk & Agarwal (2023). All of them consider dynamic strong regret, which is a stricter notion of regret as the learner's selection are compared against a dynamic benchmark arm every time step. Saha & Gupta (2022) propose with Dex3.S an algorithm inspired by EXP3 (Auer et al., 2002b) from the adversarial setting that achieves a high probability strong regret bound of $O(\sqrt{KMT} \log {}^{KT}/_\epsilon)$ with at least $1 - 2\epsilon$ probability if the number of stationary segments M is known. Buening & Saha (2023) rely on this previous work in order to propose with ANACONDA an algorithm that adapts to the environment without requiring the number of segments. It achieves an expected strong regret of $\tilde{O}(K\sqrt{MT})$. Suk & Agarwal (2023) also consider strong regret by counting Condorcet winner changes and achieve with METASWIFT $\tilde{O}(\sqrt{KMT})$ strong regret but under the relatively strong assumption of stochastic transivity and the stochastic triangle inequality.

Our main focus is on static weak regret, where the benchmark arm may only change on certain time segments, and the learner does not suffer regret as long as the segment-wise benchmark arm occurs in the duel. We ask ourselves the question to what extent the regret bounds change for weak regret if non-stationarity is now present. In particular, what the dependence on the time horizon $T$ will be. In contrast to strong regret, the notion of weak regret in dynamic environments is so far left unexplored.

## 2 Problem Statement

The piecewise-stationary dueling bandits problem involves a finite set of $K$ arms $\mathcal{A} = \{a_1, \ldots, a_K\}$, a time horizon $T \in \mathbb{N}$, and $M - 1$ changepoints $\nu_1, \ldots, \nu_{M-1} \in \mathbb{N}$ unknown to the learner with $1 < \nu_1 < \ldots < \nu_{M-1} \leq T$ dividing the entire learning time into $M$ stationary segments. We additionally define dummy changepoints $\nu_0 = 1$ and $\nu_M = T + 1$, allowing us to express the $m$-th stationary segment as the set $S_m := \{\nu_{m-1}, \ldots, \nu_m - 1\}$ spanning between the $(m-1)$-th and the $m$-th changepoint exclusively. For each stationary segment $S_m$ the environment is characterized by a preference matrix $P^{(m)} \in [0,1]^{K \times K}$ unknown to the learner, with each entry $P_{i,j}^{(m)}$ denoting the probability that the arm $a_i$ wins against $a_j$ in a duel. From here on we write $p_{i,j}^{(m)} := P_{i,j}^{(m)}$. To be well-defined, we assume that $p_{i,j}^{(m)} + p_{j,i}^{(m)} = 1$ for all $a_i, a_j$, i.e., the probability of a "draw" is zero. For each segment $S_m$, we assume the existence of the Condorcet winner $a_{m^*}$ that beats all other arms with probability greater than half, i.e., $p_{m^*,i}^{(m)} > \frac{1}{2}$ for all $a_i \in \mathcal{A} \setminus \{a_{m^*}\}$, and refer to it as the optimal arm of the $m$-th stationary segment.

To provide expressive notation for our analyses, we define

- the *change* of a pair $(a_i, a_j)$ at the $m$-th changepoint $\nu_m$ as $\delta_{i,j}^{(m)} := p_{i,j}^{(m+1)} - p_{i,j}^{(m)}$,
- the $m$-th *segmental change* $\delta^{(m)} := \max_{i,j} |\delta_{i,j}^{(m)}|$,
- the *minimal segmental change* $\delta := \min_m \delta^{(m)}$,
- the $m$-th *Condorcet Winner segmental change* $\delta_*^{(m)} := \max_j |\delta_{m^*,j}^{(m)}|$,
- and the *minimal Condorcet Winner segmental change* $\delta_* := \min_j \delta_*^{(m)}$.

Furthermore, we define

- calibrated preference probabilities $\Delta_{i,j}^{(m)} := p_{i,j}^{(m)} - \frac{1}{2}$,
- suboptimality gaps $\Delta_i^{(m)} := \Delta_{m^*,i}^{(m)}$ for each segment $S_m$,
- and the minimal suboptimality gap $\Delta := \min\limits_{1 \leq m \leq M, i \neq m^*} \Delta_i^{(m)}$ over all segments.

In each time step $t \in S_m$, the learner plays a pair of arms $(a_{I_t}, a_{J_t})$ and thereupon obverses a binary random variable $X_{I_t, J_t}^{(t)} \sim \text{Ber}\left(p_{I_t, J_t}^{(m)}\right)$. All $X_{i,j}^{(t)}$ are mutually independent. We distinguish between two types of instantaneous regret that the learner suffers in each time $t \in S_m$:

- Strong regret given by $r_t^{\text{S}} := \frac{1}{2}\left(\Delta_{I_t}^{(m)} + \Delta_{J_t}^{(m)}\right)$,
- Weak regret given by $r_t^{\text{W}} := \min\left\{\Delta_{I_t}^{(m)}, \Delta_{J_t}^{(m)}\right\}$.

Their respective binary versions are defined by $r_t^{\bar{\text{S}}} := \lceil r_t^{\text{S}} \rceil$ and $r_t^{\bar{\text{W}}} := \lceil r_t^{\text{W}} \rceil$. It is essential to involve the current preference matrix $P^{(m)}$ into the regret calculation since the regret of the same pair is free to change between segments. The cumulative regret $R^v(T)$ w.r.t. the considered version of instantaneous regret $r_t^v$ that the learner aims to minimize is given by

$$R^v(T) := \sum_{m=1}^{M} \sum_{t \in S_m} r_t^v.$$

Although we give upper bounds on expected cumulative binary strong and weak regret, they are also valid for the non-binary versions, because $r^{\text{S}} \leq r^{\bar{\text{S}}}$ and $r^{\text{W}} \leq r^{\bar{\text{W}}}$, and vice versa for lower bounds. A list of symbols used frequently throughout the paper is given by Table 2. For the remainder of the paper, we use the terms *piecewise-stationary* and *non-stationary* interchangeably.

## 3 Beat the Winner Reset

The first piecewise-stationary dueling bandits algorithm that we propose is the *Beat the Winner Reset* algorithm (BtWR) (see Algorithm 1). It is a variant of the Beat the Winner (BtW) algorithm (Peköz et al., 2020) and to be applied for weak regret. It proceeds in explicit rounds playing the same pair $(a_I, a_J)$ until a certain termination condition is fulfilled. BtWR starts by drawing an incumbent arm $a_I$ uniformly at random, puts the other $K-1$ arms in random order into a FIFO queue $Q$, and initializes the round counter $c$ to be 1. It proceeds by iterating through rounds until the time horizon is reached. In each round, the challenger arm $a_J$ is dequeued from $Q$ and the variables $w_I$ and $w_J$ counting wins of the incumbent and challenger, respectively, are initialized. The pair $(a_I, a_J)$ is played until one of them has reached $\ell_c$ many wins, called the *winner of that round*. The winner of a round becomes the incumbent $a_I$ for the next round and the loser is enqueued in $Q$, being played again $K-1$ rounds later. At last, $c$ (counting the number of successive rounds with the same incumbent $a_I$) is incremented by one if the incumbent $a_I$ has won the round. Otherwise, if the challenger $a_J$ has won, it is reset to 1. An illustration of BtWR is given in Fig. 1.

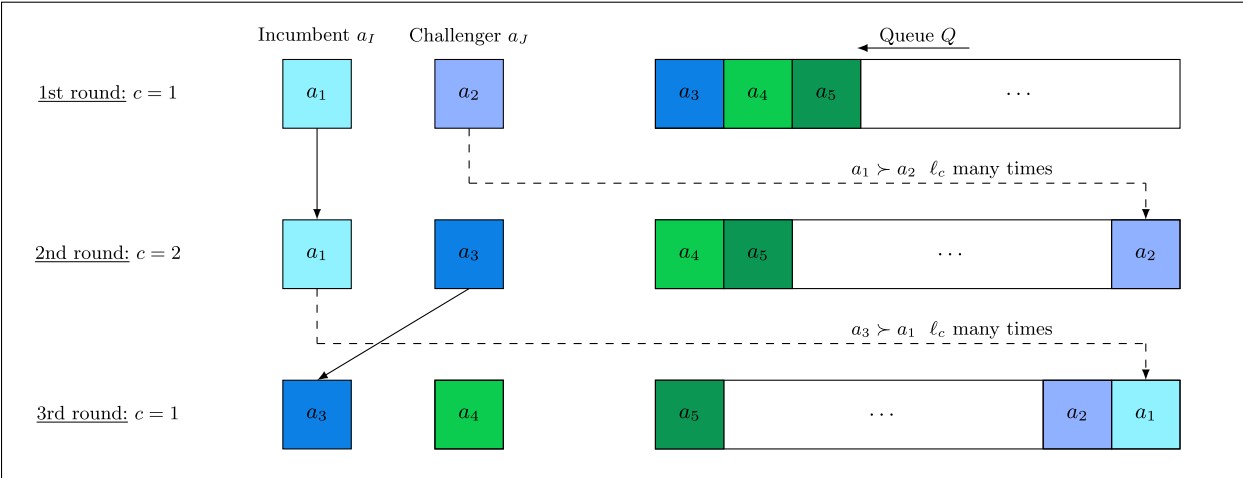

Figure 1: Scheme of BtWR: $a_1$ is chosen as the first incumbent and $a_2$ as the challenger. After $a_1$ has won $\ell_1$ many duels against $a_2$ in the first round, $a_2$ is put back to the end of the queue and $a_3$ becomes the new challenger for the second round. The pair $(a_1, a_3)$ is then played until one arm wins $\ell_2$ many times. Since the challenger arm $a_3$ has won the second round, it becomes the new incumbent and the counter $c$ is set back to 1 for the third round.

BtWR differs to its predecessor BtW in the way it updates the round length. More specifically, BtW increments it by one in each round, independent of whether the current incumbent has lost the round. Setting it back to one, allows BtWR to react to changepoints by resetting its internal state back to the time of initialization (aside from the order of arms in $Q$), whenever the new optimal arm has won a round for the first time against the current incumbent. We show in the upcoming analyses how to set $\ell_c$ in order to bound BtWR's cumulative expected weak regret.

**BtWR Stationary Regret Analysis.** In the following we sketch BtWR's regret analysis for the stationary case, i.e., $M = 1$ (all proofs are given in Appendix B). We set

$$\ell_c = \left\lceil \frac{1}{4\Delta^2} \log \frac{c(c+1)e}{\lambda} - \frac{1}{2} \right\rceil \tag{1}$$

for some $\lambda \in (0, 1)$ that we specify later. Note that this requires $\Delta$ to be known. We denote by $\tau_n$ the first time step of the algorithm's $n$-th round and define it as $\tau_1 := 1$ and $\tau_n := \inf\{t > \tau_{n-1} \mid J_{t-1} \neq J_t\}$ for $n \geq 2$. Further, let $c_n$ be the value of $c$ during the $n$-th round. First, we bound the probability of the optimal arm (w.l.o.g. $a_1$) to lose one of the remaining rounds with probability $\lambda$ as soon as it becomes the incumbent.

---

**Algorithm 1** Beat the Winner Reset (BtWR)

---

**Require:** $K \in \mathbb{N}$
 1: Draw $I$ uniformly at random from $\{1, \ldots, K\}$
 2: $Q \Leftarrow$ queue of all arms from $\mathcal{A} \setminus \{a_I\}$ in random order
 3: $c \Leftarrow 1$
 4: **while** time steps left **do**
 5:      $a_J \Leftarrow$ top arm dequeued from $Q$
 6:      $w_I, w_J \Leftarrow 0$
 7:      **while** $w_I < \ell_c$ and $w_J < \ell_c$ **do**
 8:          Play $(a_I, a_J)$ and observe $X_{I,J}^{(t)}$
 9:          $w_I \Leftarrow w_I + \mathbb{I}\{X_{I,J}^{(t)} = 1\}$
10:          $w_J \Leftarrow w_J + \mathbb{I}\{X_{I,J}^{(t)} = 0\}$
11:      **end while**
12:      **if** $w_I = \ell_c$ **then**
13:          Enqueue $a_J$ in $Q$
14:          $c \Leftarrow c + 1$
15:      **else**
16:          Enqueue $a_I$ in $Q$
17:          $I \Leftarrow J$
18:          $c \Leftarrow 1$
19:      **end if**
20: **end while**

---

**Lemma 1.** *Given that $a_1$ is the incumbent of the n-th round, the probability of $a_1$ losing at least one of the remaining rounds is at most $\lambda$.*

Next, we give a bound on the expected number of time steps it takes for $a_1$ to become the incumbent given that it finds itself in some round $\bar{n}$ not as the current incumbent.

**Lemma 2.** *Fix an arbitrary round $\bar{n}$ with $I_{\tau_{\bar{n}}} \neq 1$. Let $n_* = \inf\{n > \bar{n} \mid I_{\tau_n} = 1\}$ be the first round after $\bar{n}$ in which $a_1$ is the incumbent. The expected number of time steps needed for $a_1$ to become the incumbent is bounded by*

$$\mathbb{E}[\tau_{n_*} - \tau_{\bar{n}}] \leq \frac{6K}{\Delta^2} \log \sqrt{\frac{e}{\lambda}} (K + c_{\bar{n}} - 1).$$

Lemma 1 allows us to bound the expected number of times that $a_1$ loses a round as the current incumbent, growing with increasing $\lambda$. On the other hand, Lemma 2 implies a bound on the expected regret caused each time $a_1$ loses a round as the current incumbent. A small $\lambda$ implies longer round lengths $\ell_c$ and hence BtWR incurs more regret during rounds in which $a_1$ is placed in the queue. We set $\lambda = 1/e$ to strike a balance and derive the following result.

**Theorem 1.** *For $\lambda = 1/e$, the expected cumulative binary weak regret of BtWR in the stationary setting is bounded by*

$$\mathbb{E}\left[R^{\bar{W}}(T)\right] \leq \frac{20K \log K}{\Delta^2}.$$

The resulting bound of $\mathcal{O}\left(K \log K / \Delta^2\right)$ improves on the bound of $\mathcal{O}\left(K^2 / \tilde{\Delta}^3\right)$ for the Winner Stays algorithm (Chen & Frazier, 2017), where $\tilde{\Delta} := \min_{i \neq j} \Delta_{i,j}^{(1)}$, as well as the $\mathcal{O}\left(\exp(-\Delta^2)K / (1 - \exp(-\Delta^2))^2\right)$ bound for BtW. Although $\Delta$ is unknown in our setting, we assume its knowledge to prove theoretical results. This is commonly made also by others (see e.g. Theorem 3 for the $\epsilon$-Greedy algorithm in Auer et al. (2002a)). For completeness, we like to mention that WS and BtW do not require $\Delta$ to be known, though their dependencies are worse.

**BtWR Non-Stationary Regret Analysis.** For the non-stationary case let $a_{m^*}$ denote the optimal arm of the $m$-th segment and the round length $\ell_c$ be as in equation 1. For the proofs of the following theoretical results see Appendix C. Note that $\Delta$ now takes the suboptimality gaps of all segments into account and is thus potentially smaller than in the stationary case. We can show the following non-stationary counterpart of Lemma 1 for any segment.

**Lemma 3.** *Consider an arbitrary segment $S_m$. Given that $a_{m^*}$ is the incumbent of the $n$-th round, the probability of $a_{m^*}$ losing at least one of the remaining rounds is at most $\lambda$.*

Using Lemma 3 we can adapt Theorem 1 for the case where BtWR enters a new segment with $c$ being some number $\tilde{c} \geq 1$, providing us with a bound on the incurred expected regret in that particular segment which can be viewed as an instance of the stationary setting. Again, we set $\lambda = 1/e$.

**Lemma 4.** *For $\lambda = 1/e$ the expected cumulative binary weak regret of BtWR starting with $c_1 = \tilde{c}$ in the stationary setting, i.e., $M = 1$, is bounded by*

$$\mathbb{E}\left[R^{\bar{W}}(T)\right] \leq \frac{20K}{\Delta^2} \log(K + \tilde{c} - 1).$$

Partitioning the regret over the whole time horizon into regret incurred for each segment allows us to apply Lemma 4 per segment. Together with the fact that $c \leq T$ holds, we derive the following main result.

**Theorem 2.** *For $\lambda = 1/e$ the expected cumulative binary weak regret of BtWR in the non-stationary setting is bounded by*

$$\mathbb{E}\left[R^{\bar{W}}(T)\right] \leq \frac{22KM}{\Delta^2} \log(K + T).$$

As in the stationary setting, BtWR does not require the time horizon $T$. Additionally, the regret bounds hold with BtWR being oblivious about the number of stationary segments $M$. It is worth mentioning that, unlike the stationary case, the regret bound for the non-stationary case depends now on the time horizon $T$. This is attributable to the regret caused by the delay to have the current segment's optimal arm as the incumbent.

## 4  Monitored Dueling Bandits

Next, we present *Monitored Dueling Bandits* (MDB) (see Algorithm 2), our first algorithm using a detection-window approach to tackle dueling bandits with strong regret. Its core idea is to use the same changepoint detection mechanism as MUCB (Cao et al., 2019) by scanning for changepoints within a window of fixed length $w$ for each pair of distinct arms, thus using $K(K-1)/2$ windows. Each detection window for a distinct pair, say $(a_i, a_j)$, monitors the absolute difference of the absolute win frequencies of $a_i$ over $a_j$ of the "first" and the "second" half of the window. Once this absolute difference exceeds some threshold $b$, the changepoint detection is triggered leading to a reinitialization of MDB. In contrast to MUCB, MDB does not incorporate a specific dueling bandits algorithm with the task to minimize regret in stationary segments. Instead, it requires a black-box algorithm *Alg* for the (stationary) dueling bandits problem as an input and periodically alternates between choosing pairs selected by *Alg* and exploring the preference matrix in a round-robin manner in order to fill all detection windows with the dueling outcomes. The exploration rate by which the preference matrix is palpated for changepoints is given by a parameter $\gamma$. An illustration is shown in Fig. 2.

To be more specific, MDB requires for its parameterization the time horizon $T$, an even window length $w$, a threshold $b > 0$, an exploration rate $\gamma \in (0, 1]$, and a dueling bandits algorithm *Alg*. In particular, we assume that MDB can interact with the black-box dueling algorithm *Alg* by (i) running it for one time step leading to a concrete choice of a pair of arms and (ii) forwarding the corresponding dueling outcome for that pair to *Alg* leading to an update of its internal state. This interaction is represented by the `RunAndFeed` procedure in Algorithm 2. Moreover, we assume that MDB can reset *Alg* to its initial state. MDB starts by setting $\tau$ to zero, which represents the last time step in which a changepoint was detected, and initializes $n_{i,j}$ for each distinct pair $(a_i, a_j)$, counting the number of times the pair has been played since the last detected changepoint. An arbitrary ordering $\pi$ of all distinct pairs $(a_i, a_j)$ is fixed, starting to count at the index zero. At the beginning of each time step, MDB uses the value of $r$ to decide whether to choose a pair of arms

selected by *Alg* or to evenly explore the next pair in $\pi$. Its value is guaranteed to represent the index of a pair in $\pi$ in a proportion of $\gamma$ of all time steps, thus conducting exploration at the desired rate. In case *Alg* is run, MDB forwards the dueling outcome to it and continues with the next time step without inserting the observation into the corresponding detection window. Otherwise, if MDB performs a detection step, $n_{I_t, J_t}$ is incremented and the observed outcome $X_{I_t, J_t}^{(t)}$ is inserted into the corresponding detection window by means of $X_{I_t, J_t, n_{I_t, J_t}}$, denoting the $n_{I_t, J_t}$-th dueling outcome between $a_{I_t}$ and $a_{J_t}$ after $\tau$. The changepoint detection is triggered, if $(a_{I_t}, a_{J_t})$ has been chosen at least $w$ times after $\tau$ (i.e., its detection window is completely filled) and the window's monitored absolute difference of absolute win frequencies exceeds the threshold $b$. This induces a reset of MDB by updating $\tau$ to the current time step, setting all $n_{i,j}$ to zero, and resetting *Alg*.

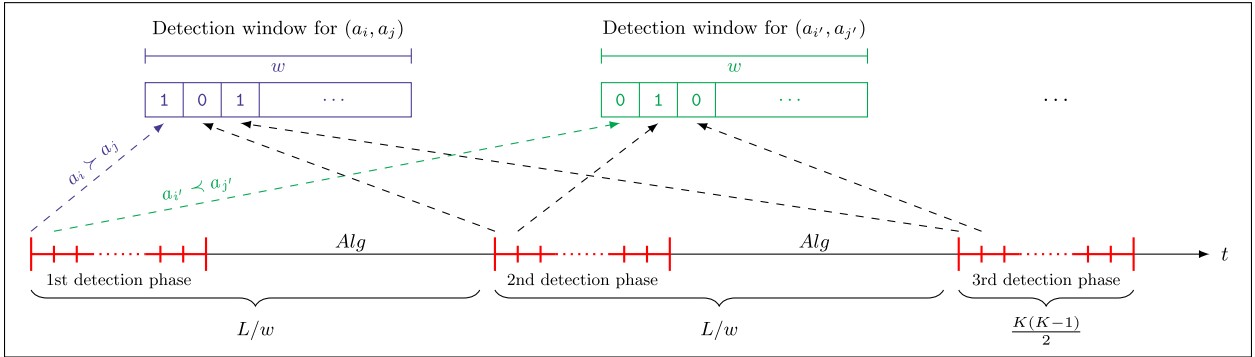

Figure 2: Scheme of MDB's detection steps (red) grouped in phases: the pairs $(a_i, a_j)$ and $(a_{i'}, a_{j'})$ are the first two of $K(K-1)/2$ many pairs in the ordering $\pi$ and thus played first in the detection phases. The time steps between the phases are used to play pairs selected by black-box dueling bandits algorithm *Alg*. All windows are filled after $w$ many phases.

**Non-Stationary Regret Analysis.** Due to the similarity of MDB in its design to MUCB, our analysis for MDB is inspired by Cao et al. (2019). Nevertheless, we adapt the analysis to the dueling bandits setting, simplify parts, merge out soft spots in their proofs and exploit room for improvement (see Appendix D for the details). A key quantity is the number of time steps $L$ needed to fill all windows completely, i.e., each distinct pair of arms $(a_i, a_j)$ has been chosen at least $w$ times. Hence, we define

$$L := w \cdot \left\lfloor \frac{K(K-1)}{2\gamma} \right\rfloor .$$

We denote by $\tilde{R}^{\bar{S}}(t_1, t_2)$ the cumulative binary strong regret that *Alg* would have incurred if it was run on its own from $t_1$ to $t_2$. Next, we impose the following assumptions for technical reasons similar to MUCB:

**Assumption 1:** $|S_m| \geq 2L$ for all $m \in \{1, \ldots, M\}$

**Assumption 2:** $\delta \geq \frac{2b}{w} + c$ for some $c > 0$

**Assumption 3:** $\gamma \in \left[ \frac{K(K-1)}{2T}, \frac{K-1}{2} \right]$

Assumption (1) is similar to (1) in Section 3.2.1 for MUCB (Cao et al., 2019) and requires a minimal length for all stationary segments depending on $L$. Thus, it can be viewed as an implicit upper bound on $w$ and lower bound for $\gamma$. It is essential and guarantees MDB enough time steps to fill all windows before the next changepoint, even after a delay of $L/2$. Assumption (2) is required to guarantee short delay with certain probability and the variable $c$ will play a role in optimizing the parameters. The difficulty of detecting a changepoint with the detection-window approach increases with shrinking $\delta$. The bound is necessary to allow

for a formal analysis, and is also to be found in Cao et al. (2019). Assumption (3) restricts the exploration rate $\gamma$ is solely of technical nature to derive a simpler bound in Lemma 5. It is not necessary from a conceptual standpoint. Our derived choice for $\gamma$ in Corollary 1 lies far away from the assumed bounds.

---

**Algorithm 2** Monitored Dueling Bandits (MDB)

---

**Require:** $K$, $T$, even $w \in \mathbb{N}$, $b > 0$, $\gamma \in (0, 1]$
 1: $\tau \Leftarrow 0$
 2: $n_{i,j} \Leftarrow 0 \ \forall a_i, a_j \in \mathcal{A}$ with $i < j$
 3: Fix any ordering $\pi$ of all pairs $(a_i, a_j) \in \mathcal{A}^2$ with $i < j$
 4: **for** $t = 1, \ldots, T$ **do**
 5: $\quad r \Leftarrow (t - \tau - 1) \mod \lfloor K(K-1)/2\gamma \rfloor$
 6: $\quad$ **if** $r < \frac{K(K-1)}{2}$ **then**
 7: $\quad\quad (a_{I_t}, a_{J_t}) \Leftarrow r$-th pair in $\pi$
 8: $\quad\quad$ Play the pair $(a_{I_t}, a_{J_t})$ and observe $X_{I_t, J_t}^{(t)}$
 9: $\quad\quad n_{I_t, J_t} \Leftarrow n_{I_t, J_t} + 1$
10: $\quad\quad X_{I_t, J_t, n_{I_t, J_t}} \Leftarrow X_{I_t, J_t}^{(t)}$
11: $\quad\quad$ **if** $n_{I_t, J_t} \geq w$ and $|\sum_{s=n_{I_t, J_t}-w+1}^{n_{I_t, J_t}-w/2} X_{I_t, J_t, s} - \sum_{s=n_{I_t, J_t}-w/2+1}^{n_{I_t, J_t}} X_{I_t, J_t, s}| > b$ **then**
12: $\quad\quad\quad \tau \Leftarrow t$
13: $\quad\quad\quad n_{i,j} \Leftarrow 0 \ \forall a_i, a_j \in \mathcal{A}$ with $i < j$
14: $\quad\quad\quad$ Reset $Alg$
15: $\quad\quad$ **end if**
16: $\quad$ **else**
17: $\quad\quad$ RunAndFeed $Alg$
18: $\quad$ **end if**
19: **end for**

---

First, we bound MDB's expected binary strong regret in the stationary setting.

**Lemma 5.** *Let $\tau_1$ be the first detection time. The expected cumulative binary strong regret of MDB in the stationary setting, i.e., $M = 1$, is bounded by*

$$\mathbb{E}\left[R^{\bar{S}}(T)\right] \leq \mathbb{P}(\tau_1 \leq T) \cdot T + \frac{2\gamma K T}{K-1} + \mathbb{E}\left[\tilde{R}^{\bar{S}}(T)\right] .$$

Next, we bound the probability that MDB wrongly triggers the changepoint detection in the stationary case and additionally the probability of MDB missing the changepoint by more than $L/2$ time steps in the non-stationary case with exactly one changepoint.

**Lemma 6.** *Let $\tau_1$ be the first detection time. The probability of MDB raising a false alarm in the stationary setting, i.e., $M = 1$, is bounded by*

$$\mathbb{P}(\tau_1 \leq T) \leq 2T \exp\left(\frac{-2b^2}{w}\right) .$$

**Lemma 7.** *Consider the non-stationary setting with $M = 2$. Then it holds*

$$\mathbb{P}(\tau_1 \geq \nu_1 + L/2 \mid \tau_1 \geq \nu_1) \leq \exp\left(-\frac{wc^2}{2}\right) .$$

We bound MDB's expected regret by an induction over $M$. More precisely, we assume for each inductive step that MDB is started at time step $\nu_{m-1}$ for a particular segment $S_m$. Then we use the bounds on the probabilities that MDB raises a false alarm in $S_m$ (via $p$) and that it misses to trigger the changepoint detection by more than $L/2$ time steps after passing the next changepoint $\nu_m$ (via $q$), respectively. For sake of convenience, we denote $Alg$'s expected cumulative binary strong regret incurred over all stationary segments by

$$R_M^{\bar{S}}(Alg) = \sum_{m=1}^{M} \mathbb{E}\left[\tilde{R}^{\bar{S}}(\nu_{m-1}, \nu_m - 1)\right] .$$

**Theorem 3.** *Let $\tau_m$ be the time step at which the changepoint detection is triggered for the first time with MDB being initialized at $\nu_{m-1}$. Let $p, q \in [0, 1]$ such that $\mathbb{P}(\tau_m < \nu_m) \leq p$ for all $m \leq M$ and $\mathbb{P}(\tau_m \geq \nu_m + L/2 \mid \tau_m \geq \nu_m) \leq q$ for all $m \leq M - 1$. Then the expected cumulative binary strong regret of MDB is bounded by*

$$\mathbb{E}\left[R^{\bar{S}}(T)\right] \leq \frac{ML}{2} + 2T\left(\frac{\gamma K}{K-1} + p + q\right) + R_M^{\bar{S}}(Alg).$$

Finally, we set MDB's parameters in accordance with Assumption 2 and 3, which results in the following bound.

**Corollary 1.** *Setting $\gamma = (K-1)\sqrt{Mw/8T}$, $b = \sqrt{wC/2}$, $c = \sqrt{2C/w}$, and $w$ to the smallest even integer $\geq {}^{8C}\!/\delta^2$ with $C = \log(\sqrt{2T}(2T+1)/\sqrt{M}K)$, it holds that*

$$\mathbb{E}\left[R^{\bar{S}}(T)\right] = \mathcal{O}\left(\frac{K\sqrt{MT\log(\delta T/KM)}}{\delta}\right) + R_M^{\bar{S}}(Alg).$$

MDB requires $M$ and $T$ to be known and is sensible to $\delta$, as smaller changes of entries in the preference matrix between segments impede its ability to detect a changepoint. Nevertheless, if a suitable dueling bandits algorithm is used for $Alg$ such as RUCB or RMED it has a nearly optimal strong regret bound w.r.t. $T$ and $M$ in light of the lower bound shown by Saha & Gupta (2022).

## 5 DETECT

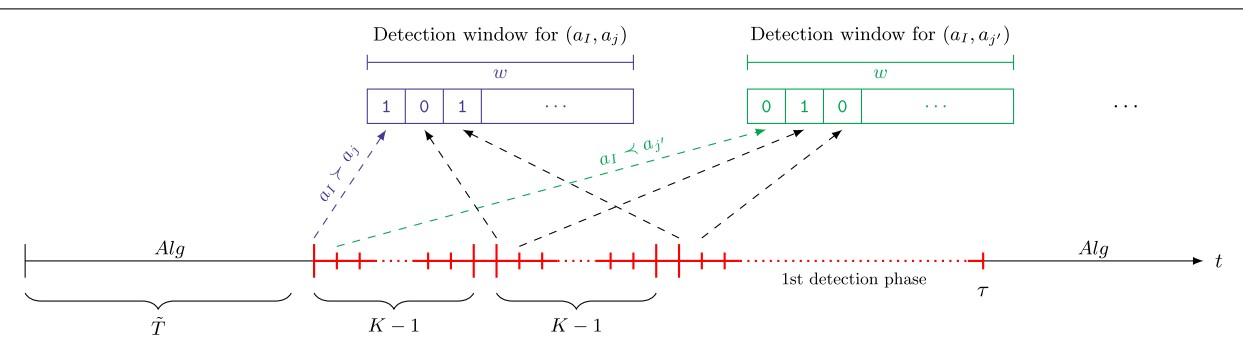

Figure 3: Scheme of DETECT: in the first running phase of $Alg$ that spans the first $\tilde{T}$ time steps only pairs selected by $Alg$ are played. It is followed by a detection phase filling the $K - 1$ many detection windows. The arms $a_j$ and $a'_j$ are the first two of $K - 1$ many arms in the ordering $\pi$. It ends with the raise of an alarm in time step $\tau$ and is followed by a new running phase of $Alg$. All windows are filled after $w(K-1)$ time steps in the detection phase.

The second algorithm utilizing detection-windows that we propose is the *Dueling Explore-Then-Exploit Changepoint Test* (DETECT) algorithm (see Algorithm 3). It is a specialization of MDB (see Section 4) for weak regret. DETECT is given a black-box dueling bandits algorithm $Alg$ for the stationary setting and alternates between two phases: one in which it is running $Alg$ for $\tilde{T}$ many time steps, and a detection phase, in which it scans for the next changepoint and resets $Alg$ upon detection. We take advantage of the opportunity to choose all pairs that contain the best arm without incurring a regret penalty, thus leaving one arm free to be chosen for exploration in the detection phases. DETECT starts by running $Alg$ for a fixed number of time steps $\tilde{T}$, where we assume that interaction with $Alg$ can be done by means of a `RunAndFeed` procedure as in MDB. In addition, we assume that $Alg$ can provide at any time step its suspected Condorcet winner (CW) $a_I$ (to which most dueling bandit algorithms can be extended effortlessly). After running $Alg$ for $\tilde{T}$ time steps and obtaining $Alg$'s suspected CW $a_I$, DETECT initializes a detection phase in which pairs

of the form $(a_I, a_j)$ are played with $a_j$ alternating between all $K-1$ arms other than $a_I$. The binary dueling outcomes in these detection steps are inserted into $K-1$ windows, one for each pair $(a_I, a_j)$ having length $w$. The detection phase ends as soon as a changepoint is detected, which follows the same principle as in MDB but using the $K-1$ detection windows for the pairs $(a_I, a_j)$ instead. The end of the detection phase is followed by a new running phase of a reinitialized instance of *Alg*. This cycle continues until the time horizon is reached. If $a_I$ is indeed the optimal arm of the current segment, we suffer zero regret in the part of the detection phase that overlaps with the stationary segment in which it started. Thus, we simultaneously perform regret minimization and changepoint detection. An illustration of DETECT is given in Fig. 3.

---

**Algorithm 3** Dueling Explore-Then-Exploit Changepoint Test (DETECT)

---

**Require:** $K, T, \tilde{T}$, even $w \in \mathbb{N}, b > 0$
1: $\tau \Leftarrow 0$
2: **for** $t = 1, \ldots, T$ **do**
3:     **if** $t - \tau \leq \tilde{T}$ **then**
4:         RunAndFeed *Alg*
5:     **else**
6:         **if** $t - \tau = \tilde{T} + 1$ **then**
7:             $I \Leftarrow$ index of suspected Condorcet winner by *Alg*
8:             $n_{I,j} \Leftarrow 0 \ \forall a_j \neq a_I \in \mathcal{A}$
9:             Fix any ordering $\pi$ of arms $a_j \neq a_I \in \mathcal{A}$
10:         **end if**
11:         $r \Leftarrow t - \tau - \tilde{T} - 1 \mod (K-1)$
12:         $a_{J_t} \Leftarrow r$-th arm in $\pi$
13:         Play $(a_I, a_{J_t})$ and observe $X_{I,J_t}^{(t)}$
14:         $n_{I,J_t} \Leftarrow n_{I,J_t} + 1$
15:         $X_{I,J_t,n_{I,J_t}} \Leftarrow X_{I,J_t}^{(t)}$
16:         **if** $n_{I,J_t} \geq w$ **and** $|\sum_{s=n_{I,J_t}-w+1}^{n_{I,J_t}-w/2} X_{I,J_s,s} - \sum_{s=n_{I,J_t}-w/2+1}^{n_{I,J_t}} X_{I,J_s,s}| > b$ **then**
17:             $\tau \Leftarrow t$
18:             Reset *Alg*
19:         **end if**
20:     **end if**
21: **end for**

---

**Non-Stationary Regret Analysis.** Our theoretical analysis of DETECT's expected binary weak regret follows a similar approach to MDB (see Appendix E for the proofs). However, due to its different mechanism, we redefine the number of time steps needed to fill all of DETECT's $K-1$ detection windows completely by

$$L' := w(K-1).$$

Due to its focus on weak regret, we modified the definition of $\delta$ to $\delta_*$ in Section 2 as well, since only changes in the preference matrix between stationary segments that relate to the winning probabilities of the CW are relevant. Note that DETECT's mechanism attempts to account for this very insight by tracking changes only at the entries related to $a_I$, the suspected CW by *Alg*. Additionally, we define $p_{\tilde{T}}^{(m)}$ as the probability that $a_I$, the suspected CW returned by *Alg* after $\tilde{T}$ time steps, is the actual CW of segment $m$. Based on that, let $p_{\tilde{T}} \leq \min_m p_{\tilde{T}}^{(m)}$ be a lower bound valid for all segments. Again, we impose some technical assumptions:

**Assumption 1:** $|S_m| \geq \tilde{T} + \frac{3}{2}L'$ for all $m \in \{1, \ldots, M\}$

**Assumption 2:** $\delta_* \geq \frac{2b}{w} + c$ for some $c > 0$

Assumption (1) requires a minimum length for all stationary segments exceeding $\tilde{T}$ to guarantee that DETECT is able to detect changepoints. Otherwise, the number of observed dueling outcomes during the exploitation phase might be too small to give sufficient guarantees for the validity of the detection mechanism.

Analogously to Assumption (2) for MDB (and also to be found for MUCB in Cao et al. (2019)), Assumption (2) restricts $\delta_*$ such that a changepoint can be detected after a short delay with a certain probability. To begin with, we bound DETECT's expected binary weak regret in the stationary setting, which unlike Lemma 5 now depends on whether $Alg$ returns with $a_I$ the true CW.

**Lemma 8.** *Let $\tau_1$ be the first detection time and $a_I$ be the first suspected CW returned by Alg. Given $\tilde{T} \leq T$, the expected cumulative binary weak regret of DETECT in the stationary setting, i.e., $M = 1$, is bounded by*

$$\mathbb{E}\left[R^{\bar{W}}(T)\right] \leq \mathbb{E}\left[\tilde{R}^{\bar{W}}(\tilde{T})\right] + \mathbb{P}(\tau_1 \leq T \vee a_I \neq a_{1^*}) \cdot (T - \tilde{T}).$$

The following Lemmas are adequate to Lemma 6 and 7 with only minor changes incorporating $a_I$.

**Lemma 9.** *Let $\tau_1$ be as in Lemma 8. The probability of DETECT raising a false alarm in the stationary setting, i.e., $M = 1$, given that $a_I = a_{1^*}$ is bounded by*

$$\mathbb{P}\left(\tau_1 \leq T \mid a_I = a_{1^*}\right) \leq \frac{2(T - \tilde{T})}{\mathbb{P}\left(a_I = a_{1^*}\right)} \cdot \exp\left(\frac{-2b^2}{w}\right).$$

**Lemma 10.** *Consider the non-stationary setting with $M = 2$. Let $\tau_1$ be as in Lemma 8. Then, the following holds*

$$\mathbb{P}\left(\tau_1 \geq \nu_1 + L'/2 \mid \tau_1 \geq \nu_1, a_I = a_{1^*}\right) \leq \exp\left(-\frac{wc^2}{2}\right).$$

The proof of the following result is an adaption of Theorem 3 for MDB. We slightly change the meaning of $p$ and $q$: for a considered segment $S_m$, $p$ is now a bound on the probability that DETECT raises a false alarm given that it started at time step $\nu_{m-1}$ and $Alg$ returned the true CW. Similarly, $q$ bounds the probability that DETECT misses to trigger the changepoint detection by more than $L'/2$ time steps after passing the next changepoint $\nu_m$. Further, let

$$R_M^{\bar{W}}(Alg) = \sum_{m=1}^{M} \mathbb{E}\left[\tilde{R}^{\bar{W}}(\nu_{m-1}, \nu_{m-1} + \tilde{T} - 1)\right]$$

be $Alg$'s expected cumulative weak regret incurred over all $\tilde{T}$-capped stationary segments $\{\nu_{m-1}, \ldots, \nu_{m-1} + \tilde{T} - 1\}$.

**Theorem 4.** *Let $\tau_m$ be the time step at which the changepoint detection is triggered for the first time with DETECT being initialized at $\nu_{m-1}$. Let $p, q \in [0, 1]$ be such that $\mathbb{P}\left(\tau_m < \nu_m \mid a_{I_m} = a_{m^*}\right) \leq p$ for all $m \leq M$ and $\mathbb{P}\left(\tau_m \geq \nu_m + L'/2 \mid \tau_m \geq \nu_m, a_{I_m} = a_{m^*}\right) \leq q$ for all $m \leq M - 1$. Then,*

$$\mathbb{E}\left[R^{\bar{W}}(T)\right] \leq \frac{ML'}{2} + (1 - p_{\tilde{T}} + pp_{\tilde{T}} + q)MT + R_M^{\bar{W}}(Alg).$$

Finally, we set DETECT's parameters in accordance with the assumptions above which results in the following bound.

**Corollary 2.** *Setting $b = \sqrt{2w \log T}$, $c = \sqrt{8 \log T / w}$, and $w$ to the lowest even integer $\geq 32 \log T / \delta_*^2$, the expected cumulative binary weak regret of DETECT is bounded by*

$$\mathbb{E}\left[R^{\bar{W}}(T)\right] = \mathcal{O}\left(\frac{KM \log T}{\delta_*^2}\right) + (1 - p_{\tilde{T}})MT + R_M^{\bar{W}}(Alg).$$

Note that the presented bound is not necessarily linear in $T$, since we still have the freedom to set $\tilde{T}$, effecting the probability $p_{\tilde{T}}$ with which the used black-box algorithm $Alg$ returns the Condorcet Winner. We analyze $p_{\tilde{T}}$ for BtW and Winner-Stays in Appendix F, and derive a suitable choice of $\tilde{T}$ in Corollary 6, which leads to a bound of

$$\mathbb{E}\left[R^{\bar{W}}(T)\right] = \mathcal{O}\left(\log T / \delta_*^2 MK + M \log T\right) + R_M^{\bar{W}}(Alg).$$

Similar to our remark in Section 3 about $\Delta$, we would like to point out that the knowledge of $\delta_*$ (and $\delta$ for MDB) is necessary in order to derive parameterizations for which we can prove theoretical results. Again, this is common practice in the multi-armed bandit literature.

# 6 Weak Regret Lower Bounds

The following result provides worst-case lower bounds for the weak regret for non-stationary and stationary cases complementing the lower bound for the strong regret by Saha & Gupta (2022).

**Theorem 5.** *(Proof in Appendix G) For every algorithm exists an instance of*

**(i)** *the piecewise-stationary dueling bandits problem with $T$, $K$, and $M$ fulfilling $M(K-1) \leq 9T$ such that the algorithm's expected weak regret is in $\Omega(\sqrt{KMT})$.*

**(ii)** *the stationary dueling bandits problem with $T$ and $K$ fulfilling $K - 1 \leq 9T$ such that the algorithm's expected weak regret is in $\Omega(\sqrt{KT})$.*

Note that the presented lower bounds do not form a contradiction with Theorem 1, 2, or 4, since they are problem-independent and do not take quantities specific to each problem instance like $\Delta$ and $\delta^*$ into account, while in turn, these are included in the latter ones.

# 7 Empirical Results

In the following we evaluate BtWR, MDB, and DETECT empirically on synthetic problem instances by comparing them to dueling bandits algorithms for the stationary setting w.r.t. their cumulative regret. Besides the regret in dependence of $T$ and $M$, we are also interested in the shape of the regret curves for a specific scenario in order to give a glimpse of how the algorithms cope with the challenge of non-stationarity. We have generated for each chosen scenario 500 random problem instances, each consisting of a sequence of $M$ randomly generated preference matrices. For each matrix $P^{(m)}$ a Condorcet winner $a_{m^*}$ exists with one of its winning probabilities being exactly $1/2 + \Delta$ and the others being drawn uniformly at random from $[1/2 + \Delta, 1]$. All winning probabilities between pairs that do not include $a_{m^*}$ are drawn uniformly at random from $[0, 1]$. The next matrix $P^{(m+1)}$ is manipulated such that at least one of $a_{m^*}$ winning probabilities changes by $\delta$. We have used $\Delta = 0.1$ and $\delta = 0.6$ for most scenarios, but also investigate the algorithms' regret curves depending on these quantities within a range of reasonable values. The Condorcet winner changes with each new preference matrix. The changepoints are distributed evenly across the time horizon, implying equal segment lengths in each scenario. This is done with the intention to obtain meaningful regret averages for a single parameter specification. For further empirical results see Fig. 8-13.

## 7.1 Weak Regret

We compare BtWR and DETECT (parameterized as given in Corollary 2) with Winner-Stays (WS) (Chen & Frazier, 2017) as the incorporated black-box algorithm against BtW in Fig. 4. Both, BtWR and DETECT improve on its stationary competitor BtW: not only is the cumulative regret lower in single scenarios, but they also show a desirable dependency on $T$. In addition with the linear growth w.r.t. $M$, this confirms or theoretical results in Theorem 2 and 4. BtW experiences difficulties adapting to its changing environment, visible by its increasing regret jumps at each new changepoint, caused by the time it takes to adapt to the new preference matrix. In comparison, BtWR and DETECT surmount the difficulty of non-stationarity without an increasing regret penalty. Further of interest are the regret curves depending on $\Delta$ and $\delta^*$. The cumulative regret of DETECT and especially of BtWR shrinks with increasing gap $\Delta$. The latter observation confirms our result in Theorem 2. In contrast to that, BtW seems to have, at least in the context of non-stationarity, constant performance across a broad range of $\Delta$ values. Inspecting the dependency on $\delta^*$, it is striking how DETECT's regret curve drops with increasing $\delta^*$, which is in accordance with our result in Corollary 2, while BtW and BtWR show constant performance.

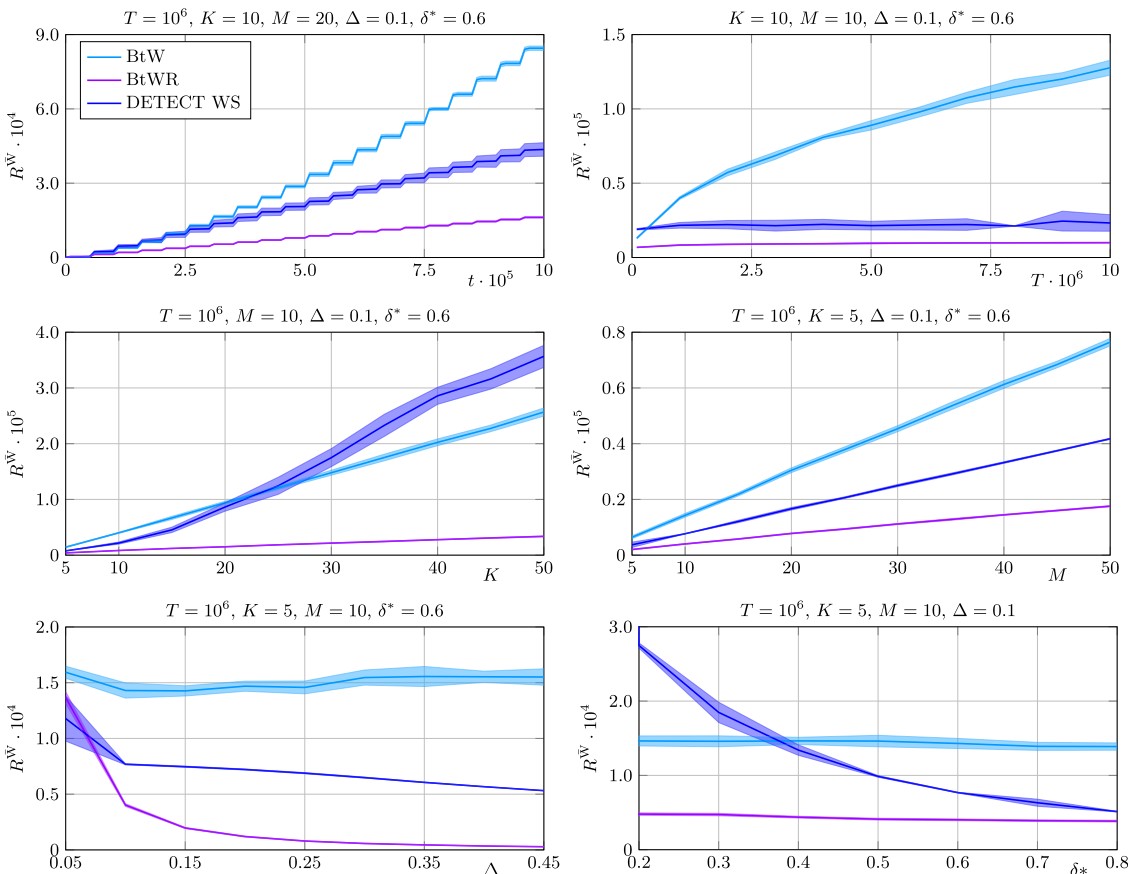

Figure 4: Cumulative binary weak regret averaged over 500 random problem instances with shaded areas showing the standard deviation across 10 groups: regret over time for fixed $T$ (Top left), dependencies on $T$ (Top right), $K$ (Center left), $M$ (Center right), $\Delta$ (Bottom left), $\delta^*$ (Bottom right).

## 7.2 Strong Regret

We compare MDB (parameterized as given in Corollary 1) with Winner Stays Strong (WSS) (Chen & Frazier, 2017) as the incorporated black-box algorithm against WSS itself and DEX3.S (Saha & Gupta, 2022) in Fig. 5. Both instances of WSS have exploitation parameter $\beta = 1.05$. MDB accumulates less regret than WSS and DEX3.S across single scenarios with similar sublinear dependencies on $M$ and $T$. Unlike MDB and DEX3.S, the considered stationary algorithm WSS shows increasing regret jumps, indicating its maladjustment for the non-stationary setting. Further, we observe how MDB's cumulative regret decreases for larger $\delta^*$, whereas the drop in regret depending on $\Delta$ is rather negligible.

## 7.3 Sensitivity Analysis for BtWR

Our satisfying regret bound in Theorem 2 and empirical findings in Section 7.1, which demonstrate BtWR's superiority in the non-stationary setting over its competitors, rely on the assumption that the gap $\Delta$ of the particular problem instance is known. But since this is not guaranteed in practice, one might ask by how much BtWR's performance detoriates if the unknown $\Delta$ is only estimated. In order to answer this question empirically, we conducted an evaluation of BtWR with the same setup as described above. With the only difference being that we have run multiple instantiations of BtWR, parameterized with different estimates of $\Delta$, on multiple problem classes that differ in their true gap $\Delta$. We consider for both the true gap $\Delta$ and its estimate values ranging from 0.025 to 0.25, which we find reasonable, as values outside our chosen interval are unlikely to bet met in practice.

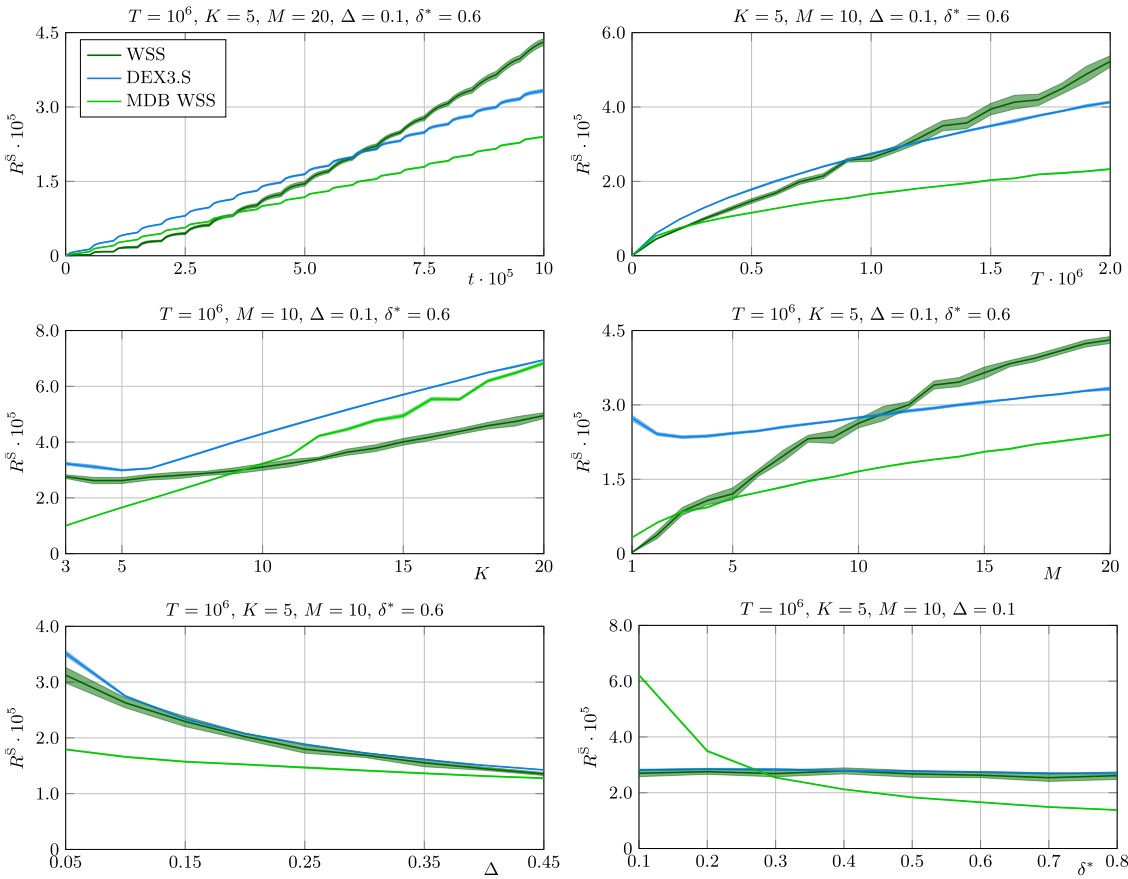

Figure 5: Cumulative binary strong regret averaged over 500 random problem instances with shaded areas showing the standard deviation across 10 groups: regret over time for fixed $T$ (Top left), dependencies on $T$ (Top right), $K$ (Center left), $M$ (Center right), $\Delta$ (Bottom left), $\delta^*$ (Bottom right).

The performance of different BtWR $\Delta$ estimates, measured in cumulative regret, depending on the true $\Delta$ is depicted in Fig. 6. The estimates $\Delta = 0.05$ and $\Delta = 0.075$ outperform the parameter-free BtW algorithm significantly across all $\Delta$ values in the considered range. Further, if one possesses additional domain knowledge related to the problem instance at hand, stating that $\Delta$ is not lower than 0.1, then BtWR parametrized with $\Delta$ between 0.1 and 0.2 achieves even lower regret numbers.

Next, we would like to present our results from a slightly different perspective. Fig. 7 shows the cumulative regret achieved by BtWR for a particular true gap $\Delta$ depending on the chosen estimate BtWR $\Delta$. Again, we observe that the estimates $\Delta = 0.05$ and $\Delta = 0.075$ perform well across the whole considered range. Interestingly, the lower the estimate is chosen to be, the less relevant the actual gap $\Delta$ becomes.

Further, one can notice how each graph for a particular gap reaches its minimum at a point, at which BtWR $\Delta$ slightly overestimates the true gap. The attentive and sharp-eyed reader has already observed this phenomenon in Fig. 6. We are confident to assert that this does not occur just by chance, but has a theoretical background that we are able to unveil. Our main result in Theorem 2 and its proof in Appendix C allow us to draw the conclusion that parameterizing BtWR with an estimate $\bar{\Delta}$ that underestimates $\Delta$, i.e. $\bar{\Delta} \leq \Delta$, leads to a regret bound of

$$\mathbb{E}\left[R^{\bar{W}}(T)\right] \leq \frac{22KM}{\bar{\Delta}^2} \log(K + T).$$

in the non-stationary setting. This bound grows the further $\bar{\Delta}$ underestimates $\Delta$. From this observation we dare to infer, although we cannot prove it, that the cumulative regret should shrink if $\bar{\Delta}$ slightly overestimates $\Delta$, since then the denominator in the bound grows.

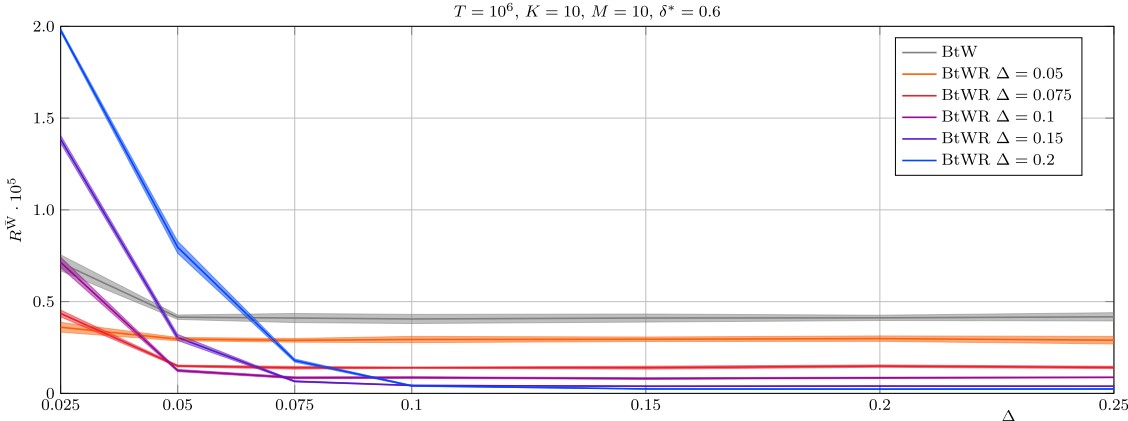

Figure 6: Cumulative binary weak regret in dependence of $\Delta$ averaged over 100 random problem instances with shaded areas showing the standard deviation across 5 groups. Except for BtW, each graph shows the performance of BtWR initialized with an estimate of $\Delta$.

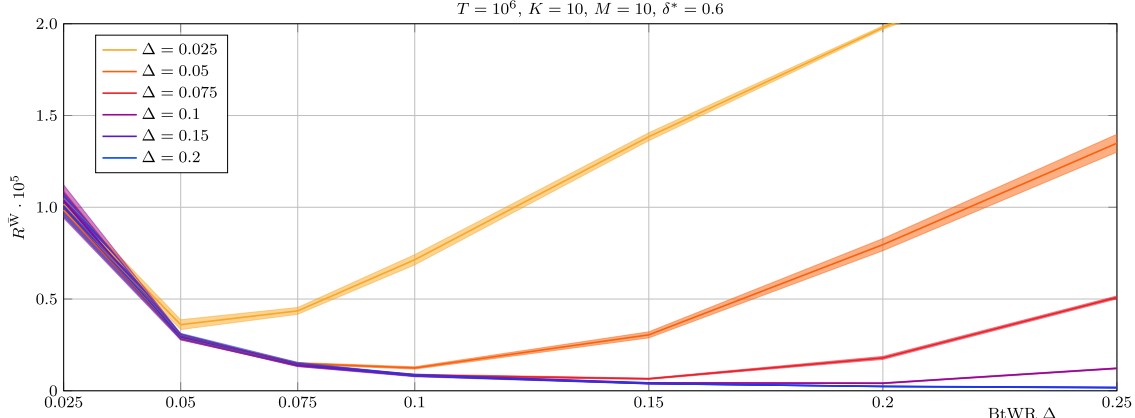

Figure 7: Cumulative binary weak regret averaged over 100 random problem instances with shaded areas showing the standard deviation across 5 groups. Each graph shows the performance of BtWR initialized with different $\Delta$ estimates depending on a fixed $\Delta$ of the problem instance.

## 8 Conclusion

We considered the piecewise-stationary dueling bandits problem with $M$ segments, for which we proposed and analyzed three actively adaptive algorithms. For BtWR, we have shown satisfactory regret bounds for weak regret in the non-stationary and stationary setting. It stands out from previous non-stationary algorithms by not requiring the time horizon $T$ or $M$ to be given, which allows it to be applied when these quantities are not known upfront. With MDB and DETECT, we have presented two meta-algorithms based on detection-windows and incorporating black-box dueling bandit algorithms. Their regret bounds can be interpreted as the sum of the black-box algorithm's regret if it would know the changepoints plus a penalty of non-stationarity. In addition, we have proven worst-case lower bounds on the expected weak regret. Our empirical results clearly suggest that all three algorithms outperform standard dueling bandits algorithms in the non-stationary setting.

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

# A List of Symbols

Table 2: List of frequently symbols used throughout the paper.

| | **Problem setting of piecewise-stationary dueling bandits** |
|---|---|
| $\mathcal{A}$ | set of arms |
| $K$ | number of arms |
| $a_i$ | $i$-th arm |
| $T$ | time horizon |
| $M$ | number of stationary segments |
| $\nu_m$ | $m$-th changepoint |
| $S_m$ | $m$-th stationary segment |
| $P^{(m)}$ | preference matrix of the $m$-th stationary segment |
| $p_{i,j}^{(m)}$ | probability that arm $a_i$ is preferred over $a_j$ in the $m$-th segment |
| $X_{I_t,J_t}^{(t)}$ | binary dueling outcome between $a_{I_t}$ and $a_{J_t}$ |
| $a_{m*}$ | Condorcet winner of the $m$-th segment |
| $\Delta_{i,j}^{(m)}$ | calibrated pereference probability of $a_i$ over $a_j$ in the $m$-th segment |
| $\Delta_i^{(m)}$ | suboptimality gap of $a_i$ in the $m$-th segment |
| $\Delta$ | minimal suboptimality gap |
| $\delta_{i,j}^{(m)}$ | change between $a_i$ and $a_j$ at the $m$-th changepoint |
| $\delta^{(m)}$ | $m$-th segmental change |
| $\delta$ | minimal segmental change |
| $\delta_*^{(m)}$ | segmental change w.r.t. CW |
| $\delta_*$ | minimal segmental change w.r.t. CW |
| | **Regret measures** |
| $r_t^S$ | instantaneous strong regret |
| $r_t^W$ | instantaneous weak regret |
| $r_t^{\bar{S}}$ | instantaneous binary strong regret |
| $r_t^{\bar{W}}$ | instantaneous binary weak regret |
| $R^v(T)$ | cumulative regret for regret type $v$ |
| $\tilde{R}^v(t_1, t_2)$ | cumulative regret of type $v$ of $Alg$ run on its own from $t_1$ to $t_2$ |
| $\tilde{R}^v(T)$ | cumulative regret of type $v$ of $Alg$ run on its own |
| $R_M^v(Alg)$ | expected cumulative regret of type $v$ of $Alg$ over all stationary segments |
| | **Algorithm-specific symbols** |
| $\ell_c$ | number of required wins to end a round of BtWR with round counter $c$ |
| $\lambda$ | probability of CW to lose a remaining round of BtWR as current incumbent |
| $\tau_n$ | first time step of BtWR's $n$-th round |
| $Alg$ | utilized black-box algorithm by MDB and DETECT |
| $\gamma$ | exploration rate of MDB |
| $w$ | window length of MDB and DETECT |
| $b$ | threshold of MDB and DETECT |
| $L$ | number of time steps to fill all of MDB's detection windows |
| $L'$ | number of time steps to fill all of DETECT's detection windows |
| $\tilde{T}$ | phase length in which DETECT runs $Alg$ |
| $p_{\tilde{T}}^{(m)}$ | probability that DETECT's suspected CW of the $m$-th segment is correct |

# B  BtWR Stationary Weak Regret Analysis

Let $L_n$ be the index of the arm that lost the $n$-th round and $c_n$ the value of $c$ during the $n$-th round. Finally, let $N = \sup\{n \in \mathbb{N} \mid \tau_n \leq T\}$ be the last round to be started.

**Lemma** 1 *Choosing $\ell_c = \left\lceil \frac{1}{4\Delta^2} \log \frac{c(c+1)e}{\lambda} - \frac{1}{2} \right\rceil$ for any $\lambda > 0$ and given that $a_1$ is the incumbent of the $n$-th round with $c_n = 1$, the probability of $a_1$ losing at least one of the remaining rounds is at most*

$$\mathbb{P}\left( \bigcup_{n'=n}^{N} \{L_{n'} = 1\} \;\middle|\; I_{\tau_n} = 1, c_n = 1 \right) \leq \lambda.$$

*Proof.* The $n'$-th round with round counter $c_{n'}$ can be seen as an experiment in which $2\ell_{c_{n'}} - 1$ coins are thrown i.i.d. with the probability of head (representing $a_{I_{\tau_{n'}}}$ to win a duel against $a_{J_{\tau_{n'}}}$) being $p_{I_{\tau_{n'}}, J_{\tau_{n'}}}$. The round is lost by $a_{I_{\tau_{n'}}}$ if at most $\ell_{c_{n'}}$ heads are thrown. Define the binomially distributed random variables $B_{c_{n'}} \sim \mathrm{Bin}(2\ell_{c_{n'}} - 1, p_{1, J_{\tau_{n'}}})$ and $B'_{c_{n'}} \sim \mathrm{Bin}(2\ell_{c_{n'}} - 1, \frac{1}{2} + \Delta)$. We derive:

$$
\begin{aligned}
\mathbb{P}(L_{n'} = 1 \mid I_{\tau_{n'}} = 1) &= \mathbb{P}(B_{c_{n'}} \leq \ell_{c_{n'}} - 1) \\
&\leq \mathbb{P}(B'_{c_{n'}} \leq \ell_{c_{n'}} - 1) \\
&= \mathbb{P}\left( B'_{c_{n'}} \leq \left( \frac{1}{2} + \Delta - \left( \frac{1}{2} + \Delta - \frac{\ell_{c_{n'}} - 1}{2\ell_{c_{n'}} - 1} \right) \right) \cdot (2\ell_{c_{n'}} - 1) \right) \\
&\leq \exp\left( -2(2\ell_{c_{n'}} - 1) \cdot \left( \frac{1}{2} + \Delta - \frac{\ell_{c_{n'}} - 1}{2\ell_{c_{n'}} - 1} \right)^2 \right) \\
&\leq \exp\left( 2\Delta^2 (1 - 2\ell_{c_{n'}}) \right),
\end{aligned}
$$

where we utilized Hoeffding's inequality and the fact that $\Delta \leq \frac{1}{2}$. Further, we derive:

$$
\begin{aligned}
&\mathbb{P}\left( \bigcup_{n'=n}^{N} \{L_{n'} = 1\} \;\middle|\; I_{\tau_n} = 1, c_n = 1 \right) \\
&= 1 - \prod_{n'=n}^{N} \mathbb{P}\left( L_{n'} \neq 1 \;\middle|\; \{I_{\tau_n} = 1, c_n = 1\} \cap \bigcap_{i=n}^{n'-1} \{L_i \neq 1\} \right) \\
&= \mathbb{P}\left( \bigcup_{n'=n}^{N} \{L_{n'} = 1 \mid I_{\tau_{n'}} = 1, c_{n'} = n' - n + 1\} \right) \\
&\leq \sum_{n'=n}^{N} \mathbb{P}(L_{n'} = 1 \mid I_{\tau_{n'}} = 1, c_{n'} = n' - n + 1) \\
&\leq \sum_{n'=1}^{\infty} e^{2\Delta^2 (1 - 2\ell_{n'})} \\
&\leq \sum_{n'=1}^{\infty} \frac{\lambda e^{4\Delta^2}}{n'(n' + 1)e} \\
&= \frac{\lambda e^{4\Delta^2}}{e} \\
&\leq \lambda,
\end{aligned}
$$

where we used the independence of events for the second equality and the previously shown bound on $\mathbb{P}(L_{n'} = 1 \mid I_{\tau_{n'}} = 1)$ for the second inequality. $\square$

**Lemma** 2  *Consider the stationary setting and let $\bar{n}$ be an arbitrary round with $I_{\tau_{\bar{n}}} \neq 1$. Let $n_* = \inf\{n > \bar{n} \mid I_{\tau_n} = 1\}$ be the first round after $\bar{n}$ in which $a_1$ is the incumbent. Choosing $\ell_c = \left\lceil \frac{1}{4\Delta^2} \log \frac{c(c+1)e}{\lambda} - \frac{1}{2} \right\rceil$ for any $\lambda > 0$, the expected number of time steps needed for $a_1$ to become the incumbent is bounded by*

$$\mathbb{E}[\tau_{n_*} - \tau_{\bar{n}}] \leq \frac{6K}{\Delta^2} \log \sqrt{\frac{e}{\lambda}} (K + c_{\bar{n}} - 1).$$

*Proof.* Let $n_1 = \inf\{n \geq \bar{n} \mid J_{\tau_n} = 1\}$ be the first round from $\bar{n}$ onward in which $a_1$ is the current challenger and $n_i = \inf\{n > n_{i-1} \mid J_{\tau_n} = 1\}$ be the $i$-th round for $i \geq 2$. Let $U$ be a random variable denoting the number of times $a_1$ has to become challenger before winning a round, i.e., $n_U = n_* - 1$. Since the queue contains $K - 2$ many arms, it takes at most $K - 2$ rounds for $a_1$, potentially sitting in last place of the queue, to become the challenger and one more round to get to the end of the queue or to get promoted to the incumbent. Consequently, we have $n_1 - \bar{n} \leq K - 2$, and $n_i - n_{i-1} \leq K - 1$ for all $i \in \{2, \ldots, U\}$. Hence, we obtain:

$$n_* - \bar{n} = (n_1 - \bar{n}) + (n_* - n_U) + \sum_{i=2}^{U} n_i - n_{i-1} \leq (K-1)U.$$

For the number of time steps depending on $U$ we further conclude:

$$
\begin{aligned}
\tau_{n_*} - \tau_{\bar{n}} &\leq \sum_{c=1}^{n_* - \bar{n}} 2\ell_{c_{\bar{n}} + c - 1} - 1 \\
&\leq \sum_{c=1}^{(K-1)U} \frac{1}{2\Delta^2} \log \frac{(c_{\bar{n}} + c - 1)(c_{\bar{n}} + c)e}{\lambda} \\
&\leq \frac{KU}{2\Delta^2} \log \frac{(c_{\bar{n}} + (K-1)U - 1)(c_{\bar{n}} + (K-1)U)e}{\lambda} \\
&\leq \frac{KU}{\Delta^2} \log \sqrt{\frac{e}{\lambda}} (c_{\bar{n}} + KU - 1) \\
&\leq \frac{KU^2}{\Delta^2} \log \sqrt{\frac{e}{\lambda}} (K + c_{\bar{n}} - 1).
\end{aligned}
$$

For all rounds $i$ holds that $\mathbb{P}(L_{n_i} \neq 1 \mid J_{\tau_{n_i}} = 1) \geq \frac{1}{2}$ since the probability of $a_1$ winning a duel against any other arm is at least $\frac{1}{2} + \Delta$. Thus, we can use the geometrically distributed random variable $V$ with parameter $p = \frac{1}{2}$ to bound the expected value of $U^2$ by $\mathbb{E}[U^2] \leq \mathbb{E}[V^2]$. Finally, we derive the stated claim:

$$
\begin{aligned}
\mathbb{E}[\tau_{n_*} - \tau_{\bar{n}} \mid \tau_{n_*} \leq \nu_m] &\leq \mathbb{E}\left[ \frac{KU^2}{\Delta^2} \log \sqrt{\frac{e}{\lambda}} (K + c_{\bar{n}} - 1) \right] \\
&= \frac{\mathbb{E}[U^2]K}{\Delta^2} \log \sqrt{\frac{e}{\lambda}} (K + c_{\bar{n}} - 1) \\
&\leq \frac{\mathbb{E}[V^2]K}{\Delta^2} \log \sqrt{\frac{e}{\lambda}} (K + c_{\bar{n}} - 1) \\
&= \frac{6K}{\Delta^2} \log \sqrt{\frac{e}{\lambda}} (K + c_{\bar{n}} - 1),
\end{aligned}
$$

where we used the fact that $\mathbb{E}[V^2] = \frac{2-p}{p^2} = 6$. $\qquad\square$

**Theorem** 1  *Setting $\lambda = \frac{1}{e}$ and thus choosing $\ell_c = \left\lceil \frac{1}{4\Delta^2} \log c(c+1)e^2 - \frac{1}{2} \right\rceil$, the expected cumulative binary weak regret of BtWR in the stationary setting is bounded by*

$$\mathbb{E}\left[ R^{\bar{W}}(T) \right] \leq \frac{20K \log K}{\Delta^2}.$$

*Proof.* Let $n_1 = \inf\{n \in \mathbb{N} \mid I_{\tau_n} = 1\}$ be the first round in which $a_1$ is the incumbent. Further define inductively $n_i = \inf\{n > m_{i-1} \mid I_{\tau_n} = 1\}$ for all $i \in \{2, \ldots, N'\}$ and $m_i = \inf\{n > n_i \mid I_{\tau_n} \neq 1\}$ for all $i \in \{1, \ldots, M'\}$ with $N'$ being the number of times $a_1$ becomes the incumbent (including the possibility of starting as such in the first round) and $M'$ the number of times $a_1$ loses a round as the current incumbent. The sequence of rounds can be partitioned in alternating subsequences: the ones that have $a_1$ as their current incumbent and those that not. This allows us to reformulate the incurred regret as:

$$R^{\bar{W}}(T) = \begin{cases} R(\tau_1, \tau_{n_1} - 1) + R(\tau_{n_{N'}}, T) \\ \quad + \sum_{i=1}^{M'} R(\tau_{m_i}, \tau_{n_{i+1}} - 1) + \sum_{i=1}^{N'-1} R(\tau_{n_i}, \tau_{m_i} - 1) & \text{if } M' = N' - 1 \\ R(\tau_1, \tau_{n_1} - 1) + R(\tau_{m_{M'}}, T) \\ \quad + \sum_{i=1}^{M'-1} R(\tau_{m_i}, \tau_{n_{i+1}} - 1) + \sum_{i=1}^{N'-1} R(\tau_{n_i}, \tau_{m_i} - 1) & \text{if } M' = N' \end{cases}$$

$$\leq \begin{cases} \tau_{n_1} - \tau_1 + \sum_{i=1}^{M'} \tau_{n_{i+1}} - \tau_{m_i} & \text{if } M' = N' - 1 \\ \tau_{n_1} - \tau_1 + T + 1 - \tau_{m_{M'}} + \sum_{i=1}^{M'-1} \tau_{n_{i+1}} - \tau_{m_i} & \text{if } M' = N' \end{cases},$$

where we used the fact that BtWR suffers no regret in sequences of rounds with $a_1$ being the current incumbent. By applying Lemma 2 with $c_{\bar{n}} = 1$ due to $c_{m_i} = 1$ for the sequences of rounds not having $a_1$ as its incumbent, we obtain for the expected regret:

$$\mathbb{E}\left[R^{\bar{W}}(T)\right] = \mathbb{E}_{M'}\left[\mathbb{E}\left[R^{\bar{W}}(T) \mid M'\right]\right]$$

$$\leq \mathbb{E}_{M'}\left[\max\left\{\mathbb{E}[\tau_{n_1} - \tau_1] + \sum_{i=1}^{M'} \mathbb{E}[\tau_{n_{i+1}} - \tau_{m_i}],\right.\right.$$

$$\left.\left.\mathbb{E}[\tau_{n_1} - \tau_1] + \mathbb{E}[T + 1 - \tau_{m_{M'}}] + \sum_{i=1}^{M'-1} \mathbb{E}[\tau_{n_{i+1}} - \tau_{m_i}]\right\}\right]$$

$$\leq \mathbb{E}_{M'}\left[\frac{6K}{\Delta^2} \log\left(\sqrt{\frac{e}{\lambda}}K\right) \cdot (M' + 1)\right]$$

$$= \frac{6K}{\Delta^2} \log\left(\sqrt{\frac{e}{\lambda}}K\right) \cdot \mathbb{E}[M' + 1].$$

Note that $c_1 = 1$ and $c_{m_i} = 1$ holds for all $i \in \{1, \ldots, M'\}$ because the incumbent $I_{\tau_{m_i - 1}} = 1$ has lost the previous round. Finally, we can bound the expectation of $M'$ by $\mathbb{E}[M'] \leq \mathbb{E}[X]$ where $X$ is a discrete random variable with $\mathbb{P}(X = x) = (1 - \lambda) \cdot \lambda^x$ for all $x \in \mathbb{N}_0$. This is justified by Lemma 1, guaranteeing that the probability of $a_1$ losing a round once it is incumbent is at most $\lambda$. Let $Y$ be a random variable with $Y = X + 1$, then it holds $Y \sim \text{Geo}(1 - \lambda)$. Since $\mathbb{E}[Y] = \frac{1}{1-\lambda}$, we have $\mathbb{E}[M' + 1] \leq \frac{1}{1-\lambda}$. We conclude the result by assuming $K \geq 3$, otherwise $\mathbb{E}[R^{\bar{W}}(T)] = 0$ since no pair of arms would cause any regret, and plugging in $\lambda = \frac{1}{e}$:

$$\mathbb{E}\left[R^{\bar{W}}(T)\right] \leq \frac{6K}{\Delta^2} \log\left(\sqrt{\frac{e}{\lambda}}K\right) \cdot \mathbb{E}[M' + 1]$$

$$= \frac{6K}{(1 - \lambda)\Delta^2} \log\sqrt{\frac{e}{\lambda}}K$$

$$\leq \frac{20K \log K}{\Delta^2}.$$

$\square$

## C  BtWR Regret Analysis Non-stationary Setting

As before, let $L_n$ be the index of the arm that lost the $n$-th round and $c_n$ the value of $c$ during the $n$-th round.

**Lemma 3**  *Consider an arbitrary segment $S_m$. Let $N = \sup\{n \mid \{\tau_n, \tau_{n+1} - 1\} \subset S_m\}$ be the last round to be started and finished within $S_m$. Choosing $\ell_c = \left\lceil \frac{1}{4\Delta^2} \log \frac{c(c+1)e}{\lambda} - \frac{1}{2} \right\rceil$ for any $\lambda > 0$ and given that $a_{m^*}$ is the incumbent of the $n$-th round with $c_n = 1$, the probability of $a_{m^*}$ losing at least one of the remaining rounds is at most*

$$\mathbb{P}\left( \bigcup_{n'=n}^{N} \{L_{n'} = 1\} \ \middle| \ I_{\tau_n} = 1, c_n = 1 \right) \leq \lambda.$$

*Proof.* In the case, where $\{n \mid \{\tau_n, \tau_{n+1} - 1\} \subset S_m\} = \emptyset$ the statement is trivial. Otherwise, one follows the lines of the proof of Lemma 1. □

**Lemma 4**  *Setting $\lambda = \frac{1}{e}$ and thus choosing $\ell_c = \left\lceil \frac{1}{4\Delta^2} \log c(c+1)e^2 - \frac{1}{2} \right\rceil$, the expected cumulative binary weak regret of BtWR starting with $c_1 = \tilde{c}$ in the stationary setting is bounded by*

$$\mathbb{E}\left[ R^{\bar{W}}(T) \right] \leq \frac{20K}{\Delta^2} \log(K + \tilde{c} - 1).$$

*Proof.* Analogous to the proof of Theorem 1 using Lemma 3 instead of Lemma 1. □

**Theorem 2**  *Choosing $\ell_c = \left\lceil \frac{1}{4\Delta^2} \log c(c+1)e^2 - \frac{1}{2} \right\rceil$, the expected cumulative binary weak regret of BtWR in the non-stationary setting is bounded by:*

$$\mathbb{E}\left[ R^{\bar{W}}(T) \right] \leq \frac{22KM}{\Delta^2} \log(K + T).$$

*Proof.* First, we decompose the expected regret over the stationary segments:

$$\mathbb{E}\left[ R^{\bar{W}}(T) \right] = \mathbb{E}\left[ R^{\bar{W}}(1, \nu_1) \right] + \sum_{m=2}^{M} \mathbb{E}\left[ R^{\bar{W}}(\nu_{m-1}, \nu_m - 1) \right].$$

For each $m \geq 2$ let $A_m = \{\{n \in \mathbb{N} \mid \tau_n \in S_m\} \neq \emptyset\}$ be the event that there exists a round starting in $S_m$ and $m_1 = \inf\{n \in \mathbb{N} \mid \tau_n \in S_m\}$ be the first round starting in $S_m$ given $A_m$. Let $m_0 = \sup\{n \in \mathbb{N} \mid \tau_n < \nu_{m-1}\}$ be the last round that started before $S_m$, on the event of $A_m$ we have $m_0 = m_1 - 1$, but not necessarily $\tau_{m_0} \in S_{m-1}$. The term in the sum can be further decomposed for all $m \geq 2$:

$$\begin{aligned}
&\mathbb{E}\left[ R^{\bar{W}}(\nu_{m-1}, \nu_m - 1) \right] \\
&\leq \mathbb{E}\left[ R^{\bar{W}}(\nu_{m-1}, \nu_m - 1)\mathbb{I}\{A_m\} \right] + \mathbb{E}\left[ R^{\bar{W}}(\nu_{m-1}, \nu_m - 1)\mathbb{I}\{A_m^C\} \right] \\
&= \mathbb{E}\left[ R^{\bar{W}}(\nu_{m-1}, \tau_{m_1} - 1)\mathbb{I}\{A_m\} \right] + \mathbb{E}\left[ R^{\bar{W}}(\tau_{m_1}, \nu_m - 1)\mathbb{I}\{A_m\} \right] \\
&\qquad + \mathbb{E}\left[ R^{\bar{W}}(\nu_{m-1}, \nu_m - 1)\mathbb{I}\{A_m^C\} \right] \\
&\leq 4\ell_{c_{m_0}} - 4 + \mathbb{E}\left[ R^{\bar{W}}(\tau_{m_1}, \nu_m - 1)\mathbb{I}\{A_m\} \right] \\
&\leq \frac{1}{\Delta^2} \log c_{m_0}(c_{m_0} + 1)e^2 - 2 + \frac{20K}{\Delta^2} \log(c_{m_1} + K - 1) \\
&\leq \frac{22K}{\Delta^2} \log(c_{m_0} + K),
\end{aligned}$$

where we used Lemma 4 in the third inequality because $\mathbb{E}\left[R^{\bar{W}}(\tau_{m_1}, \nu_m - 1)\mathbb{I}\{A_m\}\right]$ can be seen as the expected regret in a stationary setting with BtWR having $c_{m_0}$ as the round counter for its first round. Applying Theorem 1 for the first segment which can also be seen as an instance of the stationary setting and the obvious fact that $c_n \leq T$ for all rounds $n$, we obtain:

$$
\begin{aligned}
\mathbb{E}\left[R^{\bar{W}}(T)\right] &= \mathbb{E}\left[R^{\bar{W}}(1, \nu_1)\right] + \sum_{m=2}^{M} \mathbb{E}\left[R^{\bar{W}}(\nu_{m-1}, \nu_m - 1)\right] \\
&\leq \frac{20K\log K}{\Delta^2} + \sum_{m=2}^{M} \frac{22K}{\Delta^2}\log(c_{m_0} + K) \\
&\leq \frac{22KM}{\Delta^2}\log(K + T).
\end{aligned}
$$

$\square$

## D  MDB Regret Analysis

In the following we will utilize the McDiarmid's inequality which can be seen as a concentration inequality on a function $g$ of $n$ independent random variables.

**Lemma 11.** *(McDiarmid's inequality)*
*Let $X_1, \ldots, X_n$ be independent random variables all taking values in the set $\mathcal{X}$ and $c_1, \ldots, c_n \in \mathbb{R}$. Further, let $g : \mathcal{X}^n \mapsto \mathbb{R}$ be a function that satisfies $|g(x_1, \ldots, x_i, \ldots, x_n) - g(x_1, \ldots, x_i', \ldots, x_n)| \leq c_i$ for all $i$ and $x_1, \ldots, x_n, x_i' \in \mathcal{X}$. Then for all $\epsilon > 0$ holds*

$$
\mathbb{P}(g(x_1, \ldots, x_n) - \mathbb{E}[g(x_1, \ldots, x_n)] \geq \epsilon) \leq \exp\left(\frac{-2\epsilon^2}{\sum_{i=1}^n c_i^2}\right),
$$

$$
\mathbb{P}(g(x_1, \ldots, x_n) - \mathbb{E}[g(x_1, \ldots, x_n)] \leq -\epsilon) \leq \exp\left(\frac{-2\epsilon^2}{\sum_{i=1}^n c_i^2}\right).
$$

**Lemma 12.** *(similar to Cao et al. (2019) but errors corrected)*
*Consider a detection window with even window length $w$ that is filled with i.i.d. bits $X_1, \ldots, X_w \sim Ber(p)$, for fixed $p \in [0, 1]$. Let $A = \sum_{i=1}^{w/2} X_i$ be the sum of bits in the older window half and $B = \sum_{i=w/2+1}^{w} X_i$ the sum in the newer half. Then for any $b > 0$ holds*

$$
\mathbb{P}(|A - B| > b) \leq 2\exp\left(\frac{-2b^2}{w}\right).
$$

*Proof.* Obviously, $A$ and $B$ have expected values $\mathbb{E}[A] = \mathbb{E}[B] = \frac{wp}{2}$ and thus we derive:

$$
\begin{aligned}
\mathbb{P}(|A - B| > b) &= \mathbb{P}(A - B > b) + \mathbb{P}(B - A > b) \\
&= \mathbb{P}(A - B - \mathbb{E}[A - B] > b) + \mathbb{P}(B - A - \mathbb{E}[B - A] > b) \\
&\leq 2\exp\left(\frac{-2b^2}{w}\right),
\end{aligned}
$$

where we used McDiarmid's inequality (Lemma 11). The denominator results from the fact, that a single change of one random variable $X_i$ can cause a maximum absolute difference of 1 in $A$ or $B$. Thus, $|A - B|$ can differ no more than 1. $\square$

**Lemma 13.** *(similar to Cao et al. (2019) but errors corrected)*
*Consider a detection window with even window length $w$ that is filled with independent bits $X_1, \ldots, X_{w/2} \sim Ber(p)$ and $X_{w/2+1}, \ldots, X_w \sim Ber(p + \theta)$ with fixed $p \in [0, 1]$ and $\theta \in [-p, 1-p]$. Let $A = \sum_{i=1}^{w/2} X_i$ be the sum*

*of bits in the older window half and $B = \sum_{i=w/2+1}^{w} X_i$ the sum in the newer half. Then for any $b > 0$ such that some $c > 0$ exists with $|\theta| \geq \frac{2b}{w} + c$, holds*

$$\mathbb{P}(|A - B| > b) \geq 1 - \exp\left(\frac{-wc^2}{2}\right).$$

*Proof.* The expected values of $A$ and $B$ are now given by $\mathbb{E}[A] = \frac{wp}{2}$ and $\mathbb{E}[B] = \frac{w(p+\theta)}{2}$ and thus $\mathbb{E}[A - B] = \frac{-w\theta}{2}$ and $\mathbb{E}[B - A] = \frac{w\theta}{2}$. Consider the following case distinction for $\theta$:

If $\theta \geq 0$ then due to $|\theta| \geq \frac{2b}{w} + c$: $\mathbb{E}[B - A] \geq b + \frac{cw}{2}$, and thus:

$$
\begin{aligned}
\mathbb{P}(|A - B| > b) &= \mathbb{P}(A - B > b) + \mathbb{P}(B - A > b) \\
&\geq \mathbb{P}(B - A > b) \\
&= 1 - \mathbb{P}(B - A \leq b) \\
&= 1 - \mathbb{P}(B - A - \mathbb{E}[B - A] \leq b - \mathbb{E}[B - A]) \\
&\geq 1 - \exp\left(\frac{-2(\mathbb{E}[B - A] - b)^2}{w}\right) \\
&\geq 1 - \exp\left(\frac{-wc^2}{2}\right),
\end{aligned}
$$

where we used McDiarmid's inequality (Lemma 11) for the second inequality. The necessary condition $b - \mathbb{E}[B - A] < 0$ is implied by the existence of a $c > 0$ with $|\theta| \geq \frac{2b}{w} + c$, assumed in the beginning. We justify the denominator with the same reasoning as in Lemma 12. Otherwise, if $\theta < 0$ then due to $|\theta| \geq \frac{2b}{w} + c$: $\mathbb{E}[A - B] \geq b + \frac{cw}{2}$, and thus:

$$
\begin{aligned}
\mathbb{P}(|A - B| > b) &= \mathbb{P}(A - B > b) + \mathbb{P}(B - A > b) \\
&\geq \mathbb{P}(A - B > b) \\
&= 1 - \mathbb{P}(A - B \leq b) \\
&= 1 - \mathbb{P}(A - B - \mathbb{E}[A - B] \leq b - \mathbb{E}[A - B]) \\
&\geq 1 - \exp\left(\frac{-2(\mathbb{E}[A - B] - b)^2}{w}\right) \\
&\geq 1 - \exp\left(\frac{-wc^2}{2}\right),
\end{aligned}
$$

where we used McDiarmid's inequality (Lemma 11) for the third inequality. The necessary condition $b - \mathbb{E}[A - B] < 0$ is implied by the existence of a $c > 0$ with $|\theta| \geq \frac{2b}{w} + c$, assumed in the beginning. We justify the denominator with the same reasoning as in Lemma 12. $\square$

**Lemma 14.** *The cumulative regret w.r.t. to any regret measure $r^v$ during the m-th stationary segment can be rewritten as*

$$R^v(\nu_{m-1}, \nu_m - 1) = \sum_{i,j} N_{i,j}(\nu_{m-1}, \nu_m - 1) \cdot r^v(a_i, a_j),$$

*where $N_{i,j}(t_1, t_2) = \sum_{t=t_1}^{t_2} \mathbb{I}\{I_t = i, J_t = j\}$ is the number of times an algorithm chooses to duel the arms $a_i$ and $a_j$ within the time period $\{t_1, \ldots, t_2\}$.*

*Proof.*

$$R^v(\nu_{m-1}, \nu_m - 1) = \sum_{t=\nu_{m-1}}^{\nu_m - 1} r^v\left(a_{I_t}, a_{J_t}\right)$$

$$= \sum_{t=\nu_{m-1}}^{\nu_m - 1} \sum_{i,j} \mathbb{I}\{I_t = i, J_t = j\} \cdot r^v\left(a_i, a_j\right)$$

$$= \sum_{i,j} \left( \sum_{t=\nu_{m-1}}^{\nu_m - 1} \mathbb{I}\{I_t = i, J_t = j\} \right) \cdot r^v\left(a_i, a_j\right)$$

$$= \sum_{i,j} N_{i,j}(\nu_{m-1}, \nu_m - 1) \cdot r^v\left(a_i, a_j\right).$$

$\square$

We impose the following assumptions on the problem statement and the parameters:

**Assumption (1):** $|S_m| \geq 2L$ for all $m \in \{1, \ldots, M\}$

**Assumption (2):** $\delta \geq \frac{2b}{w} + c$ for some $c > 0$

**Assumption (3):** $\gamma \in \left[\frac{K(K-1)}{2T}, \frac{K-1}{2}\right]$ and $K \geq 3$

**Lemma** 5 *Consider a scenario with $M = 1$ and let $\tau_1$ be the first detection time. Then the expected cumulative binary strong regret of MDB is bounded by*

$$\mathbb{E}\left[R^{\bar{S}}(T)\right] \leq \mathbb{P}(\tau_1 \leq T) \cdot T + \frac{2\gamma KT}{K-1}T + \mathbb{E}\left[\tilde{R}^{\bar{S}}(T)\right].$$

*Proof.* Let $\pi(i,j)$ be the position of the pair $(a_i, a_j)$ in the ordering $\pi$ for $i < j$, starting to count at 0. In order to be well-defined for all $i, j$, let $\pi(i,j) = -1$ for all $i \geq j$ and define the event $A_{i,j,t} := \{(t - \tau - 1) \bmod \lfloor K(K-1)/2\gamma \rfloor = \pi(i,j)\}$ for all $i, j, t$. Given $\tau_1 > T$, we have $\tau = 0$ guaranteed for the whole runtime of the algorithm and the number of times a pair $(a_i, a_j)$ with $i < j$ is played can be bounded by:

$$N_{i,j}(1, T) = \sum_{t=1}^{T} \mathbb{I}\{I_t = i, J_t = j\}$$

$$\leq \sum_{t=1}^{T} \mathbb{I}\{A_{i,j,t}\} + \mathbb{I}\{\overline{A_{i,j,t}}, I_t = i, J_t = j\}$$

$$\leq \left\lceil \frac{T}{\left\lfloor \frac{K(K-1)}{2\gamma} \right\rfloor} \right\rceil + \sum_{t=1}^{T} \mathbb{I}\{\overline{A_{i,j,t}}, I_t = i, J_t = j\}$$

$$\leq \frac{2T}{\left\lfloor \frac{K(K-1)}{2\gamma} \right\rfloor} + \sum_{t=1}^{T} \mathbb{I}\{\overline{A_{i,j,t}}, I_t = i, J_t = j\}$$

$$\leq \frac{4\gamma T}{K(K-1) - (K-1)} + \sum_{t=1}^{T} \mathbb{I}\{\overline{A_{i,j,t}}, I_t = i, J_t = j\}$$

$$= \frac{4\gamma T}{(K-1)^2} + \sum_{t=1}^{T} \mathbb{I}\{\overline{A_{i,j,t}}, I_t = i, J_t = j\},$$

where make use of Assumption (3) in the third and fourth inequality. Whereas for $i \geq j$ the event $A_{i,j,t}$ will never occur such that

$$N_{i,j}(1,T) = \sum_{t=1}^{T} \mathbb{I}\{\overline{A_{i,j,t}}, I_t = i, J_t = j\}.$$

From that we derive a bound for $\mathbb{E}\left[R^{\bar{\mathrm{S}}}(T) \mid \tau_1 > T\right]$:

$$\mathbb{E}\left[R^{\bar{\mathrm{S}}}(T) \mid \tau_1 > T\right]$$

$$= \mathbb{E}\left[\sum_{i<j} N_{i,j}(1,T) \cdot \left\lceil \frac{\Delta_i^{(1)} + \Delta_j^{(1)}}{2} \right\rceil \sum_{i \geq j} N_{i,j}(1,T) \cdot \left\lceil \frac{\Delta_i^{(1)} + \Delta_j^{(1)}}{2} \right\rceil \,\middle|\, \tau_1 > T\right]$$

$$\leq \mathbb{E}\left[\sum_{i<j} \frac{4\gamma T}{(K-1)^2} \,\middle|\, \tau_1 > T\right] + \mathbb{E}\left[\sum_{i,j} \sum_{t=1}^{T} \mathbb{I}\{\overline{A_{i,j,t}}, I_t = i, J_t = j\} \cdot \left\lceil \frac{\Delta_i^{(1)} + \Delta_j^{(1)}}{2} \right\rceil \,\middle|\, \tau_1 > T\right]$$

$$\leq \frac{2\gamma KT}{K-1} + \mathbb{E}\left[\tilde{R}^{\bar{\mathrm{S}}}(T)\right],$$

where we use Lemma 14 in both equations. Thus, we obtain the desired bound:

$$\mathbb{E}\left[R^{\bar{\mathrm{S}}}(T)\right] = \mathbb{E}\left[R^{\bar{\mathrm{S}}}(T) \mid \tau_1 \leq T\right] \cdot \mathbb{P}(\tau_1 \leq T) + \mathbb{E}\left[R^{\bar{\mathrm{S}}}(T) \mid \tau_1 > T\right] \cdot \mathbb{P}(\tau_1 > T)$$

$$\leq \mathbb{P}(\tau_1 \leq T) \cdot T + \mathbb{E}\left[R^{\bar{\mathrm{S}}}(T) \mid \tau_1 > T\right]$$

$$\leq \mathbb{P}(\tau_1 \leq T) \cdot T + \frac{2\gamma KT}{K-1} + \mathbb{E}\left[\tilde{R}^{\bar{\mathrm{S}}}(T)\right].$$

$\square$

**Lemma** 6  *Consider a scenario with $M = 1$ and let $\tau_1$ be the first detection time. The probability of MDB raising a false alarm is bounded by*

$$\mathbb{P}(\tau_1 \leq T) \leq 2T \exp\left(\frac{-2b^2}{w}\right).$$

*Proof.* Let $n_{I_t,J_t,t}$ be the value of $n_{I_t,J_t}$ after its update in Line 9 at time step $t$, $A_t := \sum_{s=n_{I_t,J_t,t}-w+1}^{n_{I_t,J_t,t}-w/2} X_{I_t,J_t,s}$,

and $B_t := \sum_{s=n_{I_t,J_t,t}-w/2+1}^{n_{I_t,J_t,t}} X_{I_t,J_t,s}$. Moreover, denote by $r(t)$ the value of $r$ in time step $t$. We derive:

$$\mathbb{P}(\tau_1 \leq T) = \sum_{t\,:\,r(t)<\frac{K(K-1)}{2},t\leq T} \mathbb{P}(\tau_1 = t)$$

$$\leq \sum_{t\,:\,r(t)<\frac{K(K-1)}{2},t\leq T,n_{I_t,J_t,t}\geq w} \mathbb{P}(|A_t - B_t| > b)$$

$$\leq \sum_{t=1}^{T} 2 \exp\left(\frac{-2b^2}{w}\right)$$

$$= 2T \exp\left(\frac{-2b^2}{w}\right).$$

In the second inequality we used Lemma 12 with the observations of the duels being the bits filling the detection window. The requirement that all samples are drawn from the same Bernoulli distribution is

met since there is no changepoint and thus all samples from a pair $(a_i, a_j)$ are drawn from a Bernoulli distribution with parameter $p_{i,j}^{(1)}$. Note that $r(t), I_t$, and $J_t$ are not random variables because they are calculated deterministically. $\qquad\square$

**Lemma 7** *Consider a scenario with $M = 2$. Assume that $\nu_1 + L/2 \leq \nu_2$ and the existence of arms $a_i$ and $a_j$ with $i < j$ such that $|\delta_{i,j}^{(1)}| \geq \frac{2b}{w} + c$ for some $c > 0$. Then it holds*

$$\mathbb{P}(\tau_1 \geq \nu_1 + L/2 \mid \tau_1 \geq \nu_1) \leq \exp\left(-\frac{wc^2}{2}\right).$$

*Proof.* Assume that $\tau_1$ is large enough such that the pair $(a_i, a_j)$ is played at least $w/2$ times from $\nu_1$ onward during detection steps. In that case, let $t$ be the time step in which $(a_i, a_j)$ is played for the $w/2$-th time from $\nu_1$ onward during the detection steps. Then it holds $t < \nu_1 + L/2$. As a consequence, we obtain $\tau_1 < \nu_1 + L/2$ if we deny the assumption. Let $n_{i,j,t}$ be the value of $n_{i,j}$ after its update in Line 9 at time step $t$, $A_{i,j,t} := \sum_{s=n_{i,j,t}-w+1}^{n_{i,j,t}-w/2} X_{i,j,s}$, and $B_t := \sum_{s=n_{i,j,t}-w/2+1}^{n_{i,j,t}} X_{i,j,s}$. Since $|A_{i,j,t} - B_{i,j,t}| > b$ triggers the changepoint-detection in time step $t$, it implies $\tau_1 < \nu_1 + L/2$ given $\tau_1 \geq \nu_1$. Thus, we obtain

$$\mathbb{P}(\tau_1 < \nu_1 + L/2 \mid \tau_1 \geq \nu_1) \geq \mathbb{P}(|A_{i,j,t} - B_{i,j,t}| > b),$$

giving us the opportunity to apply Lemma 13 with $p = p_{i,j}^{(1)}$ and $\theta = \delta_{i,j}^{(1)}$ to conclude:

$$\begin{aligned}
\mathbb{P}(\tau_1 \geq \nu_1 + L/2 \mid \tau_1 \geq \nu_1) &= 1 - \mathbb{P}(\tau_1 < \nu_1 + L/2 \mid \tau_1 \geq \nu_1) \\
&\leq 1 - \mathbb{P}(|A_{i,j,t} - B_{i,j,t}| > b) \\
&\leq \exp\left(\frac{-wc^2}{2}\right).
\end{aligned}$$

$\qquad\square$

The required distance between $\nu_1$ and $\nu_2$ of at least $L/2$ time steps is implied by Assumption (1), whereas the resulting condition on $\delta_{i,j}^{(1)}$ in order to apply Lemma 13 is given by Assumption (2).

Before stating our main result in Theorem 3, we introduce notation specifically tailored to the upcoming proof. Let $\mathbb{E}_m\left[R^{\bar{S}}(t_1, t_2)\right]$ be the expected cumulative binary strong regret from time steps $t_1$ to $t_2$ inclusively of MDB started at $\nu_{m-1}$. Note that $\mathbb{E}_1\left[R^{\bar{S}}(\nu_0, T)\right] = \mathbb{E}\left[R^{\bar{S}}(T)\right]$. In order to capture false alarms and delay, we further define $\tau_m$ to be the time step of the first triggered changepoint-detection of MDB started at $\nu_{m-1}$. Thus, we can express a false alarm in the $m$-th segment raised by MDB started at $\nu_{m-1}$ as the event $\{\tau_m < \nu_m\}$. A *large* delay of $L/2$ or more time steps for $\nu_m$ in the absence of a false alarm is then given by $\{\tau_m \geq \nu_m + L/2\}$.

**Theorem 3** *Let $p, q \in [0, 1]$ with $\mathbb{P}(\tau_m < \nu_m) \leq p$ for all $m \leq M$ and $\mathbb{P}(\tau_m \geq \nu_m + L/2 \mid \tau_m \geq \nu_m) \leq q$ for all $m \leq M - 1$. Then the expected cumulative binary strong regret of MDB is bounded by*

$$\mathbb{E}\left[R^{\bar{S}}(T)\right] \leq \frac{ML}{2} + 2T\left(\frac{\gamma K}{K - 1} + p + q\right) + \sum_{m=1}^{M} \mathbb{E}\left[\tilde{R}^{\bar{S}}(\nu_{m-1}, \nu_m - 1)\right].$$

In conjunction with the previous explanation on how to formalize false alarms and large delays, $p$ represents an upper bound for the probability of MDB raising a false alarm in the $m$-th segment given MDB is started at $\nu_{m-1}$ for each $m$. Similarly, $q$ bounds the probability of MDB having a delay of at least $L/2$ for the

detection of $\nu_m$ for each $m$. Our proof is basically an adaption of its equivalent for MUCB in Cao et al. (2019) with some refinements made to quantify and lower the impact of $p$ and $q$ on the regret bound. In order to highlight these and provide the reader with a premature understanding of its concept, we first explain the structure used in Cao et al. (2019) and explain our improvements based on that. We can categorize the set of sample paths that MDB takes in *good* and *bad* ones. The good paths are the ones in which MDB raises no false alarms and has no large detection delay for any changepoint, meaning a delay of at least $L/2$. Then the bad paths are characterized by MDB raising at least one false alarm or having at least for one changepoint a large delay. We can divide each path into $M$ pieces, with each piece corresponding to a stationary segment, and utilize this concept by recursively analyzing MDB in bottom-up fashion from the last segment $S_M$ to the first $S_1$. This is done by an induction over $M$. For each segment $S_m$ we distinguish whether the path taken by MDB is at this point still good or becomes bad. In case it is good, we can apply Lemma 5 to bound the regret suffered in $S_m$ as a stationary setting and add the regret incurred in the remaining path from $S_{m+1}$ onward. Otherwise, if a false alarm occurs or the changepoint $\nu_m$ is detected with large delay, the regret can be naively bounded by $T$. In order to control the expected regret in these cases with the help of $p$ and $q$, Lemma 6 and 7 come into play, which bound the probability of a false alarm and large delay, respectively. In Cao et al. (2019) both probabilities are set ad hoc to $1/T$, reducing the additional expected regret to a constant. The bound $T$ on the regret in case of MDB entering a bad path is penalizing MDB harsh and one could ask if there is a possibility for MDB to reenter into a good path by coming across a changepoint which it detects with delay shorter than $L/2$. Based on this idea, we extend the proof in Cao et al. (2019) substantially. Indeed, we remove these "dead ends" from the previous analysis and are able to analyze the expected regret entailed in the "detours" from false alarms or delays to the reentrance into a good path.

*Proof.* First, we prove the following statement for all $m \leq M$ by induction:

$$\mathbb{E}_m\left[R^{\bar{S}}(\nu_{m-1}, T)\right] \leq (M - m + 1) \cdot \frac{L}{2} + \frac{2\gamma K}{K-1} \sum_{i=m}^{M} |S_i| + (p+q)\left(|S_m| + 2\sum_{i=m+1}^{M} |S_i|\right)$$
$$+ \sum_{i=m}^{M} \mathbb{E}\left[\tilde{R}^{\bar{S}}(\nu_{i-1}, \nu_i - 1)\right].$$

Note that $T = \sum_{m=1}^{M} |S_m|$. Then $\mathbb{E}\left[R^{\bar{S}}(T)\right]$ can be bounded by a simpler expression:

$$\mathbb{E}\left[R^{\bar{S}}(T)\right]$$
$$= \mathbb{E}_1\left[R^{\bar{S}}(\nu_0, T)\right]$$
$$\leq \frac{ML}{2} + \frac{2\gamma K}{K-1} \sum_{m=1}^{M} |S_m| + (p+q)\left(|S_1| + 2\sum_{m=2}^{M} |S_m|\right) + \sum_{m=1}^{M} \mathbb{E}\left[\tilde{R}^{\bar{S}}(\nu_{m-1}, \nu_m - 1)\right]$$
$$\leq \frac{ML}{2} + \frac{2\gamma K}{K-1} \sum_{m=1}^{M} |S_m| + 2(p+q)\sum_{m=1}^{M} |S_m| + \sum_{m=1}^{M} \mathbb{E}\left[\tilde{R}^{\bar{S}}(\nu_{m-1}, \nu_m - 1)\right]$$
$$= \frac{ML}{2} + 2T\left(\frac{\gamma K}{K-1} + p + q\right) + \sum_{m=1}^{M} \mathbb{E}\left[\tilde{R}^{\bar{S}}(\nu_{m-1}, \nu_m - 1)\right].$$

**Base case:** $m = M$
Since there is only one stationary segment from $\nu_{M-1}$ to $\nu_M - 1 = T$, we can consider this as the stationary

setting starting at $\nu_{M-1}$ and apply Lemma 5 in the second inequality:

$$\mathbb{E}_M\left[R^{\bar{S}}(\nu_{M-1}, T)\right] \leq |S_M| \cdot \mathbb{P}(\tau_M < \nu_M) + \mathbb{E}_M[R^{\bar{S}}(\nu_{M-1}, \nu_M) \mid \tau_M \geq \nu_M]$$

$$\leq |S_M|p + \frac{2\gamma K|S_M|}{K-1} + \mathbb{E}\left[\tilde{R}^{\bar{S}}(\nu_{M-1}, \nu_M - 1)\right]$$

$$\leq \frac{L}{2} + \frac{2\gamma K|S_M|}{K-1} + (p+q)|S_M| + \sum_{i=M}^{M} \mathbb{E}\left[\tilde{R}^{\bar{S}}(\nu_{i-1}, \nu_i - 1)\right].$$

**Induction hypothesis:**

For any arbitrary but fixed $m \leq M-1$, the expected cumulative binary strong regret from time steps $\nu_m$ to $T$ of MDB started at $\nu_m$ is bounded by

$$\mathbb{E}_{m+1}\left[R^{\bar{S}}(\nu_m, T)\right] \leq (M-m) \cdot \frac{L}{2} + \frac{2\gamma K}{K-1} \sum_{i=m+1}^{M} |S_i| + (p+q)\left(|S_{m+1}| + 2\sum_{i=m+2}^{M} |S_i|\right)$$

$$+ \sum_{i=m+1}^{M} \mathbb{E}\left[\tilde{R}^{\bar{S}}(\nu_{i-1}, \nu_i - 1)\right].$$

**Inductive step:** $m+1 \to m$:

Let $p_m := \mathbb{P}(\tau_m < \nu_m)$. We can decompose the regret based on the distinction whether a false alarm occurs:

$$\mathbb{E}_m\left[R^{\bar{S}}(\nu_{m-1}, T)\right]$$

$$= \mathbb{E}_m\left[R^{\bar{S}}(\nu_{m-1}, T) \mid \tau_m \geq \nu_m\right] \cdot (1 - p_m) + \mathbb{E}_m\left[R^{\bar{S}}(\nu_{m-1}, T) \mid \tau_m < \nu_m\right] \cdot p_m.$$

We can divide the expected regret in the case of no false alarm into two parts: that incurred in the $m$-th stationary segment and that from the next segment onward to the time horizon, which allows us to apply the intermediate result in Lemma 5 for the $m$-th segment:

$$\mathbb{E}_m\left[R^{\bar{S}}(\nu_{m-1}, T) \mid \tau_m \geq \nu_m\right]$$

$$= \mathbb{E}_m\left[R^{\bar{S}}(\nu_{m-1}, \nu_m - 1) \mid \tau_m \geq \nu_m\right] + \mathbb{E}_m\left[R^{\bar{S}}(\nu_m, T) \mid \tau_m \geq \nu_m\right]$$

$$\leq \frac{2\gamma K|S_m|}{K-1} + \mathbb{E}\left[\tilde{R}^{\bar{S}}(\nu_{m-1}, \nu_m - 1)\right] + \mathbb{E}_m\left[R^{\bar{S}}(\nu_m, T) \mid \tau_m \geq \nu_m\right].$$

Let $q_m := \mathbb{P}(\tau_m \geq \nu_m + L/2 \mid \tau_m \geq \nu_m)$. We split the righthand side term into:

$$\mathbb{E}_m\left[R^{\bar{S}}(\nu_m, T) \mid \tau_m \geq \nu_m\right] = \mathbb{E}_m\left[R^{\bar{S}}(\nu_m, T) \mid \nu_m \leq \tau_m < \nu_m + L/2\right] \cdot (1 - q_m)$$

$$+ \mathbb{E}_m\left[R^{\bar{S}}(\nu_m, T) \mid \tau_m \geq \nu_m + L/2\right] \cdot q_m.$$

On the event that the changepoint is detected with short delay, i.e., $\nu_m \leq \tau_m < \nu_m + L/2$, we can rewrite the expected regret as:

$$\mathbb{E}_m\left[R^{\bar{S}}(\nu_m, T) \mid \nu_m \leq \tau_m < \nu_m + L/2\right]$$

$$= \mathbb{E}_m\left[R^{\bar{S}}(\nu_m, \tau_m) \mid \nu_m \leq \tau_m < \nu_m + L/2\right] + \mathbb{E}_m\left[R^{\bar{S}}(\tau_m + 1, T) \mid \nu_m \leq \tau_m < \nu_m + L/2\right]$$

$$\leq \frac{L}{2} + \mathbb{E}_{m+1}\left[R^{\bar{S}}(\nu_m, T)\right],$$

where we used that after MDB being resetted in $\tau_m$, no detection window contains any samples from the previous segment before $\nu_m$ and that there are still at least $L$ time steps left before $\nu_{m+1}$ (given by Assumption

(1)), such that $\nu_{m+1}$ can be detected as if MDB was started at $\nu_m$. On the other hand, for longer delay we can distinguish further:

$$\mathbb{E}_m\left[R^{\bar{S}}(\nu_m, T) \mid \tau_m \geq \nu_m + L/2\right]$$
$$\leq \max\left\{\mathbb{E}_m\left[R^{\bar{S}}(\nu_m, T) \mid \nu_m + L/2 \leq \tau_m < \nu_m + L\right], \mathbb{E}_m\left[R^{\bar{S}}(\nu_m, T) \mid \tau_m \geq \nu_m + L\right]\right\}.$$

In the case of $\nu_m + L/2 \leq \tau_m < \nu_m + L$ and under Assumption (1), guaranteeing at least $L$ time steps between $\tau_m$ and $\nu_m$, we can make the same argument as above and derive:

$$\mathbb{E}_m\left[R^{\bar{S}}(\nu_m, T) \mid \nu_m + L/2 \leq \tau_m < \nu_m + L\right]$$
$$= \mathbb{E}_m\left[R^{\bar{S}}(\nu_m, \tau_m) \mid \nu_m + L/2 \leq \tau_m < \nu_m + L\right]$$
$$\quad + \mathbb{E}_m\left[R^{\bar{S}}(\tau_m + 1, T) \mid \nu_m + L/2 \leq \tau_m < \nu_m + L\right]$$
$$\leq L + \mathbb{E}_{m+1}\left[R^{\bar{S}}(\nu_m, T)\right].$$

In the case of $\tau_m \geq \nu_m + L$ the detection windows contain no samples from the previous preference matrix $P^{(m)}$ due to the fact that after $L$ time steps all detection windows are filled with new samples from $P^{(m+1)}$. Thus, the detection of $\nu_{m+1}$ applies as if MDB would have been started at $\nu_m$, but $Alg$ is still running with invalid observations from the previous segment $S_m$. Hence, it potentially suffers maximal regret for $S_{m+1}$:

$$\mathbb{E}_m\left[R^{\bar{S}}(\nu_m, T) \mid \tau_m \geq \nu_m + L\right] \leq |S_{m+1}| + \mathbb{E}_{m+1}\left[R^{\bar{S}}(\nu_m, T)\right].$$

Combining both cases on the event of $\tau_m \geq \nu_m + L/2$, we obtain:

$$\mathbb{E}_m\left[R^{\bar{S}}(\nu_m, T) \mid \tau_m \geq \nu_m + L/2\right] \leq |S_{m+1}| + \mathbb{E}_{m+1}\left[R^{\bar{S}}(\nu_m, T)\right].$$

Summarizing, on the event of $\tau_m \geq \nu_m$ we obtain:

$$\mathbb{E}_m\left[R^{\bar{S}}(\nu_m, T) \mid \tau_m \geq \nu_m\right]$$
$$= \mathbb{E}_m\left[R^{\bar{S}}(\nu_m, T) \mid \nu_m \leq \tau_m < \nu_m + L/2\right] \cdot (1 - q_m)$$
$$\quad + \mathbb{E}_m\left[R^{\bar{S}}(\nu_m, T) \mid \tau_m \geq \nu_m + L/2\right] \cdot q_m$$
$$\leq \left(\frac{L}{2} + \mathbb{E}_{m+1}\left[R^{\bar{S}}(\nu_m, T)\right]\right) \cdot (1 - q_m) + \left(|S_{m+1}| + \mathbb{E}_{m+1}\left[R^{\bar{S}}(\nu_m, T)\right]\right) \cdot q_m$$
$$= \frac{L}{2} \cdot (1 - q_m) + |S_{m+1}| \cdot q_m + \mathbb{E}_{m+1}\left[R^{\bar{S}}(\nu_m, T)\right]$$
$$\leq \frac{L}{2} + |S_{m+1}| \cdot q_m + \mathbb{E}_{m+1}\left[R^{\bar{S}}(\nu_m, T)\right].$$

Coming back to the case of $\tau_m < \nu_m$, we can make use of the same arguments as for the intermediate results of the previous cases with a case distinction depending on whether $\tau_{m+1} < \nu_m + L/2$:

$$\mathbb{E}_m\left[R^{\bar{S}}(\nu_{m-1}, T) \mid \tau_m < \nu_m\right]$$
$$= \mathbb{E}_m\left[R^{\bar{S}}(\nu_{m-1}, \nu_m - 1) \mid \tau_m < \nu_m\right] + \mathbb{E}_m\left[R^{\bar{S}}(\nu_m, T) \mid \tau_m < \nu_m\right]$$
$$\leq |S_m| + \max\left\{\mathbb{E}_m\left[R^{\bar{S}}(\nu_m, T) \mid \tau_{m+1} < \nu_m + L/2, \tau_m < \nu_m\right],\right.$$
$$\left.\mathbb{E}_m\left[R^{\bar{S}}(\nu_m, T) \mid \tau_{m+1} \geq \nu_m + L/2, \tau_m < \nu_m\right]\right\}$$
$$\leq |S_m| + \max\left\{\frac{L}{2} + \mathbb{E}_{m+1}\left[R^{\bar{S}}(\nu_m, T)\right], |S_{m+1}| + \mathbb{E}_{m+1}\left[R^{\bar{S}}(\nu_m, T)\right]\right\}$$
$$= |S_m| + |S_{m+1}| + \mathbb{E}_{m+1}\left[R^{\bar{S}}(\nu_m, T)\right].$$

Finally, we can combine the bounds on the expected regret in both major cases $\tau_m \geq \nu_m$ and $\tau_m < \nu_m$, and apply the induction hypothesis in the third inequality in order to derive:

$$
\begin{aligned}
&\mathbb{E}_m \left[ R^{\bar{S}}(\nu_{m-1}, T) \right] \\
&= \mathbb{E}_m \left[ R^{\bar{S}}(\nu_{m-1}, T) \mid \tau_m \geq \nu_m \right] \cdot (1 - p_m) + \mathbb{E}_m \left[ R^{\bar{S}}(\nu_{m-1}, T) \mid \tau_m < \nu_m \right] \cdot p_m \\
&\leq \left( \frac{L}{2} + \frac{2\gamma K |S_m|}{K-1} + |S_{m+1}| \cdot q_m + \mathbb{E} \left[ \tilde{R}^{\bar{S}}(\nu_{m-1}, \nu_m - 1) \right] + \mathbb{E}_{m+1} \left[ R^{\bar{S}}(\nu_m, T) \right] \right) \cdot (1 - p_m) \\
&\quad + \left( |S_m| + |S_{m+1}| + \mathbb{E}_{m+1} \left[ R^{\bar{S}}(\nu_m, T) \right] \right) \cdot p_m \\
&= \left( \frac{L}{2} + \frac{2\gamma K |S_m|}{K-1} + \mathbb{E} \left[ \tilde{R}^{\bar{S}}(\nu_{m-1}, \nu_m - 1) \right] \right) \cdot (1 - p_m) + |S_{m+1}| \cdot (1 - p_m) q_m \\
&\quad + (|S_{m+1}| + |S_m|) \cdot p_m + \mathbb{E}_{m+1} \left[ R^{\bar{S}}(\nu_m, T) \right] \\
&\leq \frac{L}{2} + \frac{2\gamma K |S_m|}{K-1} + (|S_{m+1}| + |S_m|) \cdot (p + q) + \mathbb{E} \left[ \tilde{R}^{\bar{S}}(\nu_{m-1}, \nu_m - 1) \right] + \mathbb{E}_{m+1} \left[ R^{\bar{S}}(\nu_m, T) \right] \\
&\leq \frac{L}{2} + \frac{2\gamma K |S_m|}{K-1} + (|S_{m+1}| + |S_m|) \cdot (p + q) + \mathbb{E} \left[ \tilde{R}^{\bar{S}}(\nu_{m-1}, \nu_m - 1) \right] + (M - m) \cdot \frac{L}{2} \\
&\quad + \frac{2\gamma K}{K-1} \sum_{i=m+1}^{M} |S_i| + (p + q) \left( |S_{m+1}| + 2 \sum_{i=m+2}^{M} |S_i| \right) + \sum_{i=m+1}^{M} \mathbb{E} \left[ \tilde{R}^{\bar{S}}(\nu_{i-1}, \nu_i - 1) \right] \\
&= (M - m + 1) \cdot \frac{L}{2} + \frac{2\gamma K}{K-1} \sum_{i=m}^{M} |S_i| + (p + q) \left( |S_m| + 2 \sum_{i=m+1}^{M} |S_i| \right) \\
&\quad + \sum_{i=m}^{M} \mathbb{E} \left[ \tilde{R}^{\bar{S}}(\nu_{i-1}, \nu_i - 1) \right],
\end{aligned}
$$

which proves the hypothesis. For the second inequality we used:

$$
\begin{aligned}
&|S_{m+1}| \cdot (1 - p_m) q_m + (|S_{m+1}| + |S_m|) \cdot p_m \\
&\leq (|S_{m+1}| + |S_m|) \cdot ((1 - p_m) q_m + p_m) \\
&\leq (|S_{m+1}| + |S_m|) \cdot (p_m + q_m) \\
&\leq (|S_{m+1}| + |S_m|) \cdot (p + q),
\end{aligned}
$$

since $p \geq \max_m p_m$ and $q \geq \max_m q_m$. $\qquad\square$

Theorem 3 bounds MDB's expected regret depending on $p$ and $q$, but these are not explicitly given by the problem statement nor the choice of our parameters. A bound solely based on the given parameters is more desirable. We catch up on this by plugging in the bounds obtained in Lemma 6 and 7 directly.

**Corollary 3.** *For any choice of parameters satisfying Assumptions (1) to (3) the expected cumulative binary strong regret of MDB is bounded by*

$$
\mathbb{E} \left[ R^{\bar{S}}(T) \right] \leq \frac{ML}{2} + 2T \left( \frac{\gamma K}{K-1} + p + q \right) + \sum_{m=1}^{M} \mathbb{E} \left[ \tilde{R}^{\bar{S}}(\nu_{m-1}, \nu_m - 1) \right],
$$

*with $p = 2T \exp \left( \frac{-2b^2}{w} \right)$ and $q = \exp \left( -\frac{wc^2}{2} \right)$, while $c > 0$ stems from Assumption (2).*

Being equipped with a bound depending on the chosen parameters, we are interested in an optimal parameterization in the sense that it minimizes MDB's expected regret. We start by considering $\gamma$ in isolation.

**Corollary 4.** *Let $p, q \in [0, 1]$ with $\mathbb{P}(\tau_m < \nu_m) \leq p$ for all $m \leq M$ and $\mathbb{P}(\tau_m \geq \nu_m + L/2 \mid \tau_m \geq \nu_m) \leq q$ for all $m - 1 \leq M$. Setting $\gamma = \sqrt{\frac{Mw}{8T}} \cdot (K - 1)$, the expected cumulative binary strong regret of MDB is*

*bounded by*

$$\mathbb{E}\left[R^{\bar{S}}(T)\right] \leq \sqrt{2wMT}K + 2(p+q)T + \sum_{m=1}^{M} \mathbb{E}\left[\tilde{R}^{\bar{S}}(\nu_m, \nu_{m+1} - 1)\right].$$

*Proof.* In order to minimize the bound in Theorem 3 depending on $\gamma$ we only have to consider the global minimum of

$$\frac{Mw\left\lfloor \frac{K(K-1)}{2\gamma} \right\rfloor}{2} + \frac{2\gamma KT}{(K-1)}$$

for $\gamma \in (0, 1]$, where we inserted the definition of $L$ given earlier. To circumvent rounding problems caused by the floor, we instead consider the following function bounding that term:

$$g(\gamma) := \frac{MwK(K-1)}{4\gamma} + \frac{2\gamma KT}{K-1}.$$

The first derivative is

$$g'(\gamma) = -\frac{MwK(K-1)}{4\gamma^2} + \frac{2KT}{K-1}.$$

Setting $g'(\gamma^*) = 0$ in search of a local minimum, we obtain

$$\gamma^* = \sqrt{\frac{Mw}{8T}} \cdot (K-1).$$

This is indeed a local minimum because it holds $g''(\gamma^*) > 0$ for the second derivative given by

$$g''(\gamma) = \frac{MwK(K-1)}{2\gamma^3}.$$

It is also a global minimum because $g$ and its domain are convex. Plugging $\gamma^*$ into $g$, we obtain the following bound:

$$\frac{ML}{2} + \frac{2\gamma^* KT}{K-1} \leq \sqrt{2wMT}K.$$

$\square$

For certain values of $M$, $T$, and $K$ (and also $w$ which is the only parameter of our choice) the derived formula for $\gamma$ violates the condition $\gamma \leq 1$. MDB could still be executed, but it would have maximal possible strong regret, same as for $\gamma = 1$, since detection steps are conducted perpetually without a break and thus only pairs containing two different arms are played. At last, we derive the optimal choices for the missing parameters $w$ and $b$.

**Corollary** 1 *Setting* $\gamma = \sqrt{\frac{Mw}{8T}} \cdot (K-1)$, $b = \sqrt{\frac{w}{2}\log\left(\frac{\sqrt{2T}(2T+1)\delta}{\sqrt{M}K}\right)}$, $c = \sqrt{\frac{2}{w}\log\left(\frac{\sqrt{2T}(2T+1)\delta}{\sqrt{M}K}\right)}$, $w$ *to the lowest even integer greater or equal* $\frac{8}{\delta^2}\log\left(\frac{\sqrt{2T}(2T+1)\delta}{\sqrt{M}K}\right)$, *the expected cumulative binary strong regret of MDB is bounded by*

$$\mathbb{E}\left[R^{\bar{S}}(T)\right] \leq \left(\sqrt{8\log\left(\frac{\sqrt{2T}(2T+1)\delta}{\sqrt{M}K}\right)} + 1\right) \cdot \frac{\sqrt{2MT}K}{\delta} + \sum_{m=1}^{M} \mathbb{E}\left[\tilde{R}^{\bar{S}}(\nu_m, \nu_{m+1} - 1)\right].$$

*Proof.* Considering the third term in Corollary Corollary 4 to guarantee the bound $2(p+q)T \leq C$ for $C \in \mathbb{R}^+$ of our choice, we need to have

$$p \leq \frac{aC}{2T} \text{ and } q \leq \frac{(1-a)C}{2T}$$

for some $a \in [0, 1]$. The intention is to choose $C$ as high as possible without changing the asymptotic behavior of the bound given in Corollary 4. With greater $C$ there is more space for the probabilities $p$ and $q$ to increase such that the window length $w$ can be chosen smaller. We fulfill the demands towards $p$ and $q$ posed in Theorem 3 by ensuring that the above mentioned bounds are greater than those given by Lemma 6 and 7, and therefore obtain

$$2T \exp\left(\frac{-2b^2}{w}\right) \leq \frac{aC}{2T} \text{ and } \exp\left(-\frac{wc^2}{2}\right) \leq \frac{(1-a)C}{2T}.$$

These inequalities are equivalent to

$$b \geq \sqrt{\frac{w}{2} \log\left(\frac{4T^2}{aC}\right)} \text{ and } c \geq \sqrt{\frac{2}{w} \log\left(\frac{2T}{(1-a)C}\right)}.$$

In order to simplify further calculations we set $\frac{4T^2}{aC} = \frac{2T}{(1-a)C}$, from which we derive $a = \frac{2T}{2T+1}$. Next, we choose $C = \frac{\sqrt{2MTK}}{\delta}$, which gives us the intermediate result:

$$\mathbb{E}\left[R^{\bar{S}}(T)\right] \leq \left(\sqrt{w} + \frac{1}{\delta}\right) \cdot \sqrt{2MTK} + \sum_{m=1}^{M} \mathbb{E}\left[\tilde{R}^{\bar{S}}(\nu_m, \nu_{m+1} - 1)\right].$$

We set

$$b = \sqrt{\frac{w}{2} \log\left(\frac{\sqrt{2T}(2T+1)\delta}{\sqrt{M}K}\right)} \text{ and } c = \sqrt{\frac{2}{w} \log\left(\frac{\sqrt{2T}(2T+1)\delta}{\sqrt{M}K}\right)},$$

such that the lower bounds stated above are met. Then we plug this into Assumption (2) in order to obtain for $w$:

$$\delta \geq \frac{2b}{w} + c \Leftrightarrow w \geq \frac{8}{\delta^2} \log\left(\frac{\sqrt{2T}(2T+1)\delta}{\sqrt{M}K}\right).$$

Finally, we set $w$ to the lowest even integer above its lower bound. $\qquad \square$

# E  DETECT Regret Analysis

Again, the number of time steps $L$ needed to fill all windows completely during a detection phase, i.e., each pair of arms $(a_I, a_j)$ with $a_I \neq a_j$ is played at least $w$ times, remains a key quantity used in the regret analysis. Since it differs from MDB, we define:

$$L' := w(K - 1).$$

Another new parameter we have to take into consideration is the adaptation of $\delta$ to entries in the preference matrices that relate to winning probabilities of the Condorcet winner in each segment. The definition of $\delta$ in Section 4 is not appropriate anymore because DETECT cannot detect changes at any other entries than those being related to the by *Alg* suspected Condorcet winner. Hence, we define

$$\delta_*^{(m)} := \max_j |\delta_{m^*,j}^{(m)}| \quad \text{and} \quad \delta_* := \min_m \delta_*^{(m)},$$

where $\delta_{i,j}^{(m)} := p_{i,j}^{(m+1)} - p_{i,j}^{(m)}$ denotes the magnitude of change of the preference probability for the pair of arms $(a_i, a_j)$ between segment $S_m$ and $S_{m+1}$. Continuing, we impose the following assumptions on the problem statement and the parameters:

**Assumption (1):** $|S_m| \geq \tilde{T} + \frac{3}{2}L'$ for all $m \in \{1, \ldots, M\}$

**Assumption (2):** $\delta_* \geq \frac{2b}{w} + c$ for some $c > 0$

Assumption (1) requires a minimal length for all stationary segments depending on $\tilde{T}$ to guarantee that DETECT is able to detect changepoints as touched on above. Assumption (2) is required to allow short delay with certain probability, analogously to Assumption (2) for MDB. It also implies $\delta_*^{(m)} > 0$ for each $m$, meaning that for every changepoint $\nu_m$ there is at least one entry $p_{m^*,j}^{(m+1)}$ in $P^{(m+1)}$ different to the winning probability $p_{m^*,j}^{(m)}$ in the previous segment. Otherwise DETECT would automatically fail to detect these kind of changepoints.

We adopt the notation used in Appendix D, but assume $r^v$ to be the binary weak regret aggregation function (instead of binary strong regret) in the course of the following theoretical analysis.

**Lemma 8** *Consider a scenario with $M = 1$. Let $\tau_1$ be the first detection time and $a_I$ be the first suspected Condorcet winner returned by Alg. Given $\tilde{T} \le T$, the expected cumulative binary weak regret of DETECT is bounded by*

$$\mathbb{E}\left[R^{\bar{W}}(T)\right] \le \mathbb{E}\left[\tilde{R}^{\bar{W}}(\tilde{T})\right] + (T - \tilde{T}) \cdot (1 - \mathbb{P}(\tau_1 > T, a_I = a_{1^*})).$$

*Proof.* We can split the expected regret by distinguishing whether a false alarm is raised or the optimal arm has not been found by *Alg* as follows:

$$
\begin{aligned}
\mathbb{E}\left[R^{\bar{W}}(T)\right] &= \mathbb{E}\left[R^{\bar{W}}(\tilde{T})\right] + \mathbb{E}\left[R^{\bar{W}}(\tilde{T}+1, T)\right] \\
&= \mathbb{E}\left[\tilde{R}^{\bar{W}}(\tilde{T})\right] + \mathbb{E}\left[R^{\bar{W}}(\tilde{T}+1, T) \mid \tau_1 \le T\right] \cdot \mathbb{P}(\tau_1 \le T) \\
&\quad + \mathbb{E}\left[R^{\bar{W}}(\tilde{T}+1, T) \mid \tau_1 > T, a_I = a_{1^*}\right] \cdot \mathbb{P}\left(\tau_1 > T, a_I = a_{1^*}\right) \\
&\quad + \mathbb{E}\left[R^{\bar{W}}(\tilde{T}+1, T) \mid \tau_1 > T, a_I \ne a_{1^*}\right] \cdot \mathbb{P}\left(\tau_1 > T, a_I \ne a_{1^*}\right) \\
&\le \mathbb{E}\left[\tilde{R}^{\bar{W}}(\tilde{T})\right] + (T - \tilde{T}) \cdot \mathbb{P}(\tau_1 \le T) + (T - \tilde{T}) \cdot \mathbb{P}\left(\tau_1 > T, a_I \ne a_{1^*}\right) \\
&= \mathbb{E}\left[\tilde{R}^{\bar{W}}(\tilde{T})\right] + (T - \tilde{T}) \cdot (1 - \mathbb{P}\left(\tau_1 > T, a_I = a_{1^*}\right)),
\end{aligned}
$$

where we rely on $\tilde{T} \le T$ for the first equation and utilized for the first inequality the fact that in the case of $\tau_1 > T$ and $a_I = a_{1^*}$ no regret is incurred in the detection phase due to properties of the binary weak regret. $\square$

The condition of $\tilde{T} \le T$ is implied by Assumption (1) since we have $|S_1| = T$ due to the absence of changepoints. We can bound the probability of a false alarm given that *Alg* has identified the optimal arm successfully by using Lemma 12.

**Lemma 9** *Consider a scenario with $M = 1$. Let $\tau_1$ be the first detection time and $a_I$ be the first suspected Condorcet winner returned by Alg. The probability of DETECT raising a false alarm given that $a_I = a_{1^*}$ is bounded by*

$$\mathbb{P}\left(\tau_1 \le T \mid a_I = a_{1^*}\right) \le \frac{2(T - \tilde{T})}{\mathbb{P}\left(a_I = a_{1^*}\right)} \cdot \exp\left(\frac{-2b^2}{w}\right).$$

*Proof.* Let $n_{I,J_t,t}$ be the value of $n_{I,J_t}$ after its update in Line 13 at time step $t$, $A_{I,t} := \sum_{s=n_{I,J_t,t}-w+1}^{n_{I,J_t,t}-w/2} X_{I,J_t,s}$, and $B_{I,t} := \sum_{s=n_{I,J_t,t}-w/2+1}^{n_{I,J_t,t}} X_{I,J_t,s}$. Let $r(t)$ be the value of $r$ in time step $t$ during a detection step. We

derive:

$$
\mathbb{P}\left(\tau_1 \leq T \mid a_I = a_{1^*}\right) = \sum_{t=\tilde{T}+1}^{T} \mathbb{P}\left(\tau_1 = t \mid a_I = a_{1^*}\right)
$$

$$
\leq \sum_{t=\tilde{T}+1 \, : \, n_{I,J_t,t} \geq w}^{T} \mathbb{P}\left(|A_{I,t} - B_{I,t}| > b \mid a_I = a_{1^*}\right)
$$

$$
\leq \sum_{t=\tilde{T}+1 \, : \, n_{I,J_t,t} \geq w}^{T} \frac{\mathbb{P}(|A_{1^*,t} - B_{1^*,t}| > b)}{\mathbb{P}\left(a_I = a_{1^*}\right)}
$$

$$
\leq \frac{1}{\mathbb{P}\left(a_I = a_{1^*}\right)} \sum_{t=\tilde{T}+1}^{T} 2\exp\left(\frac{-2b^2}{w}\right)
$$

$$
= \frac{2(T - \tilde{T})}{\mathbb{P}\left(a_I = a_{1^*}\right)} \cdot \exp\left(\frac{-2b^2}{w}\right).
$$

In the third inequality we used Lemma 12 with the observations of the duels between the pair $(a_I, a_{J_t})$ being the bits filling the detection window $D_{I,J_t}$. The requirement that all samples are drawn from the same Bernoulli distribution is met since there is no changepoint and thus all samples from a pair $(a_i, a_j)$ are drawn from a Bernoulli distribution with parameter $p_{i,j}^{(1)}$. Note that $J_t$ is deterministic in each time step $t$ later than $\tilde{T}$. $\qquad\square$

Next, we bound the probability of a delay of at least $L/2$ time steps given that the suspected Condorcet winner returned by *Alg* is indeed the true optimal arm.

**Lemma** 10 *Consider a scenario with $M = 2$. Let $\tau$ be the first detection time and $a_I$ the first suspected Condorcet winner returned by Alg. Assume that $\nu_1 + L'/2 \leq \nu_2$ and the existence of an arm $a_j$ such that $\delta_*^{(1)} \geq \frac{2b}{w} + c$ for some $c > 0$. Then it holds*

$$
\mathbb{P}\left(\tau_1 \geq \nu_1 + L'/2 \mid \tau_1 \geq \nu_1, a_I = a_{1^*}\right) \leq \exp\left(-\frac{wc^2}{2}\right).
$$

*Proof.* Assume that $\tau_1$ is large enough such that the pair $(a_I, a_j)$ is played at least $w/2$ times from $\nu_1$ onward during the detection phase. In that case, let $t$ be the time step in which the pair $(a_{1^*}, a_j)$ is played for the $w/2$-th time from $\nu_1$ onward during the first detection phase. Then it holds $t < \nu_1 + L'/2$. As a consequence, we obtain $\tau_1 < \nu_1 + L'/2$ if we deny the assumption. Let $n_{I,j,t}$ be the value of $n_{I,j}$ after its update at time step $t$, $A_{i,j,t} := \sum_{s=n_{i,j,t}-w+1}^{n_{i,j,t}-w/2} X_{i,j,s}$, and $B_t := \sum_{s=n_{i,j,t}-w/2+1}^{n_{i,j,t}} X_{i,j,s}$. Since $|A_{I,j,t} - B_{I,j,t}| > b$ triggers the changepoint-detection in time step $t$, it implies $\tau_1 < \nu_1 + L'/2$ given $\tau_1 \geq \nu_1$. Thus, we obtain

$$
\mathbb{P}\left(\tau_1 < \nu_1 + L'/2 \mid \tau_1 \geq \nu_1, a_I = a_{1^*}\right) \geq \mathbb{P}\left(|A_{I,j,t} - B_{I,j,t}| > b \mid a_I = a_{1^*}\right).
$$

We can apply Lemma 13 with $p = p_{1^*,j}^{(1)}$ and $\theta = \delta_{1^*,j}^{(1)}$ and conclude:

$$
\mathbb{P}\left(\tau_1 \geq \nu_1 + L'/2 \mid \tau_1 \geq \nu_1, a_I = a_{1^*}\right) = 1 - \mathbb{P}\left(\tau_1 < \nu_1 + L'/2 \mid \tau_1 \geq \nu_1, a_I = a_{1^*}\right)
$$

$$
\leq 1 - \mathbb{P}\left(|A_{I,j,t} - B_{I,j,t}| > b \mid a_I = a_{1^*}\right)
$$

$$
\leq 1 - \frac{\mathbb{P}(|A_{1^*,j,t} - B_{1^*,j,t}| > b)}{\mathbb{P}\left(a_I = a_{1^*}\right)}
$$

$$
\leq \exp\left(-\frac{wc^2}{2}\right).
$$

$\qquad\square$

The required distance between $\nu_1$ and $\nu_2$ of at least $L'/2$ time steps is implied by Assumption (1), whereas the condition on $\delta_*^{(1)}$ is given by Assumption (2).

Before stating our main result in Theorem 4, we adopt again the notation used in Appendix D for using $\mathbb{E}_m\left[R(t_1, t_2)\right]$ with $f$ being the binary weak regret aggregation function. Furthermore, let $a_{I_m}$ be the first suspected Condorcet winner returned by $Alg$ when DETECT is started at $\nu_{m-1}$. Since DETECT's success of detecting changepoints is highly depending on $Alg$'s ability to find the true optimal arm after $\tilde{T}$ time steps and our analysis capitalizes on that, we define $p_{\tilde{T}}^{(m)}$ as the probability that the suspected Condorcet winner $\tilde{a}_{\tilde{T}}$ returned by $Alg$ after $\tilde{T}$ time steps is the optimal arm, i.e., $p_{\tilde{T}}^{(m)} := \mathbb{P}\left(\tilde{a}_{\tilde{T}} = a_{m^*}\right)$, given it is run in a stationary setting with preference matrix $P^{(m)}$. Based on that, let $p_{\tilde{T}} \leq \min_m p_{\tilde{T}}^{(m)}$, stating a lower bound valid for all $M$ preference matrices. Since Assumption (1) ensures that the first running phase ends before $\nu_m$ given that DETECT is started at $\nu_{m-1}$, we can later relate $p_{\tilde{T}}$ to the results in Lemma 21 and 23 if we are given WS or BtW as $Alg$, respectively.

**Theorem** 4 *Let $p, q \in [0,1]$ with $\mathbb{P}\left(\tau_m < \nu_m \mid a_{I_m} = a_{m^*}\right) \leq p$ for all $m \leq M$ and $\mathbb{P}\left(\tau_m \geq \nu_m + L'/2 \mid \tau_m \geq \nu_m, a_{I_m} = a_{m^*}\right) \leq q$ for all $m \leq M-1$. Then the expected cumulative binary weak regret of DETECT is bounded by*

$$\mathbb{E}\left[R^{\bar{W}}(T)\right] \leq \frac{ML'}{2} + (1 - p_{\tilde{T}} + pp_{\tilde{T}} + q)MT + \sum_{m=1}^{M} \mathbb{E}\left[\tilde{R}^{\bar{W}}(\nu_{m-1}, \nu_{m-1} + \tilde{T} - 1)\right].$$

The proof is an induction over the number of segments $M$, same as its adequate for MDB in Appendix D, and thus also inspired by the one presented for MUCB in Cao et al. (2019).

In the case of an false alarm, DETECT could recover completely if the next changepoint is at least $\tilde{T} + L'$ time steps away, such that the running phase is finished and all detection windows are filled and prepared to encounter the next changepoint. Since we cannot simply assume this to happen, we have to deal with the opposite event in which either the detection windows are not filled to a sufficient extent, or even worse, the running phase of $Alg$ is not finished yet, leaving it in a corrupted state to enter the next segment, thus invalidating the lower bound $p_{\tilde{T}}$.

*Proof.* First, we prove the following statement for all $m \leq M$ by induction:

$$\mathbb{E}_m\left[R^{\bar{W}}(\nu_{m-1}, T)\right]$$

$$\leq (M-m) \cdot \frac{L'}{2} + q \sum_{i=m+1}^{M} (i-m)|S_i| + \sum_{i=m}^{M} \mathbb{E}\left[\tilde{R}^{\bar{W}}(\nu_{i-1}, \nu_{i-1} + \tilde{T} - 1)\right]$$

$$+ (1 - (1-p)p_{\tilde{T}}) \sum_{i=m}^{M} (i-m+1)|S_i|.$$

Note that $T = \sum_{m=1}^{M} |S_m|$. Then $\mathbb{E}\left[R^{\bar{W}}(T)\right]$ can be bounded by a simpler expression:

$$\mathbb{E}\left[R^{\bar{W}}(T)\right]$$

$$= \mathbb{E}_1\left[R^{\bar{W}}(\nu_0, T)\right]$$

$$\leq \frac{ML'}{2} + q \sum_{m=2}^{M} (m-1)|S_m| + (1 - (1-p)p_{\tilde{T}}) \sum_{m=1}^{M} m|S_m|$$

$$+ \sum_{m=1}^{M} \mathbb{E}\left[\tilde{R}^{\bar{W}}(\nu_{m-1}, \nu_{m-1} + \tilde{T} - 1)\right]$$

$$= \frac{ML'}{2} + q\left((M-1)T - \sum_{m=1}^{M-1}\sum_{i=1}^{m}|S_i|\right) + (1 - p_{\tilde{T}} + pp_{\tilde{T}})\left(MT - \sum_{m=1}^{M}\sum_{i=1}^{m-1}|S_i|\right)$$

$$+ \sum_{m=1}^{M} \mathbb{E}\left[\tilde{R}^{\bar{W}}(\nu_{m-1}, \nu_{m-1} + \tilde{T} - 1)\right]$$

$$\leq \frac{ML'}{2} + q(M-1)T + (1 - p_{\tilde{T}} + pp_{\tilde{T}})MT + \sum_{m=1}^{M} \mathbb{E}\left[\tilde{R}^{\bar{W}}(\nu_{m-1}, \nu_{m-1} + \tilde{T} - 1)\right]$$

$$\leq \frac{ML'}{2} + (1 - p_{\tilde{T}} + pp_{\tilde{T}} + q)MT + \sum_{m=1}^{M} \mathbb{E}\left[\tilde{R}^{\bar{W}}(\nu_{m-1}, \nu_{m-1} + \tilde{T} - 1)\right].$$

**Base case:** $m = M$

Since there is only one stationary segment from $\nu_{M-1}$ to $\nu_M - 1 = T$, we can consider this as the stationary setting starting at $\nu_{M-1}$ and apply Lemma 8 in the first inequality:

$$\mathbb{E}_M\left[R^{\bar{W}}(\nu_{M-1}, T)\right]$$

$$\leq \mathbb{E}\left[\tilde{R}^{\bar{W}}(\nu_{M-1}, \nu_{M-1} + \tilde{T} - 1)\right] + \left(|S_M| - \tilde{T}\right) \cdot \left(1 - \mathbb{P}\left(\tau_M \geq \nu_M, a_{I_M} = a_{M^*}\right)\right)$$

$$= \mathbb{E}\left[\tilde{R}^{\bar{W}}(\nu_{M-1}, \nu_{M-1} + \tilde{T} - 1)\right]$$

$$\quad + \left(|S_M| - \tilde{T}\right) \cdot \left(1 - \left(1 - \mathbb{P}\left(\tau_M < \nu_M \mid a_{I_M} = a_{M^*}\right)\right) \cdot \mathbb{P}\left(a_{I_M} = a_{M^*}\right)\right)$$

$$\leq \mathbb{E}\left[\tilde{R}^{\bar{W}}(\nu_{M-1}, \nu_{M-1} + \tilde{T} - 1)\right] + (1 - (1-p)p_{\tilde{T}})|S_M|$$

$$= (M - M) \cdot \frac{L'}{2} + q\sum_{i=M+1}^{M}(i - M)|S_i| + \sum_{i=M}^{M}\mathbb{E}\left[\tilde{R}^{\bar{W}}(\nu_{i-1}, \nu_{i-1} + \tilde{T} - 1)\right]$$

$$\quad + (1 - (1-p)p_{\tilde{T}})\sum_{i=M}^{M}(i - M + 1)|S_i|.$$

**Induction hypothesis**:

For any arbitrary but fixed $m \leq M - 1$, the expected cumulative binary weak regret from time steps $\nu_m$ to $T$ of DETECT started at $\nu_m$ is bounded by

$$\mathbb{E}_{m+1}\left[R^{\bar{W}}(\nu_{m-1}, T)\right]$$

$$\leq (M - m - 1) \cdot \frac{L'}{2} + q\sum_{i=m+2}^{M}(i - m - 1)|S_i| + \sum_{i=m+1}^{M}\mathbb{E}\left[\tilde{R}^{\bar{W}}(\nu_{i-1}, \nu_{i-1} + \tilde{T} - 1)\right]$$

$$\quad + (1 - (1-p)p_{\tilde{T}})\sum_{i=m+1}^{M}(i - m)|S_i|.$$

**Inductive step:** $m + 1 \rightarrow m$

Let $p_m = \mathbb{P}\left(\tau_m < \nu_m \mid a_{I_m} = a_{m^*}\right)$. We can decompose the regret based on the distinction whether *Alg* returned the true optimal arm, i.e., $a_{I_m} = a_{m^*}$ and then whether a false alarm, i.e., $\tau_m < \nu_m$, occurs:

$$\mathbb{E}_m\left[R^{\bar{W}}(\nu_{m-1}, T)\right]$$

$$= \mathbb{E}_m\left[R^{\bar{W}}(\nu_{m-1}, \nu_{m-1} + \tilde{T} - 1)\right] + \mathbb{E}_m\left[R^{\bar{W}}(\nu_{m-1} + \tilde{T}, T)\right]$$

$$= \mathbb{E}\left[\tilde{R}^{\bar{W}}(\nu_{m-1}, \nu_{m-1} + \tilde{T} - 1)\right] + \mathbb{E}_m\left[R^{\bar{W}}(\nu_{m-1} + \tilde{T}, T) \mid a_{I_m} \neq a_{m^*}\right] \cdot \left(1 - p_{\tilde{T}}^{(m)}\right)$$

$$+ \mathbb{E}_m \left[ R^{\bar{W}}(\nu_{m-1} + \tilde{T}, T) \mid \tau_m < \nu_m, a_{I_m} = a_{m^*} \right] \cdot p_m \cdot p_{\tilde{T}}^{(m)}$$

$$+ \mathbb{E}_m \left[ R^{\bar{W}}(\nu_{m-1} + \tilde{T}, T) \mid \tau_m \geq \nu_m, a_{I_m} = a_{m^*} \right] \cdot (1 - p_m) p_{\tilde{T}}^{(m)}$$

$$\leq \mathbb{E} \left[ \tilde{R}^{\bar{W}}(\nu_{m-1}, \nu_{m-1} + \tilde{T} - 1) \right] + \left( 1 - (1 - p_m) p_{\tilde{T}}^{(m)} \right) \sum_{i=m}^{M} |S_i|$$

$$+ \mathbb{E}_m \left[ R^{\bar{W}}(\nu_{m-1} + \tilde{T}, T) \mid \tau_m \geq \nu_m, a_{I_m} = a_{m^*} \right].$$

We can reduce the expected regret from the first detection phase onward in the case of no false alarm and correctly returned Condorcet winner to the regret incurred from the next segment onward to the time horizon:

$$\mathbb{E}_m \left[ R^{\bar{W}}(\nu_{m-1} + \tilde{T}, T) \mid \tau_m \geq \nu_m, a_{I_m} = a_{m^*} \right]$$

$$= \mathbb{E}_m \left[ R^{\bar{W}}(\nu_{m-1} + \tilde{T}, \nu_m - 1) \mid \tau_m \geq \nu_m, a_{I_m} = a_{m^*} \right]$$

$$+ \mathbb{E}_m \left[ R^{\bar{W}}(\nu_m, T) \mid \tau_m \geq \nu_m, a_{I_m} = a_{m^*} \right]$$

$$= \mathbb{E}_m \left[ R^{\bar{W}}(\nu_m, T) \mid \tau_m \geq \nu_m, a_{I_m} = a_{m^*} \right].$$

This is justified by the fact that in every time step of the first detection phase the true Condorcet winner returned by *Alg* is played given that $\tau_m \geq \nu_m$ and $a_{I_m} = a_{m^*}$. Thus, there is no weak regret incurred from $\nu_{m-1} + \tilde{T}$ to $\nu_m - 1$, which we used for the second equation. We split the remaining term into:

$$\mathbb{E}_m \left[ R^{\bar{W}}(\nu_m, T) \mid \tau_m \geq \nu_m, a_{I_m} = a_{m^*} \right]$$

$$\leq \mathbb{E}_m \left[ R^{\bar{W}}(\nu_m, T) \mid \nu_m \leq \tau_m < \nu_m + L'/2, a_{I_m} = a_{m^*} \right]$$

$$+ \mathbb{E}_m \left[ R^{\bar{W}}(\nu_m, T) \mid \tau_m \geq \nu_m + L'/2, a_{I_m} = a_{m^*} \right] \cdot q$$

$$\leq \mathbb{E}_m \left[ R^{\bar{W}}(\nu_m, T) \mid \nu_m \leq \tau_m < \nu_m + L'/2, a_{I_m} = a_{m^*} \right] + q \sum_{i=m+1}^{M} |S_i|.$$

On the event that the changepoint is detected with short delay, i.e., $\nu_m \leq \tau_m < \nu_m + L'/2$, we can rewrite the expected regret as:

$$\mathbb{E}_m \left[ R^{\bar{W}}(\nu_m, T) \mid \nu_m \leq \tau_m < \nu_m + L'/2, a_{I_m} = a_{m^*} \right]$$

$$= \mathbb{E}_m \left[ R^{\bar{W}}(\nu_m, \tau_m) \mid \nu_m \leq \tau_m < \nu_m + L'/2, a_{I_m} = a_{m^*} \right]$$

$$+ \mathbb{E}_m \left[ R^{\bar{W}}(\tau_m + 1, T) \mid \nu_m \leq \tau_m < \nu_m + L'/2, a_{I_m} = a_{m^*} \right]$$

$$\leq \frac{L'}{2} + \mathbb{E}_{m+1} \left[ R^{\bar{W}}(\nu_m, T) \right],$$

where we utilized Assumption (1) ensuring us that after DETECT being resetted in $\tau_m$, the next detection phase has at least $L$ time steps left before $\nu_{m+1}$ such that $\nu_{m+1}$ can be detected as if DETECT was started at $\nu_m$. Summarizing, on the event of $\tau_m \geq \nu_m$ and $a_{I_m} = a_{m^*}$ we obtain:

$$\mathbb{E}_m \left[ R^{\bar{W}}(\nu_{m-1} + \tilde{T}, T) \mid \tau_m \geq \nu_m, a_{I_m} = a_{m^*} \right]$$

$$= \mathbb{E}_m \left[ R^{\bar{W}}(\nu_m, T) \mid \tau_m \geq \nu_m, a_{I_m} = a_{m^*} \right]$$

$$\leq \mathbb{E}_m \left[ R^{\bar{W}}(\nu_m, T) \mid \nu_m \leq \tau_m < \nu_m + L'/2, a_{I_m} = a_{m^*} \right] + q \sum_{i=m+1}^{M} |S_i|$$

$$\leq \frac{L'}{2} + \mathbb{E}_{m+1}\left[R^{\bar{\mathrm{W}}}(\nu_m, T)\right] + q\sum_{i=m+1}^{M}|S_i|.$$

Finally, we can combine the intermediate results and apply the induction hypothesis in the fourth inequality to in order to derive:

$$\mathbb{E}_m\left[R^{\bar{\mathrm{W}}}(\nu_{m-1}, T)\right]$$

$$\leq \mathbb{E}\left[\tilde{R}^{\bar{\mathrm{W}}}(\nu_{m-1}, \nu_{m-1} + \tilde{T} - 1)\right] + \left(1 - (1-p_m)p_{\tilde{T}}^{(m)}\right)\sum_{i=m}^{M}|S_i|$$

$$\quad + \mathbb{E}_m\left[R^{\bar{\mathrm{W}}}(\nu_{m-1} + \tilde{T}, T) \mid \tau_m \geq \nu_m, a_{I_m} = a_{m^*}\right]$$

$$\leq \mathbb{E}\left[\tilde{R}^{\bar{\mathrm{W}}}(\nu_{m-1}, \nu_{m-1} + \tilde{T} - 1)\right] + \left(1 - (1-p_m)p_{\tilde{T}}^{(m)}\right)\sum_{i=m}^{M}|S_i|$$

$$\quad + \frac{L'}{2} + \mathbb{E}_{m+1}\left[R^{\bar{\mathrm{W}}}(\nu_m, T)\right] + q\sum_{i=m+1}^{M}|S_i|$$

$$\leq \frac{L'}{2} + q\sum_{i=m+1}^{M}|S_i| + (1 - (1-p)p_{\tilde{T}})\sum_{i=m}^{M}|S_i| + \mathbb{E}\left[\tilde{R}^{\bar{\mathrm{W}}}(\nu_{m-1}, \nu_{m-1} + \tilde{T} - 1)\right]$$

$$\quad + \mathbb{E}_{m+1}\left[R^{\bar{\mathrm{W}}}(\nu_m, T)\right]$$

$$\leq \frac{L'}{2} + q\sum_{i=m+1}^{M}|S_i| + (1 - (1-p)p_{\tilde{T}})\sum_{i=m}^{M}|S_i| + \mathbb{E}\left[\tilde{R}^{\bar{\mathrm{W}}}(\nu_{m-1}, \nu_{m-1} + \tilde{T} - 1)\right]$$

$$\quad + (M - m - 1)\cdot\frac{L'}{2} + q\sum_{i=m+2}^{M}(i - m - 1)|S_i| + \sum_{i=m+1}^{M}\mathbb{E}\left[\tilde{R}^{\bar{\mathrm{W}}}(\nu_{i-1}, \nu_{i-1} + \tilde{T} - 1)\right]$$

$$\quad + (1 - (1-p)p_{\tilde{T}})\sum_{i=m+1}^{M}(i - m)|S_i|$$

$$= (M - m)\cdot\frac{L'}{2} + q\sum_{i=m+1}^{M}(i - m)|S_i| + \sum_{i=m}^{M}\mathbb{E}\left[\tilde{R}^{\bar{\mathrm{W}}}(\nu_{i-1}, \nu_{i-1} + \tilde{T} - 1)\right]$$

$$\quad + (1 - (1-p)p_{\tilde{T}})\sum_{i=m}^{M}(i - m + 1)|S_i|,$$

which proves the hypothesis. $\qquad\square$

Theorem 4 bounds DETECT's expected regret depending on $p$ and $q$, but these are not explicitly given by the problem statement nor the choice of our parameters. A bound solely based on the given parameters is more desirable. We catch up on this by plugging in the bounds obtained in Lemma 9 and 10 directly.

**Corollary 5.** *For any choice of parameters satisfying Assumptions (1) and (2) the expected cumulative binary weak regret of DETECT is bounded by*

$$\mathbb{E}\left[R^{\bar{W}}(T)\right] \leq \frac{ML'}{2} + (1 - p_{\tilde{T}} + pp_{\tilde{T}} + q)MT + \sum_{m=1}^{M}\mathbb{E}\left[\tilde{R}^{\bar{W}}(\nu_{m-1}, \nu_{m-1} + \tilde{T} - 1)\right],$$

*with $p = \frac{2(T-\tilde{T})}{p_{\tilde{T}}}\exp\left(\frac{-2b^2}{w}\right)$ and $q = \exp\left(-\frac{wc^2}{2}\right)$, while $c > 0$ stems from Assumption (2).*

We are again interested in an optimal parameterization that minimizes DETECT's expected regret and derive it to the best of our ability, neglecting some imprecision incurred in the previous intermediate results

and further technical difficulties. First, we consider the window length $w$ and the threshold $b$ independent of the used dueling bandits algorithm and its bound $p_{\tilde{T}}$.

**Corollary** 2 *Setting* $b = \sqrt{\frac{wC'}{2}}$, $c = \sqrt{\frac{2C'}{w}}$, *and* $w$ *to the lowest even integer greater or equal* $\frac{8C'}{\delta_*^2}$ *with any* $C \in \mathbb{R}^+$ *and* $C' \geq \log\left(\frac{M(2T^2+T)}{C}\right)$, *the expected cumulative binary weak regret of DETECT is bounded by*

$$\mathbb{E}\left[R^{\bar{W}}(T)\right] \leq \frac{4C'+1}{\delta_*^2}MK + (1-p_{\tilde{T}})MT + C + \sum_{m=1}^{M} \mathbb{E}\left[\tilde{R}^{\bar{W}}(\nu_{m-1}, \nu_{m-1} + \tilde{T} - 1)\right].$$

*Proof.* To guarantee the bound $(pp_{\tilde{T}} + q)MT \leq C$ for $C \in \mathbb{R}^+$ of our choice of the term contained in Theorem 4, we need to have

$$p \leq \frac{aC}{p_{\tilde{T}}MT} \text{ and } q \leq \frac{(1-a)C}{MT}$$

for some $a \in [0, 1]$. The intention is to choose $C$ as high as possible without changing the asymptotic behavior of the bound given in Theorem 4. With greater $C$ there is more space for the probabilities $p$ and $q$ to increase such that the window length $w$ can be chosen smaller. We fulfill the demands towards $p$ and $q$ posed in Theorem 4 by ensuring that the above mentioned bounds are greater than the ones given in Lemma 9 and 10 to obtain

$$\frac{2T}{p_{\tilde{T}}} \cdot \exp\left(\frac{-2b^2}{w}\right) \leq \frac{aC}{p_{\tilde{T}}MT} \text{ and } \exp\left(-\frac{wc^2}{2}\right) \leq \frac{(1-a)C}{MT}.$$

These inequalities are equivalent to

$$b \geq \sqrt{\frac{w}{2}\log\left(\frac{2MT^2}{aC}\right)} \text{ and } c \geq \sqrt{\frac{2}{w}\log\left(\frac{MT}{(1-a)C}\right)}.$$

In order to simplify further calculations we want $\frac{2MT^2}{aC} = \frac{MT}{(1-a)C}$, from which we derive $a = \frac{2T}{2T+1}$. Using $C' \geq \log\left(\frac{M(2T^2+T)}{C}\right)$ we set:

$$b = \sqrt{\frac{wC'}{2}} \text{ and } c = \sqrt{\frac{2C'}{w}},$$

such that the lower bounds stated above are met. Then we plug this into Assumption (2) in order to obtain for $w$:

$$\delta_* \geq \frac{2b}{w} + c$$

$$\Leftrightarrow \delta_* \geq \sqrt{\frac{8C'}{w}}$$

$$\Leftrightarrow w \geq \frac{8C'}{\delta_*^2}.$$

Finally, we set $w$ to be the lowest even integer above its lower bound and hence due to $L' = w(K-1)$:

$$\frac{ML'}{2} \leq \frac{4C'+1}{\delta_*^2}MK.$$

$\square$

Finally, we want to close the analysis by setting $\tilde{T}$ and consequently $p_{\tilde{T}}$ specifically for the usage of WS and BtW. To this end, we derive $\tilde{T}$ in Corollary 6 and 8 such that the summand $(1 - p_{\tilde{T}})MT$ in Corollary 2 is bounded by $M \log T$ and we set $C = (1 - p_{\tilde{T}})MT$ in order to keep the same asymptotic bounds. We tackle the trade-off between $\tilde{T}$ and $p_{\tilde{T}}$ in Corollary 7 and 9 from the other side by assuming $\tilde{T} = \sqrt{T}$ to be given and calculate the expected regret bounds on the basis of the resulting $p_{\tilde{T}}$. Since we need a valid lower bound $p_{\tilde{T}}$ for all stationary segments, we define $p_{\min}^* := \min_m p_{\min}^{(m)}$ and plug it into the respective lower bounds in Lemma 21 and 23. Note that the value of $\tilde{T}$ can exceed segment lengths for small enough $p_{\min}^*$.

**Corollary 6.** *Setting* $\tilde{T} = \left( \dfrac{\log\left( \dfrac{T}{\left( 1 - e^{-(2p^*_{\min} - 1)^2} \right) \log T} \right)}{(2p^*_{\min} - 1)^2} + K - 1 \right)^2$, $b = \sqrt{w \left( \log T + \frac{3}{8} \right)}$, $c =$

$\sqrt{\frac{1}{w} \left( 4 \log T + \frac{3}{2} \right)}$*, and $w$ to the lowest even integer greater or equal $\frac{16 \log T + 6}{\delta^2_*}$ by the usage of Corollary 2 with $C = M \log T$ the expected cumulative binary weak regret of DETECT using BtW is bounded by*

$$\mathbb{E}\left[ R^{\bar{W}}(T) \right] \leq \frac{8 \log T + 4}{\delta^2_*} MK + 2M \log T + \sum_{m=1}^{M} \mathbb{E}\left[ \tilde{R}^{\bar{W}}\left( \nu_{m-1}, \nu_{m-1} + \tilde{T} - 1 \right) \right].$$

**Corollary 7.** *Let* $x := \dfrac{e^{-\left( T^{1/4} - K + 1 \right)(2p^*_{\min} - 1)^2}}{1 - e^{-(2p^*_{\min} - 1)^2}}$. *Setting* $\tilde{T} = \sqrt{T}$, $b = \sqrt{\frac{w}{2} \log\left( \frac{2T}{x} \right)}$, $c = \sqrt{\frac{2}{w} \log\left( \frac{2T}{x} \right)}$, *and $w$ to the lowest even integer greater or equal $\frac{8}{\delta^2_*} \log\left( \frac{2T}{x} \right)$ by the usage of Corollary 2 with $C = xMT$ the expected cumulative binary weak regret of DETECT using BtW is bounded by*

$$\mathbb{E}\left[ R^{\bar{W}}(T) \right] \leq \frac{4 \log\left( \frac{2T}{x} \right) + 1}{\delta^2_*} MK + 2xMT + \sum_{m=1}^{M} \mathbb{E}\left[ \tilde{R}^{\bar{W}}\left( \nu_{m-1}, \nu_{m-1} + \sqrt{T} - 1 \right) \right].$$

**Corollary 8.** *Setting* $r = \max\left\{ 2, \left\lceil \dfrac{\log\left( \dfrac{2T \cdot \left( 1 + \frac{1 - p^*_{\min}}{2p^*_{\min} - 1} \right)}{\log T} \right)}{\log\left( \frac{p^*_{\min}}{1 - p^*_{\min}} \right)} \right\rceil \right\}$, $\tilde{T} = \frac{r^3 K^3 T}{\log T}$, $b = \sqrt{w \left( \log T + \frac{3}{8} \right)}$, $c =$

$\sqrt{\frac{1}{w} \log\left( 4 \log T + \frac{3}{2} \right)}$*, and $w$ to the lowest even integer greater or equal $\frac{16 \log T + 6}{\delta^2_*}$ by the usage of Corollary 2 with $C = M \log T$ the expected cumulative binary weak regret of DETECT using WS is bounded by*

$$\mathbb{E}\left[ R^{\bar{W}}(T) \right] \leq \frac{8 \log T + 4}{\delta^2_*} MK + 2M \log T + \sum_{m=1}^{M} \mathbb{E}\left[ \tilde{R}^{\bar{W}}(\nu_{m-1}, \nu_{m-1} + \tilde{T} - 1) \right].$$

**Corollary 9.**

*Let* $y := \min_{r \in \mathbb{N}} \left( 1 + \frac{1 - p^*_{\min}}{2p^*_{\min} - 1} \right) \cdot \left( \frac{1 - p^*_{min}}{p^*_{min}} \right)^r - \frac{r^3 K^3}{\sqrt{T}}$. *Setting* $\tilde{T} = \sqrt{T}$, $b = \sqrt{w \log\left( \frac{2T}{y} \right)}$, $c = \sqrt{\frac{2}{w} \log\left( \frac{2T}{y} \right)}$, *and*

*$w$ to the lowest even integer greater or equal $\frac{8}{\delta^2_*} \log\left( \frac{2T}{y} \right)$ by the usage of Corollary 2 with $C = yMT$ the expected cumulative binary weak regret of DETECT using WS is bounded by*

$$\mathbb{E}\left[ R^{\bar{W}}(T) \right] \leq \frac{4 \log\left( \frac{2T}{y} \right) + 1}{\delta^2_*} MK + 2yMT + \sum_{m=1}^{M} \mathbb{E}\left[ \tilde{R}^{\bar{W}}(\nu_{m-1}, \nu_{m-1} + \sqrt{T} - 1) \right].$$

# F  Condorcet Winner Identification Analysis

## F.1  Winner Stays

---

**Algorithm 4** Winner Stays (WS) Chen & Frazier (2017)

---

**Require:** $K \in \mathcal{N}$
  $C_i \Leftarrow 0 \ \forall a_i \in \mathcal{A}$
  **for** $t = 1, \ldots, T$ **do**
    **if** $t > 1$ and $I_{t-1} \in \arg \max_i C_i$ **then**
      $I_t \Leftarrow I_{t-1}$
    **else if** $t > 1$ and $J_{t-1} \in \arg \max_i C_i$ **then**
      $I_t \Leftarrow J_{t-1}$
    **else**
      Draw $I_t$ uniformly at random from $\arg \max_i C_i$
    **end if**
    **if** $t > 1$ and $I_{t-1} \in \arg \max_{i \neq I_t} C_i$ **then**
      $J_t \Leftarrow I_{t-1}$
    **else if  then**$t > 1$ and $J_{t-1} \in \arg \max_{i \neq I_t} C_i$
      $J_t \Leftarrow J_{t-1}$
    **else**
      Draw $J_t$ uniformly at random from $\arg \max_{i \neq I_t} C_i$
    **end if**
    Play $(a_{I_t}, a_{J_t})$ and observe $X^{(t)}_{I_t, J_t}$
    **if** $X^{(t)}_{I_t, J_t} = 1$ **then**
      $C_{I_t} \Leftarrow C_{I_t} + 1$
      $C_{J_t} \Leftarrow C_{J_t} - 1$
    **else**
      $C_{I_t} \Leftarrow C_{I_t} - 1$
      $C_{J_t} \Leftarrow C_{J_t} + 1$
    **end if**
  **end for**

---

The arms' scores depending on the round and iteration are generalized in Lemma 15 that we leave without proof.

**Lemma 15.** *Fix any arbitrary round $r \in \mathbb{N}$ and iteration $i \in \{1, \ldots, K-1\}$ of the Winner Stays algorithm. Then in the beginning of iteration $i$ in round $r$ the incumbent has score $(r-1)(K-1) + i - 1$ and the challenger $1 - r$.*

Next, we will derive a bound for the probability that the winner of the last round is the Condorcet winner given a number of time steps $\tilde{T}$ based on the connection to Gambler's ruin game. We use parts of the proof for the regret bound of WS provided in Peköz et al. (2020), but correct errors that the authors have made. In Gambler's ruin two players participate, one having $a$ dollars at disposal, the other $b$. The game consists of consecutive rounds (not to confuse with rounds of WS) in which the first player wins with probability $p$. The winner of a round receives one dollar from the losing player. As soon as one player has run out of money, the game is over. Each iteration of WS can be viewed as such a game with the dueling arms being the players of the game and the difference between each arm's score to the score at which it loses the round being its money at hand. Throughout the analysis we utilize the following result:

**Lemma 16.** *(Gambler's ruin)*
*In a game of Gambler's ruin with the first player having $a$ dollars and the second player having $b$ dollars, given the winning probability $p$ of the first player over the second player, the first player's probability to win*

*the game is*

$$\frac{1 - \left(\frac{1-p}{p}\right)^a}{1 - \left(\frac{1-p}{p}\right)^{a+b}}.$$

Next, we prove similar to Peköz et al. (2020) lower bounds for the probability of $a_*$ winning a round given that it won the previous round or lost in the other case. For that purpose let $W_{r,i}$ and $L_{r,i}$ be the events that $a_*$ wins iteration $i$ in round $r$ and loses iteration $i$ in round $r$, respectively. Further define $p_{\min} := \min_{i \neq *} p_{*,i}$ and $x := \frac{1-p_{\min}}{p_{\min}}$. Note that $x \in [0,1)$ since $p_{\min} \in \left(\frac{1}{2}, 1\right]$, otherwise $a_*$ would not be the Condorcet winner.

**Lemma 17.** *For all rounds $r \geq 2$ holds*

$$\mathbb{P}(W_{r,K-1} \mid W_{r-1,K-1}) \geq \frac{1 - x^{(r-1)K+1}}{1 - x^{rK}},$$

$$\mathbb{P}(W_{r,K-1} \mid L_{r-1,K-1}) \geq \frac{x}{1 - x^{rK}}.$$

*Proof.* Consider a fixed iteration $i$ in round $r$ with $a_*$ being part of the played pair. Since the incumbent starts with score $(r-1)(K-1) + i - 1$ and the challenger with score $1 - r$ (Lemma 15), the iteration can be seen as a Gambler's ruin game between the incumbent having $(r-1)K + i$ dollars and the challenger having one. Thus using Lemma 16 $a_*$ wins this iteration given that it is the incumbent or the challenger, respectively, with probability

$$\mathbb{P}(W_{r,i} \mid a_* \text{ starts iteration } i \text{ in round } r \text{ as the incumbent}) \geq \frac{1 - x^{(r-1)K+i}}{1 - x^{(r-1)K+i+1}},$$

$$\mathbb{P}(W_{r,i} \mid a_* \text{ starts iteration } i \text{ in round } r \text{ as the challenger}) \geq \frac{1 - x}{1 - x^{(r-1)K+i+1}}.$$

And hence we get for a whole round $r \geq 2$:

$$\mathbb{P}(W_{r,K-1} \mid W_{r-1,K-1}) = \mathbb{P}(W_{r,1} \mid W_{r-1,K-1}) \cdot \prod_{i=2}^{K-1} \mathbb{P}(W_{r,i} \mid W_{r,i-1})$$

$$\geq \prod_{i=1}^{K-1} \frac{1 - x^{(r-1)K+i}}{1 - x^{(r-1)K+i+1}}$$

$$= \frac{1 - x^{(r-1)K+1}}{1 - x^{rK}}.$$

In the other case of $a_*$ having lost the previous round $r - 1$ we can upper bound its winning probability in round $r$ similarly. It is chosen randomly as the challenger for some iteration $j$ in which it has to beat the incumbent before winning in all remaining $K - 1 - j$ iterations. Let $C_{r,i}$ be the event that $a_*$ is chosen as the challenger in iteration $i$ of round $r \geq 2$ given that $L_{r-1,K-1}$, obviously we have $\mathbb{P}(C_{r,i}) \geq \frac{1}{K-1}$ for all $r \geq 2$ and $i$. From which we derive for $r \geq 2$:

$$\mathbb{P}(W_{r,K-1} \mid L_{r-1,K-1}) = \sum_{j=1}^{K-1} \mathbb{P}(C_{r,j}) \cdot \mathbb{P}(W_{r,j} \mid C_{r,j}) \cdot \prod_{i=j+1}^{K-1} \mathbb{P}(W_{r,i} \mid W_{r,i-1})$$

$$\geq \frac{1}{K-1} \sum_{j=1}^{K-1} \frac{1 - x}{1 - x^{(r-1)K+j+1}} \cdot \prod_{i=j+1}^{K-1} \frac{1 - x^{(r-1)K+i}}{1 - x^{(r-1)K+i+1}}$$

$$= \frac{1}{K-1} \sum_{j=1}^{K-1} \frac{1 - x}{1 - x^{rK}}$$

$$= \frac{1 - x}{1 - x^{rK}}.$$

$\square$

With the probability given that $a_*$ wins the next round, we continue by bounding the probability for it to lose the $r$-th round in Lemma 18. Let $W_r := W_{r,K-1}$ and $L_r := L_{r,K-1}$.

**Lemma 18.** *For all rounds $r \geq 1$ holds*

$$\mathbb{P}(L_r) \leq \left(1 + \frac{x}{1-x}\right) \cdot x^r.$$

*Proof.* In the special case of $r = 1$ we say that in the first iteration both randomly chosen arms are challengers such that $\mathbb{P}(C_{1,1}) = \frac{2}{K}$ and $\mathbb{P}(C_{1,j}) \geq \frac{1}{K}$ for any iteration $j \geq 2$. Thus for $r = 1$ we derive with applying Lemma 17:

$$\mathbb{P}(L_1) = 1 - \mathbb{P}(W_1)$$

$$= 1 - \left(\sum_{j=1}^{K-1} \mathbb{P}(C_{1,j}) \cdot \mathbb{P}(W_{1,j} \mid C_{1,j}) \cdot \prod_{i=j+1}^{K-1} \mathbb{P}(W_{1,i} \mid W_{r,i-1})\right)$$

$$\leq 1 - \left(\sum_{j=1}^{K-1} \mathbb{P}(C_{1,j}) \cdot \frac{1-x}{1-x^{j+1}} \cdot \prod_{i=j+1}^{K-1} \frac{1-x^i}{1-x^{i+1}}\right)$$

$$= 1 - \left(\sum_{j=1}^{K-1} \mathbb{P}(C_{1,j}) \cdot \frac{1-x}{1-x^K}\right)$$

$$\leq 1 - \frac{1-x}{1-x^K}$$

$$\leq x.$$

For $r \geq 2$ we can bound the probability recursively by using Lemma 17 again:

$$\mathbb{P}(L_r) = \mathbb{P}(L_r \mid W_{r-1}) \cdot \mathbb{P}(W_{r-1}) + \mathbb{P}(L_r \mid L_{r-1}) \cdot \mathbb{P}(L_{r-1})$$

$$= (1 - \mathbb{P}(W_r \mid W_{r-1})) \cdot (1 - \mathbb{P}(L_{r-1})) + (1 - \mathbb{P}(W_r \mid L_{r-1})) \cdot \mathbb{P}(L_{r-1})$$

$$\leq \frac{x^{(r-1)K+1} - x^{rK}}{1 - x^{rK}} \cdot (1 - \mathbb{P}(L_{r-1})) + \frac{x - x^{rK}}{1 - x^{rK}} \cdot \mathbb{P}(L_{r-1})$$

$$= \mathbb{P}(L_{r-1}) \cdot \frac{x - x^{(r-1)K+1}}{1 - x^{rK}} + \frac{x^{(r-1)K+1} - x^{rK}}{1 - x^{rK}}$$

$$\leq \mathbb{P}(L_{r-1}) \cdot x + x^{(r-1)K+1},$$

which leads to:

$$\mathbb{P}(L_r) \leq x^r + \sum_{s=1}^{r-1} x^{s(K-1)+r},$$

as we prove by induction over $r$. We have already shown the base case for $r = 1$ above, hence only the inductive step for $r + 1$ is left to show:

$$\mathbb{P}(L_{r+1}) \leq \mathbb{P}(L_r) \cdot x + x^{rK+1}$$

$$\leq \left(x^r + \sum_{s=1}^{r-1} x^{s(K-1)+r}\right) \cdot x + x^{rK+1}$$

$$= x^{r+1} + \sum_{s=1}^{r} x^{s(K-1)+r+1}.$$

We can bound the second term in the second last display by:

$$\sum_{s=1}^{r-1} x^{s(K-1)+r} \leq \sum_{s=0}^{r-1} x^{s+r}$$

$$= x^{r+1} \cdot \sum_{s=0}^{r-2} x^s$$

$$= x^{r+1} \cdot \frac{1 - x^{r-1}}{1 - x}$$

$$\leq x^r \cdot \frac{x}{1 - x}.$$

$\square$

We can now say with which probability $a_*$ loses any round $r$, but we are not finished yet because we need to know which is the last round completed by WS after $\tilde{T}$ many time steps. A key quantity for this is the length of a Gambler's ruin game, which can be bounded in expectation.

**Lemma 19.** *Peköz et al. (2020)*
*Let $X$ be the number of plays in a game of Gambler's ruin with one player having $m$ dollars and the other one. Then the expected game length is bounded by*

$$\mathbb{E}[X] < 2(m + 1).$$

**Lemma 20.** *Let $X_{r',i}$ be the number of time steps played in iteration $i$ in round $r'$. For any $a \in \mathbb{R}^+$ and round $r \geq 2$ holds*

$$\mathbb{P}\left(\sum_{r'=1}^{r} \sum_{i=1}^{K-1} X_{r',i} \geq a r^2 K^2\right) < \frac{rK}{a}.$$

*Proof.* Due to Lemma 15 we can interpret each iteration $i$ in any round $r'$ as a game of Gambler's ruin with one player having $(r' - 1)K + i$ dollars and the other one. Thus we obtain

$$\mathbb{E}[X_{r',i}] < 2((r' - 1)K + i + 1).$$

Using Markov's inequality, we can state for any $a \in \mathbb{R}^+$ that:

$$\mathbb{P}(X_{r',i} \geq 2a((r' - 1)K + i + 1)) < \frac{1}{a},$$

and thus further conclude with the help of an average argument for the first inequality:

$$\mathbb{P}\left(\sum_{r'=1}^{r} \sum_{i=1}^{K-1} X_{r',i} \geq \sum_{r'=1}^{r} \sum_{i=1}^{K} 2a((r' - 1)K + i + 1)\right)$$

$$\leq \mathbb{P}\left(\bigcup_{r'=1}^{r} \bigcup_{i=1}^{K-1} \{X_{r',i} \geq 2a((r' - 1)K + i + 1)\}\right)$$

$$\leq \sum_{r'=1}^{r} \sum_{i=1}^{K-1} \mathbb{P}(X_{r',i} \geq 2a((r' - 1)K + i + 1))$$

$$< \sum_{r'=1}^{r} \sum_{i=1}^{K-1} \frac{1}{a}$$

$$< \frac{rK}{a}.$$

The lower bound for the total number of time steps can be rewritten as:

$$\sum_{r'=1}^{r} \sum_{i=1}^{K-1} 2a((r' - 1)K + i + 1)$$

$$= 2a \sum_{r'=1}^{r} r'K(K - 1) + K - 1 - \frac{K(K - 1)}{2}$$

$$= 2a \left( r \left( K - 1 - \frac{K(K-1)}{2} \right) + \frac{K(K-1)r(r+1)}{2} \right)$$
$$= ar(K-1)(rK+2),$$

which can be upper bounded by $ar^2 K^2$ for $r \geq 2$. □

Next, by substituting $a$ with $\frac{\tilde{T}}{r^2 K^2}$, we obtain a probabilistic bound for the first $r$ rounds to take at least $\tilde{T}$ time steps, which gives us a minimal probability for the first $r$ rounds to be completed.

**Corollary 10.** *For any round $r \geq 2$ and number of time steps $\tilde{T}$ holds*

$$\mathbb{P} \left( \sum_{r'=1}^{r} \sum_{i=1}^{K-1} X_{r',i} \geq \tilde{T} \right) < \frac{r^3 K^3}{\tilde{T}}.$$

Finally, what is left to show is with which probability $a^*$ wins the last completed round by combining Lemma 18 and Corollary 10. Let $\tilde{a}_{\tilde{T}}$ be the winner of that round, meaning that it is the incumbent of the first iteration in the succeeding round. We call $\tilde{a}_{\tilde{T}}$ also the *suspected optimal arm* being the one that WS would return if we were to ask it for the optimal arm.

**Lemma 21.** *Fix any $r \in \mathbb{N}$ with $r \geq 2$. Given $\tilde{T}$, the suspected optimal arm $\tilde{a}_{\tilde{T}}$ returned by Winner Stays after $\tilde{T}$ time steps is the true optimal arm $a_*$ with probability*

$$\mathbb{P}(\tilde{a}_{\tilde{T}} = a_*) > 1 - \left( 1 + \frac{x}{1-x} \right) \cdot x^r - \frac{r^3 K^3}{\tilde{T}}.$$

*Proof.* Let $U_{\tilde{T}}$ be the random variable denoting the last round that has been completed after $\tilde{T}$ time steps. We use both Lemma 18 and Corollary 10 in the fourth inequality to derive that:

$$\mathbb{P}(\tilde{a}_{\tilde{T}} = a_*) = \sum_{s=0}^{\infty} \mathbb{P}(\tilde{a}_{\tilde{T}} = a_* \mid U_{\tilde{T}} = s) \cdot \mathbb{P}(U_{\tilde{T}} = s)$$

$$\geq \sum_{s=r}^{\infty} (1 - \mathbb{P}(L_s)) \cdot \mathbb{P}(U_{\tilde{T}} = s)$$

$$\geq (1 - \mathbb{P}(L_r)) \cdot \sum_{s=r}^{\infty} \mathbb{P}(U_{\tilde{T}} = s)$$

$$\geq (1 - \mathbb{P}(L_r)) \cdot (1 - \mathbb{P}(U_{\tilde{T}} \leq r))$$

$$\geq \left( 1 - \left( 1 + \frac{x}{1-x} \right) \cdot x^r \right) \cdot \left( 1 - \frac{r^3 K^3}{\tilde{T}} \right)$$

$$\geq 1 - \left( 1 + \frac{x}{1-x} \right) \cdot x^r - \frac{r^3 K^3}{\tilde{T}}.$$

□

## F.2 Beat the Winner

Same as for WS we will derive a lower bound for the probability that after $\tilde{T}$ time steps the suspected optimal arm by BtW is indeed optimal. Therefore, we define its suspected optimal arm to be the current incumbent. As before, we use $p_{\min} := \min_{i \neq *} p_{*,i}$. The authors in Peköz et al. (2020) have already given a useful property about the last round lost by the optimal arm:

**Lemma 22.** *Peköz et al. (2020)*
*For the last round $U$ lost by the optimal arm and any $r \in \mathbb{N}$ holds*

$$\mathbb{P}(U \geq r) \leq \frac{e^{-r(2p_{min}-1)^2}}{1 - e^{-(2p_{min}-1)^2}}.$$

---

**Algorithm 5** Beat the Winner (BtW) Peköz et al. (2020)

---

**Require:** $K \in \mathbb{N}$
    Draw $I$ uniformly at random from $\{1, \ldots, K\}$
    $Q \Leftarrow$ queue of all arms from $\mathcal{A} \setminus \{a_I\}$ in random order
    $r \Leftarrow 1$
    **while** true **do**
        $a_J \Leftarrow$ top arm removed from $Q$
        $w_I, w_J \Leftarrow 0$
        **while** $w_I < r$ and $w_J < r$ **do**
            Play $(a_I, a_J)$ and observe $X_{I,J}^{(t)}$
            $w_I \Leftarrow w_I + \mathbb{I}\{X_{I,J}^{(t)} = 1\}$
            $w_J \Leftarrow w_J + \mathbb{I}\{X_{I,J}^{(t)} = 0\}$
        **end while**
        **if** $w_I = r$ **then**
            Put $a_J$ to the end of $Q$
        **else**
            Put $a_I$ to the end of $Q$
            $I \Leftarrow J$
        **end if**
        $r \Leftarrow r + 1$
    **end while**

---

Next, we can derive the desired probability bound in Lemma 23 by combining Lemma 22 together with an upper bound on the time steps that are needed for the optimal arm to win its first round after the round $U$ in which it has lost for the last time.

**Lemma 23.** *Given $\tilde{T} \geq K^2$, the suspected optimal arm $\tilde{a}_{\tilde{T}}$ returned by BtW after $\tilde{T}$ time steps is the true optimal arm $a_*$ with probability*

$$\mathbb{P}(\tilde{a}_{\tilde{T}} = a_*) \geq 1 - \frac{e^{-\left(\sqrt{\tilde{T}} - K + 1\right)(2p_{min} - 1)^2}}{1 - e^{-(2p_{min} - 1)^2}}.$$

*Proof.* Let $U$ be the last round lost by $a_*$. Then $a_*$ is played again in round $U + K - 1$ and wins that round. Thus, BtW will always suggest the true optimal arm as the suspected Condorcet winner after the end of round $U + K - 1$. Since the $r$-th round consists of at most $2r - 1$ time steps, we get the following condition for $\tilde{T}$ to guarantee that round $U + K - 1$ is finished:

$$\tilde{T} \geq \sum_{r=1}^{U+K-1} 2r - 1,$$

which is equivalent to

$$U^2 + (2K - 2)U + K^2 - 2K - \tilde{T} + 1 \leq 0.$$

Due to $U \geq 1$, the inequality becomes sharp for $U = \sqrt{\tilde{T}} - K + 1$, which requires $\tilde{T} \geq K^2$. The lefthand side of the inequality decreases for smaller $U \geq 1$, which means that $U \leq \sqrt{\tilde{T}} - K + 1$ implies the above mentioned condition for $\tilde{T}$. Now, we can use Lemma 22 to conclude:

$$\begin{aligned} \mathbb{P}(\tilde{a}_{\tilde{T}} = a_*) &\geq \mathbb{P}\left(\tilde{T} \geq \sum_{i=1}^{U+K-1} 2i - 1\right) \\ &\geq \mathbb{P}(U \leq \sqrt{\tilde{T}} - K + 1) \\ &\geq 1 - \mathbb{P}(U \geq \sqrt{\tilde{T}} - K + 1) \end{aligned}$$

$$\geq 1 - \frac{e^{-\left(\sqrt{\tilde{T}} - K + 1\right)(2p_{\min} - 1)^2}}{1 - e^{-(2p_{\min} - 1)^2}}.$$

$\square$

## G  Non-stationary Dueling Bandits with Weak Regret Lower Bound

In this section, we assume without loss of generality that $T$ is divisible by $M$. Let $\varepsilon \in (0, 1)$. Define segment lengths $|S_m| = T/M$ for all $m \in \{1, \ldots, M\}$. Define preference matrices $P_1, \ldots, P_K$ as follows:

$$P_1(i, j) = \begin{cases} \frac{1}{2} + \varepsilon & \text{if } i = 1, j \neq 1 \\ \frac{1}{2} - \varepsilon & \text{if } i \neq 1, j = 1 \\ \frac{1}{2} & \text{otherwise} \end{cases},$$

$$P_k(i, j) = \begin{cases} \frac{1}{2} + \varepsilon & \text{if } i = k, j \neq k \\ \frac{1}{2} - \varepsilon & \text{if } i \neq k, j = k \\ \frac{1}{2} + \varepsilon & \text{if } i = 1, j \neq 1, j \neq k \\ \frac{1}{2} - \varepsilon & \text{if } i \neq 1, i \neq j, j = 1 \\ \frac{1}{2} & \text{otherwise} \end{cases} \quad \text{for } k \neq 1.$$

Let $H_t = \{(a_{I_{t'}}, a_{J_{t'}}, X^{(t')}_{I_{t'}, J_{t'}})\}_{t' \leq t}$ be the history of observations up to time step $t$. Let $Q_k^m$ denote the probability distribution over histories induced by assuming that the preference matrix $P_k$ is used for the $m$-th segment, i.e., $P^{(m)} = P_k$, and thus $a_k$ is the Condorcet winner in the $m$-th segment. The corresponding expectation is denoted by $\mathbb{E}_k^m[\cdot]$. In the case of $M = 1$ we omit the superscripts in order to simplify notation.

**Lemma 24.** *Saha & Gupta (2022)*
*Let $\mathcal{H}_T$ be the set of all possible histories $H_T$ for $M = 1$ and let $f : \mathcal{H}_T \to [0, B]$ be a measurable function that maps a history $H_T$ to number in the interval $[0, B]$. Then for every $k \neq 1$ holds*

$$\mathbb{E}_k[f(H_T)] \leq \mathbb{E}_1[f(H_T)] + B\sqrt{\varepsilon \ln\left(\frac{1 + 2\varepsilon}{1 - 2\varepsilon}\right) \mathbb{E}_1[N_{1,k} + N_k]},$$

*where $N_{1,k} = \sum_{t=1}^{T} \mathbb{I}\{I_t = 1, J_t = k\} + \mathbb{I}\{I_t = k, J_t = 1\}$ is the number of times the algorithm chooses to duel the arms $a_1$ and $a_k$, and $N_k = \sum_{t=1}^{T} \mathbb{I}\{I_t = k, J_t \neq 1, J_t \neq k\} + \mathbb{I}\{I_t \neq 1, I_t \neq k, J_t = k\}$ is the number of times $a_k$ is played with an arm other than itself and or $a_1$.*

**Theorem 5** *For every algorithm exists an instance of*

**(i)** *the piecewise-stationary dueling bandits problem with $T$, $K$, and $M$ fulfilling $M(K - 1) \leq 9T$ such that the algorithm's expected weak regret is in $\Omega(\sqrt{KMT})$.*

**(ii)** *the stationary dueling bandits problem with $T$ and $K$ fulfilling $K - 1 \leq 9T$ such that the algorithm's expected weak regret is in $\Omega(\sqrt{KT})$.*

*Proof.* Let $t \in S_m$ and $a_{k_m}$ be the Condorcet winner in the $m$-th stationary segment, i.e., $P^{(m)} = P_{k_m}$. Define the following events:

- $\mathcal{E}_a^t = \{I_t = 1, J_t = k_m\} \cup \{I_t = k_m, J_t = 1\}$

- $\mathcal{E}_b^t = \{I_t = k_m, J_t \notin \{1, k_m\}\} \cup \{I_t \notin \{1, k_m\}, J_t = k_m\}$

- $\mathcal{E}_c^t = \{I_t = k_m, J_t = k_m\}$

- $\mathcal{E}_d^t = \{I_t \neq k_m, J_t \neq k_m\}$

Recall that

$$r_t^{\mathrm{W}} := \min\{\Delta_{I_t}^{(m)}, \Delta_{J_t}^{(m)}\} = \min\left\{p_{k_m,I_t}^{(m)}, p_{k_m,J_t}^{(m)}\right\} - \frac{1}{2}$$

is the incurred weak regret at time step $t \in S_m$. For the expected regret at time step $t \in S_m$ under the distribution $Q_{k_m}^m$ we derive:

$$
\begin{aligned}
\mathbb{E}_{k_m}^m[r_t^{\mathrm{W}}] &= \varepsilon Q_{k_m}^m(\mathcal{E}_d^t) \\
&= \varepsilon(1 - Q_{k_m}^m(\mathcal{E}_a^t) - Q_{k_m}^m(\mathcal{E}_b^t) - Q_{k_m}^m(\mathcal{E}_c^t)) \\
&= \varepsilon - \varepsilon(Q_{k_m}^m(\mathcal{E}_a^t) + Q_{k_m}^m(\mathcal{E}_b^t) + Q_{k_m}^m(\mathcal{E}_c^t)).
\end{aligned}
$$

Define $N_{1,k_m}^m = \sum_{t \in S_m} \mathbb{I}\{\mathcal{E}_a^t\}$, $N_{k_m}^m = \sum_{t \in S_m} \mathbb{I}\{\mathcal{E}_b^t\}$, $\bar{N}_{k_m}^m = \sum_{t \in S_m} \mathbb{I}\{\mathcal{E}_c^t\}$. We further obtain:

$$
\begin{aligned}
\mathbb{E}_{k_m}^m\left[\sum_{t \in S_m} r_t^{\mathrm{W}}\right] &= \mathbb{E}_{k_m}^m\left[\sum_{t \in S_m} \varepsilon - \varepsilon(Q_{k_m}^m(\mathcal{E}_a^t) + Q_{k_m}^m(\mathcal{E}_b^t) + Q_{k_m}^m(\mathcal{E}_c^t))\right] \\
&= |S_m|\varepsilon - \varepsilon\mathbb{E}_{k_m}^m\left[\sum_{t \in S_m} Q_{k_m}^m(\mathcal{E}_a^t) + Q_{k_m}^m(\mathcal{E}_b^t) + Q_{k_m}^m(\mathcal{E}_c^t)\right] \\
&= |S_m|\varepsilon - \varepsilon\mathbb{E}_{k_m}^m[N_{1,k_m}^m + N_{k_m}^m + \bar{N}_{k_m}^m].
\end{aligned}
$$

Note that $N_{1,k_m}^m + N_{k_m}^m + \bar{N}_{k_m}^m \leq |S_m|$ for all $H_T \in \mathcal{H}_T$. Next, we can apply Lemma 24 since $N_{1,k_m}^m + N_{k_m}^m + \bar{N}_{k_m}^m$ is measurable for all histories:

$$
\begin{aligned}
&\mathbb{E}_{k_m}^m[N_{1,k_m}^m + N_{k_m}^m + \bar{N}_{k_m}^m] \\
&\leq \mathbb{E}_1^m[N_{1,k_m}^m + N_{k_m}^m + \bar{N}_{k_m}^m] + |S_m|\sqrt{\varepsilon\ln\left(\frac{1+2\varepsilon}{1-2\varepsilon}\right)\mathbb{E}_1^m[N_{1,k_m}^m + N_{k_m}^m]}.
\end{aligned}
$$

The switch from $M = 1$ as it is demanded in Lemma 24 to the non-stationary case is justified because for the measure $N_{1,k_m}^m + N_{k_m}^m + \bar{N}_{k_m}^m$ only the part of a history containing the $m$-th segment is relevant. Using Cauchy-Schwarz and $\ln\left(\frac{1+2\varepsilon}{1-2\varepsilon}\right) \leq 9\varepsilon$ for $\varepsilon \in (0, 1/4)$ we get:

$$
\begin{aligned}
&\sum_{k_m=2}^{K} \mathbb{E}_{k_m}^m[N_{1,k_m}^m + N_{k_m}^m + \bar{N}_{k_m}^m] \\
&\leq \mathbb{E}_1^m\left[\sum_{k_m=2}^{m} N_{1,k_m}^m + N_{k_m}^m + \bar{N}_{k_m}^m\right] + |S_m|\sum_{k_m=2}^{m}\sqrt{\varepsilon\ln\left(\frac{1+2\varepsilon}{1-\varepsilon}\right)\mathbb{E}_1^m[N_{1,k_m}^m + N_{k_m}^m]} \\
&\leq \mathbb{E}_1^m\left[\sum_{k_m=2}^{m} N_{1,k_m}^m + N_{k_m}^m + \bar{N}_{k_m}^m\right] + |S_m|\sqrt{\varepsilon\ln\left(\frac{1+2\varepsilon}{1-\varepsilon}\right)(K-1)\sum_{k_m=2}^{m}\mathbb{E}_1^m[N_{1,k_m}^m + N_{k_m}^m]}
\end{aligned}
$$

$$\leq |S_m| + |S_m| \sqrt{\varepsilon \ln\left(\frac{1+2\varepsilon}{1-2\varepsilon}\right)(K-1)|S_m|}$$

$$\leq |S_m| + 3|S_m| \sqrt{\varepsilon^2(K-1)|S_m|},$$

which results in the following bound:

$$\sum_{k_m=2}^{K} \mathbb{E}_{k_m}^m \left[\sum_{t\in S_m} r_t^{\mathrm{W}}\right] = \sum_{k_m=2}^{K} |S_m|\varepsilon - \varepsilon\mathbb{E}_{k_m}^m[N_{1,k_m}^m + N_{k_m}^m + \bar{N}_{k_m}^m]$$

$$= |S_m|\varepsilon(K-1) - \varepsilon \sum_{k_m=2}^{K} \mathbb{E}_{k_m}^m[N_{1,k_m}^m + N_{k_m}^m + \bar{N}_{k_m}^m]$$

$$\geq |S_m|\varepsilon(K-1) - \varepsilon(|S_m| + 3|S_m|\sqrt{\varepsilon^2(K-1)|S_m|})$$

$$= |S_m|\varepsilon(K-1-1-3\varepsilon\sqrt{(K-1)|S_m|}).$$

Recall that $R^{\mathrm{W}}(T) = \sum_{m=1}^{M}\sum_{t\in S_m} r_t^{\mathrm{W}}$. Thus, assuming $K \geq 3$, the expected cumulative weak regret of any deterministic algorithm under a randomly sampled problem instance (with segment lengths $|S_m| = T/M$) with randomly chosen preference matrices $P^{(m)} = P_k$ with $k \neq 1$ for all $m \in \{1,\ldots,M\}$, is

$$\mathbb{E}[R^{\mathrm{W}}(T)] = \sum_{m=1}^{M} \frac{1}{K-1} \sum_{k_m=2}^{K} \mathbb{E}_{k_m}^m\left[\sum_{t\in S_m} r_t^{\mathrm{W}}\right]$$

$$\geq \sum_{m=1}^{M} \frac{1}{K-1}|S_m|\varepsilon(K-1-1-3\varepsilon\sqrt{(K-1)|S_m|})$$

$$= \sum_{m=1}^{M} \frac{1}{K-1}\frac{T}{M}\varepsilon\left(K-1-1-3\varepsilon\sqrt{(K-1)\frac{T}{M}}\right)$$

$$= \frac{T(K-2)}{K-1}\varepsilon - 3\varepsilon^2\sqrt{\frac{T^3}{M(K-1)}}$$

$$\geq \frac{T}{2}\varepsilon - 3\varepsilon^2\sqrt{\frac{T^3}{M(K-1)}}.$$

We choose $\varepsilon = \frac{1}{12}\sqrt{\frac{M(K-1)}{T}} \leq 1/4$ to maximize the expression and obtain non-stationary setting:

$$\mathbb{E}\left[R^{\mathrm{W}}(T)\right] \geq \frac{1}{48}\sqrt{TM(K-1)}.$$

Using Yao's minimax principle (Yao, 1977), we can infer the latter lower bound also for randomized algorithms. Finally, we conclude for the stationary setting, i.e. $M = 1$:

$$\mathbb{E}\left[R^{\mathrm{W}}(T)\right] \geq \frac{1}{48}\sqrt{T(K-1)}.$$

$\square$

## H   Further Empirical Results [1]

### H.1   Weak Regret

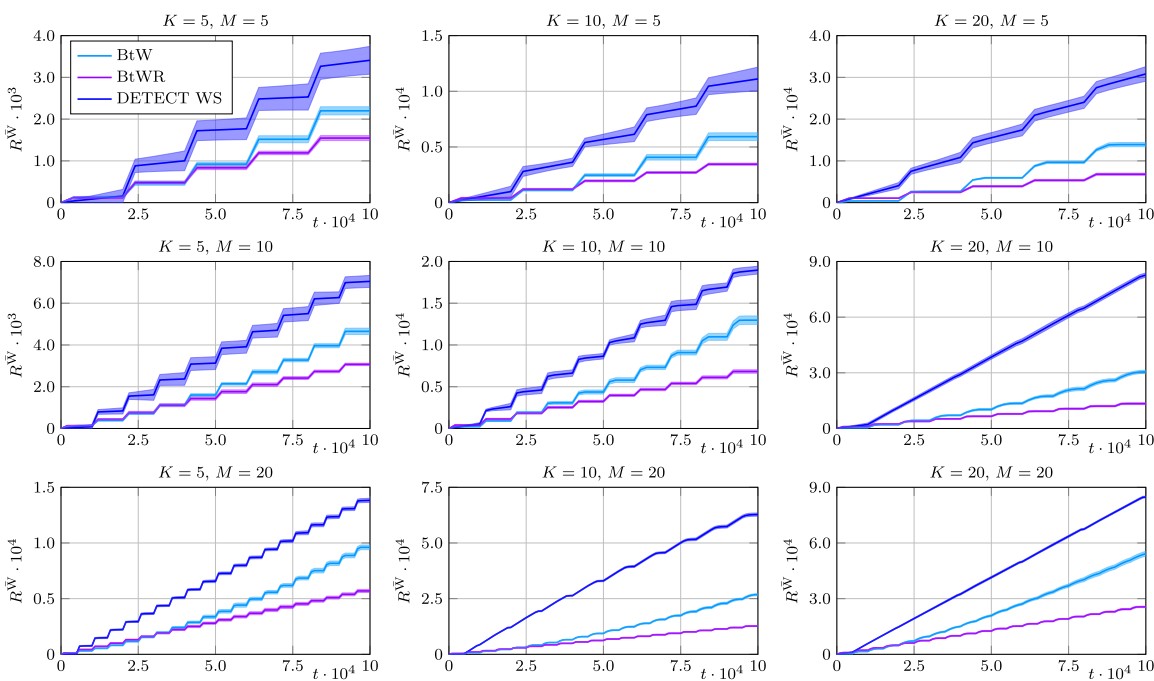

Figure 8:   Cumulative binary weak regret averaged over 500 random problem instances with shaded areas showing standard deviation across 10 groups: $T = 10^5$, $\Delta = 0.1$, $\delta^* = 0.6$.

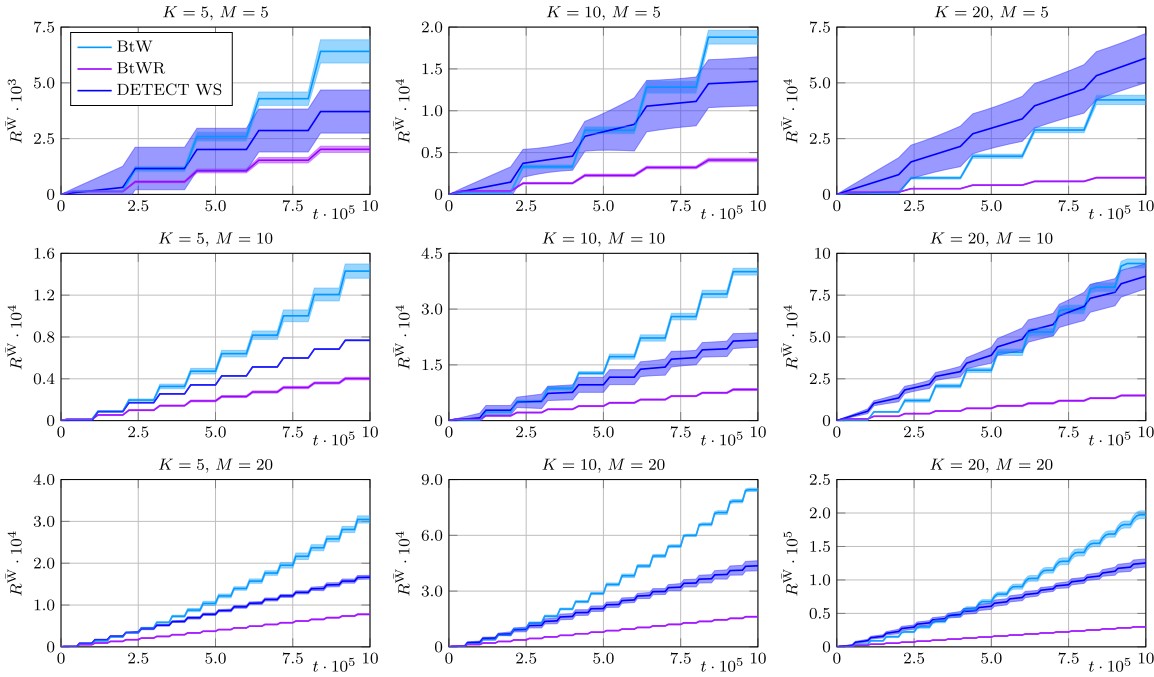

Figure 9:   Cumulative binary weak regret averaged over 500 random problem instances with shaded areas showing standard deviation across 10 groups: $T = 10^6$, $\Delta = 0.1$, $\delta^* = 0.6$.

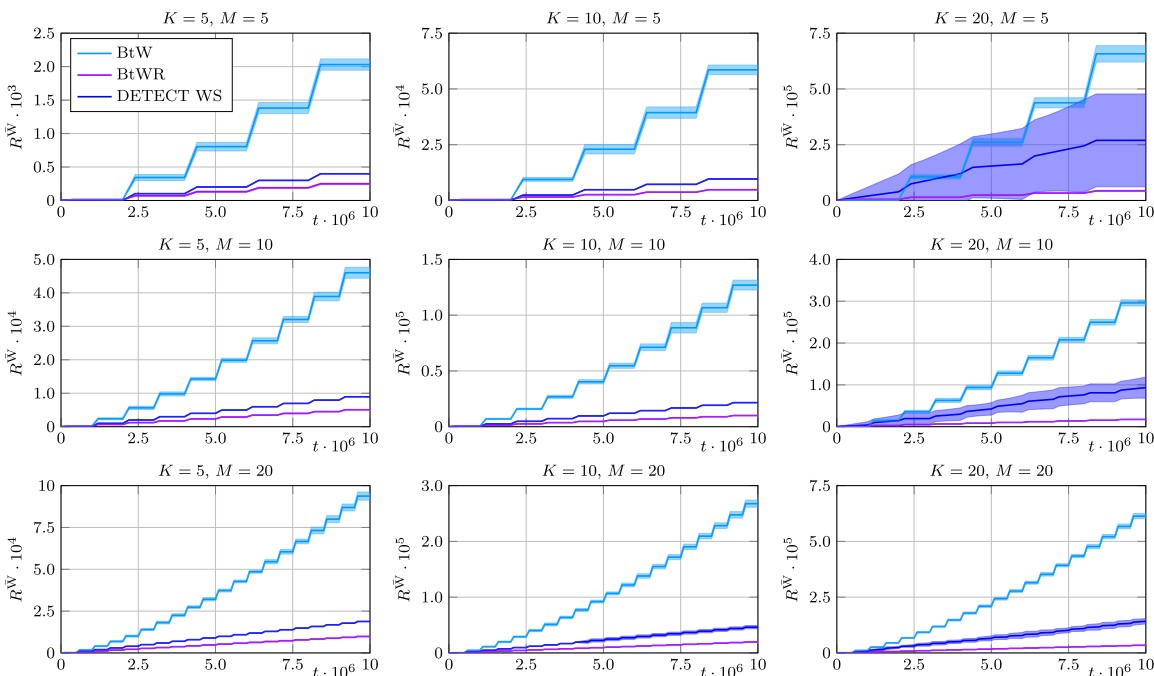

Figure 10: Cumulative binary weak regret averaged over 500 random problem instances with shaded areas showing standard deviation across 10 groups: $T = 10^7$, $\Delta = 0.1$, $\delta^* = 0.6$.

## H.2 Strong Regret

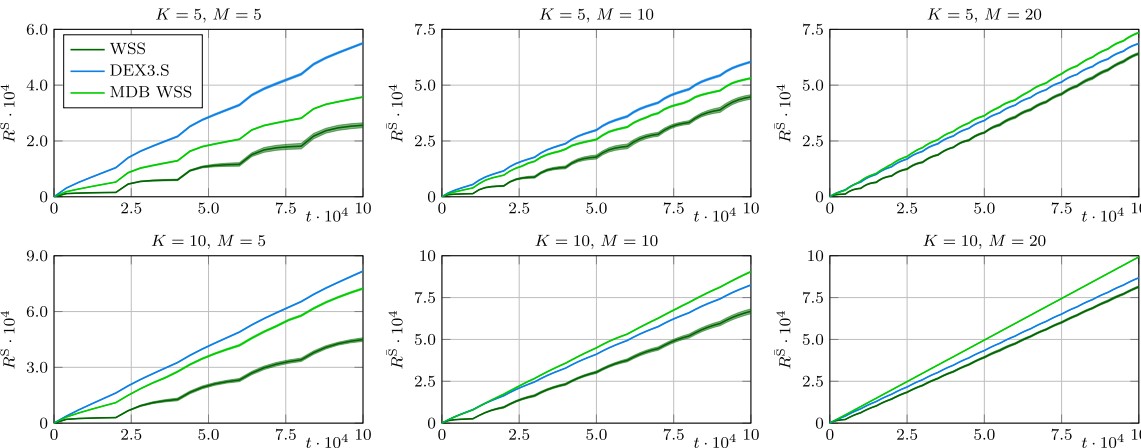

Figure 11: Cumulative binary strong regret averaged over 500 random problem instances with shaded areas showing standard deviation across 10 groups, WSS with exploitation parameter $\beta = 1.05$: $T = 10^5$, $\Delta = 0.1$, $\delta^* = 0.6$.

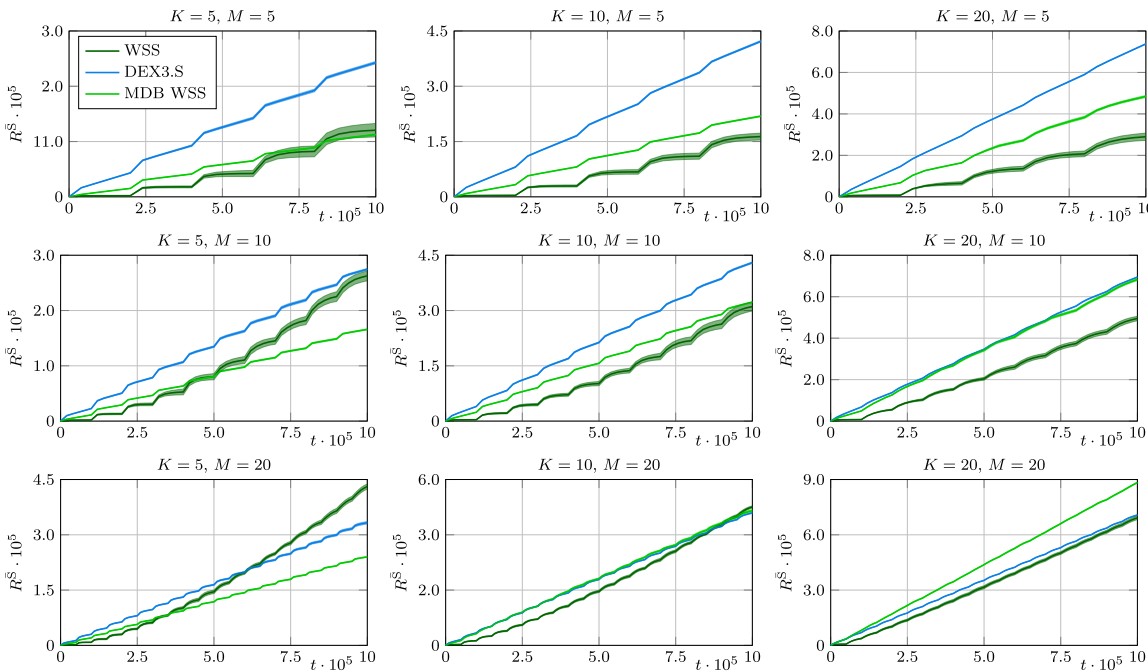

Figure 12: Cumulative binary strong regret averaged over 500 random problem instances with shaded areas showing standard deviation across 10 groups, WSS with exploitation parameter $\beta = 1.05$: $T = 10^6$, $\Delta = 0.1$, $\delta^* = 0.6$.

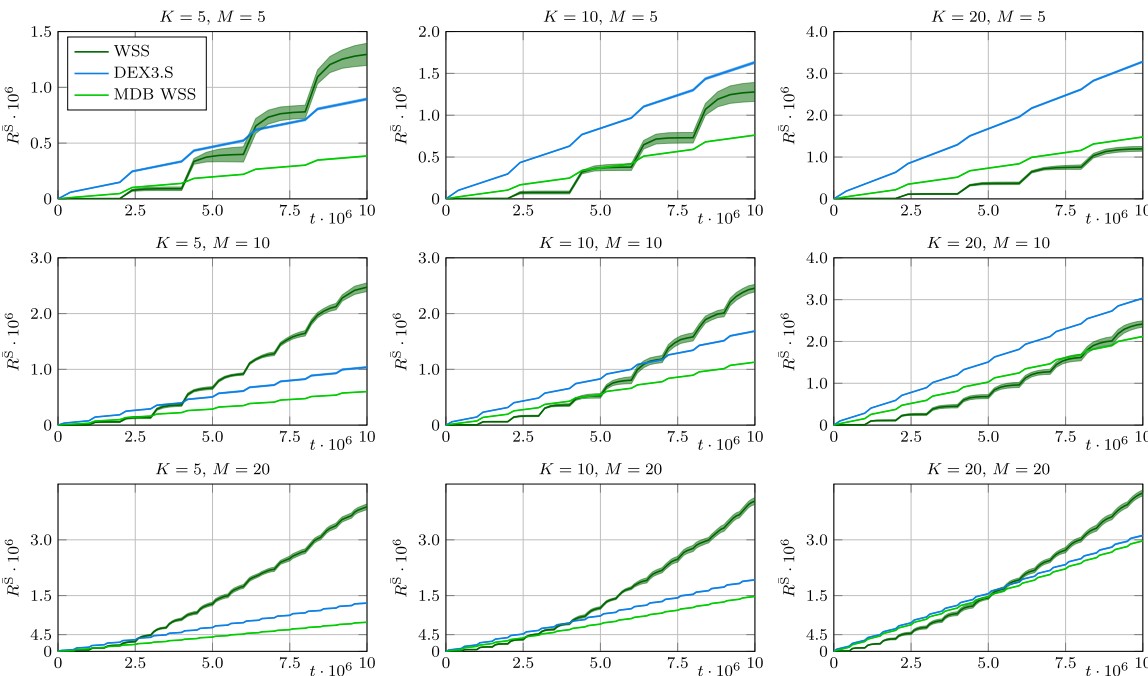

Figure 13: Cumulative binary strong regret averaged over 500 random problem instances with shaded areas showing standard deviation across 10 groups, WSS with exploitation parameter $\beta = 1.05$: $T = 10^7$, $\Delta = 0.1$, $\delta^* = 0.6$.

