# OpenReview forum: "Piecewise-Stationary Dueling Bandits"
_TMLR — Accepted by TMLR_

### Review · Reviewer_NJ8p · 2024-04-29

**Summary Of Contributions:**

This paper studies the piecewise-stationary dueling bandits problem with $K$ actions,
in which $M-1$ change points and preference matrix $P^{(m)} \in [0,1]^{K \times K}$  can be different in segment $m \in [M]$. This study considered the Condorcet winner of each stationary segment to define the regret and propose three algorithms for piecewise stationary dueling bandits:Beat the Winner Reset, Monitored Dueling Bandits, and DETECT algorithms. They consider the weak regret (Yue et al., 2012), which is the minimum of the chosen arms’ calibrated probabilities as well as a standard dynamic regret. Finally, they evaluate BtWR, MDB, and DETECT empirically on synthetic problem instances, to compare with algorithms for the stationary setting.

**Audience:**

Yes

**Claims And Evidence:**

Yes

**Requested Changes:**

Request Changes:

1. Please add a more detailed discussion with existing bounds  (Saha and Gupta, 2022; Buening and Saha, 2023; Suk and Agarwal, 2023) . For example, MDB requires the knowledge of M, while some of existing algorithms does not.

2. In Table, $\delta$ or $\delta^*$ should be introduced in a higher level like minimal segmental change w.r.t. CW since these are not defined until Section 4.

3. In Table, please add references for corresponding Theorems if it is useful. As a minor comment, to denote whether or not the method requires knowledge, cmark or check mark might be confusing and can be replaced with better notation here.

4. In Section 4: p.8. -line 8 typo: with with → with

5. In Section 4, what is the lower bound shown by Gupta & Saha (2021)? It can be stated here.

6.  In Theorem 5,  “For every algorithm exists an instance of..” seems grammatically strange

Questions

- Does Winner Stays algorithm (Chen & Frazier, 2017) require the knowledge of $\Delta$?

- In Section 3, what is $\tilde{c}$? We also have $c$ and $c_1$ here.  Is $c$ just a constant in the end or can be grow with time step t?

- In MDB in Section 4, change-point detection scheme is triggered depending on the w and thresholds $b$ and $\tau$.  So $w, b$ and $γ$ are the tuning parameters. However, Assumption 2 requires $\delta$ to be larger than $2b/w +c$. Should we set $w, b$ and $γ$  so that it holds that $2b/w +c$, instead of adding this condition as an assumption?

 - In experiments, cannot (Saha and Gupta, 2022; Buening and Saha, 2023; Suk and Agarwal, 2023) be implemented in the experiments as a baseline?

**Strengths And Weaknesses:**

Strength:

- For weak regret, they first propose the Winner Reset algorithm (BtWR) based on (Peköz et al., 2020) and prove $\mathcal{O}\left(\frac{K \log K}{\Delta^{2}}\right)$ bound.

- They proposed Monitored Dueling Bandits (MDB),  detection window approach for the strong regret,  based on MUCB (Cao et al., 2019)  and showed $\mathcal{O}\left(\frac{K \sqrt{M T \log (\delta T / K M)}}{\delta}\right)+R_{M}^{\bar{S}}(A l g)$. The algorithm works with black-box dueling bandits algorithm Alg for the stationary setting

- They further devise DETECT algorithm, a specialization of MDB for weak regret and attain $\mathcal{O}\left(\frac{M K \log T}{\delta_{*}^{2}}+M \log T\right)+R_{M}^{\bar{W}}(A l g)$ regret bound.

- Algorithm descriptions are well-explained and highlight the core of ideas as well. Overall, this paper provided an important series of algorithms and regret analysis. Theoretical analyses are based on existing work (Cao et al., 2019) but modification to dueling bandits requires different techniques.

Weakness

- (BtWR) requires the knowledge of suboptimality gap $\Delta$.

- Comparison with existing bounds for non-stationary dueling bandits is not satisfied. Specifically, there are three work for non-stationary dueling bandits (Saha and Gupta, 2022; Buening and Saha, 2023; Suk and Agarwal, 2023) and the precise comparison seems missing in the current version.

---

> ### Author Response · Authors · 2024-06-24
> **Answer to Reviewer NJ8p**
>
> We thank the reviewer for the questions and suggested changes and his/her eye for detail! We will incorporate them as much as the page limit allows.
>
> No, the Winner Stays algorithm does not require the knowledge of $\Delta$.
>
> $c$ is a variable in the BtWR algorithm and counts the length of the winning streak of the current incumbent arm $a_I$. $c_1$ is the c-value in the very first round. It holds $c_1=1$ at initialization. However, by inspecting a single stationary segment succeeding a previous segment, BtWR can be viewed as tackling the stationary dueling problem with a higher $c_1$ caused by the rounds in the previous segment (see Lemma 4). For this case, we assume $c_1$ to be some $\tilde{c} \geq 1$.
>
> Your suggestion regarding Assumption 2 is exactly what we pursued. Requiring $\delta$ to be larger than $2b/w +c$ necessary for the analysis and the derivation of the parameters. Our specified values of $w$, $b$, and $c$ in Corollary 1 satisfy the inequality automatically. Hence, this assumption is only directed at the parameters and not the problem statement itself and can be left out given our specification.
>
> We politely point out that the mentioned works have yet not existed at the point of time we made our work public for the first time (February 2022). Hence, the state of the art to take as baselines was thinner.

---

> > ### Comment · Reviewer_NJ8p · 2024-07-24
> > **Re: Answer to Reviewer NJ8p**
> >
> > Thank you for the feedback.
> >
> > >Comparison with existing bounds for non-stationary dueling bandits is not satisfied. Specifically, there are three work for non-stationary dueling bandits (Saha and Gupta, 2022; Buening and Saha, 2023; Suk and Agarwal, 2023) and the precise comparison seems missing in the current version.
> >
> > My concern regarding the above still needs to be resolved. I am not asking authors to implement algorithms of (Saha and Gupta, 2022; Buening and Saha, 2023; Suk and Agarwal, 2023) experimentally, but include a comparison or discussion such as regret analysis or algorithm designs. Those works have already been published one or two years ago, thus missing such discussion makes this submission incomplete, in my opinion. If a prior version of this submission appeared publicly before these studies, this could be noted in a footnote or something to that effect, which would not detract from the contribution of this paper.

---

> ### Author Response · Authors · 2024-08-06
>
> Thanks for the literature hint! It turns out that our manuscript shares with the mentioned works only the setting in which the number of preference matrix or Condorcet winner changes is counted (other proposed settings bound the change within the preference matrix and come with different environmental parameters) and the regret is measured via dynamic strong regret as we did for our proposed algorithm MDB. None of these works include results for weak regret as we analyzed for BtWR and DETECT.
>
> Saha & Gupta (2022) propose with Dex3.S an algorithm inspired by EXP3 from the adverserial setting that achieves in our notation a high probability regret bound: $O(\sqrt{KMT} \log KT/\delta)$ strong regret with at least $1-2\delta$ probability if the number of stationary segments M is known. We instead, derive for MDB a bound on the expected regret.
>
> Buening & Saha (2023) rely on this previous work and ours in order to propose with ANACONDA an algorithm that adapts to the environment without requiring the number of segments. It achieves an expected strong regret of $\tilde{O}(K \sqrt{MT})$, hence scaling very similarly to the bounds we provided for MDB.
>
> Suk & Agarwal (2023) also consider strong regret by counting Condorcet winner changes and achieve with METASWIFT $\tilde{O}(\sqrt{KMT})$ strong regret but under the relatively strong assumption of Stochastic Transivity and the Stochastic Triangle Inequality wich we and none of the other works require.
>
> In contrast, considering weak regret allows for better results as we achieve $O(\log T)$ in the dynamic setting whereas $\Omega(\sqrt{T})$ is the lower bound for strong regret.

---

### Review · Reviewer_P87q · 2024-05-14

**Summary Of Contributions:**

This paper studies a piecewise stationary K-armed dueling bandit problem, where the time horizon is divided into M segments on each of which the environment is characterized by a fixed pairwise preference matrix. The M-1 change points as well as the preference matrices are unknown to the learner. It is assumed that on each segment, there exists some Condorcet winner arm. The goal of the learner is to minimize regret over T rounds. The authors propose three algorithms (Beat the Winner Reset, Monitored Dueling Bandits and DETECT) and prove theoretical guarantees on their performance. Numerical experiments demonstrating gains are provided as well.

**Audience:**

Yes

**Broader Impact Concerns:**

N/A.

**Claims And Evidence:**

Yes

**Requested Changes:**

None.

**Strengths And Weaknesses:**

I must preface my comments with the disclaimer that I lack familiarity with referenced literature on dueling bandits. That said, I found this paper easy to read, and the arguments sound. The pictorial representations of the update schemes underling the 3 algorithms were quite helpful in understanding the dynamics (more so than the text or the algorithmic pseudo-code itself).

Proofs were not evaluated for technical correctness.

---

> ### Author Response · Authors · 2024-06-24
> **Answer to Reviewer P87q**
>
> We are thankful to read that the readability and provided illustrations within our manuscript contributed to a better understanding of the paper!

---

### Review · Reviewer_EUKZ · 2024-06-11

**Summary Of Contributions:**

The paper studies non-stationary dueling bandits where the preferences change over time and give regret bounds scaling with the number of such changes. Unlike prior/related works, they focus on binary versions of weak and strong regret, and have several different algorithms which get such regret bounds by restarting stationary algorithms or using detection-window approaches.

**Audience:**

Yes

**Broader Impact Concerns:**

None.

**Claims And Evidence:**

Yes

**Requested Changes:**

I think this work could be of interest to TMLR readers, but there could be more substantial discussion on the limitations of this work (e.g., better comparison with other non-stationary dueling bandit works, whether regret bounds are tight, binary vs. non-binary regret) as outlined above.

**Strengths And Weaknesses:**

I don't quite understand the comparison with other recent non-stationary dueling bandit works in "Related Work" on p. 3. It is said that "these works consider dynamic average regret, which is a stricter notion of regret as the learner’s selection are compared against a dynamic benchmark arm every time step", but isn't this the same as the piecewise-stationary strong regret you define in Section 2? Note that the mentioned works like Gupta & Saha (2021) or Buening & Saha (2022) also get strong regret bounds in terms of $M$ changes and their dynamic regret boils down to arm-sequence benchmark which only changes $M$ times. So, I think the results of such works should be comparable at least for strong regret.

In fact, I see that the real reason it may be hard to compare results with these other works is because you also consider the _binary_ version of the regret in this paper. For me, this is like counting the sample complexity of finding the best arm (i.e., you pay regret of $1$ even when the gap of your played arms is very small). This explains why your regret bounds scale like $K/\Delta^2$ rather than $K/\Delta$. But, it makes me wonder why we cannot just use a "winner arm identification" procedure instead to minimize binary strong or weak regret.

One of the claims of this work is that, for stationary setting and static regret, BtWR has superior binary weak regret $K/\Delta^2$ over previous state-of-the-art. But, for winner-arm-identification in dueling bandits, it was recently shown how to get $K/\Delta^2$ sample complexity (see "Near-Optimal Pure Exploration in Matrix Games: A Generalization of Stochastic Bandits & Dueling Bandits", Maiti et al., 2023 which gets actually a slightly tighter quantity than $K/\Delta^2$, and discussion in Section 1.3 of the reference about dueling bandits). So, one of my main questions is why such procedures cannot be used instead of BtWR?

On a related note, the lower bounds of Section 6 for weak regret don't seem very relatable to the upper bounds, I suspect because this paper focuses on lower bounding the non-binary version of the regret. It seems like, intuitively, the binary regret can also be lower bounded just by reducing to the winner identification problem and showing a regret lower bound of order $K/\Delta^2$? This way, you can have lower bounds which match your upper bounds.

The other weakness I see is that all the non-stationary regret upper bounds of this work seem to require some knowledge of the minimum gap $\Delta$ (across all stationary periods) or number of stationary periods $M$. This is in contrast to recent works like Buening & Saha (2023) and Suk & Agarwal (2023) which design _adaptive_ procedures and the related non-stationary MAB problem where it's been known for a while that one can get adaptive dynamic regret without knowledge of $M$ (e.g., Auer et al., 2019). The saving grace here is that it's unclear whether the other works give any meaningful guarantees for the binary version of regret since they focus on really minimizing the strong regret and not on sample complexity. I think the submission deserves some more careful discussion of why it might be hard to get matching results like that of Buening & Saha (2023) though for the binary version of regret.

It's also unclear that the regret upper bounds really are tight. For instance, the $MK/\Delta^2$ rate is definitely worst-case as $\Delta$ is the smallest suboptimality gap across all stationary segments. I would have thought a rate like $\sum_{i=1}^M K/\Delta_i^2$ makes more sense where $\Delta_i$ is the suboptimality gap of segment $i$. But, I imagine one has to assume a lot of knowledge (like knowing all the $\Delta_i$'s or when the segments begin and end) in order to get such results using the techniques of this paper. A similar argument can be made for the regret upper bounds of MDB and DETECT.

There are also some additional structural assumptions (Assumptions 1-3 on p. 8 and Assumptions 1-2 on p. 10) seemingly required for the MDB and DETECT results, which I feel should be mentioned earlier in the Contribution/Table 1 section. There could be some more discussion on whether such assumptions are realistic or, for example, embodied in the environments used for experiments in this papaer.

# Writing Notes/Typos
* the _segmental change_ quantities $\delta_{i,j}^{(m)},\delta,\delta_*$ could be introduced earlier in Section 2 or near Table 1 as otherwise the results of Table 1 have undefined quantities.
* Figure 2 caption typo: "bandis".
* page 2 typo: "suboptimatlity"

---

> ### Author Response · Authors · 2024-06-24
> **Answer to Reviewer EUKZ**
>
> We thank the reviewer for his fruitful questions and the effort made to detect spelling mistakes!
> Yes, the regret notions in the mentioned works are (except for the binarization) identical to ours in the dynamic / piecewise-stationary setting. Our term "stricter notion" is in relation to the previously mentioned works considering stationary bandits for which the reference arm is fixed over time.
>
> The binarization of the regret expresses that wrongly chosen arms are punished equally independent of how close their probability comes to beating the Condorcet winner on average. This alone still does not lead to a “winner arm identification” problem as the total number of wrongly chosen pairs (equalling binary regret) is to be minimized. Returning the optimal arm at termination (when all budget is spent) with maximal probability or spending as few samples as possible to reach a certain confidence threshold is a task, known as pure exploration, of different nature which we do not seek to solve.
>
> The work by Maiti et al. considers the pure exploration problem in which the optimal is to be found with high confidence using a minimal number of samples. We instead are not interested in the identification of the optimal arm but aim to minimize the number of times it has not been chosen in a pair (twice for weak regret and once for strong regret). These two problems are of different kind hence a comparison is not meaningful to us.
>
> It is correct and we do not hide that the minimum gap $\Delta$ must be known by BtWR to achieve the stated regret bounds. As we are aware of this, we investigated the impact of a misspecification in Section 7.3. Figure 6 and 7 show that a rough estimate is sufficient enough to achieve desirable regret. We connected the observed behaviour with our theoretical insights. We  politely point out that adaptive dueling strategies have yet not existed at the point of time we made our work public for the first time (February 2022).
>
> Our upper regret bound for BtWR is oblivious to the individual gaps $\Delta_i$ of each segment because Lemma 3 requires a choice of $\lambda$ depending on the minimal $\Delta_i$. As rightfully pointed out by the reviewer, including the knowledge of all $\Delta_i$ and their order could further tighten the bound. We abstained from doing so, as this would require further assumptions beyond the knowledge of $\Delta$.
>
> We tried to state our assumptions for MDB and DETECT very clearly in their respective analyses. At the same time we acknowledge that these could also be stated earlier in the introducing parts of our manuscript.

---

### Decision · Action_Editor_PiLw · 2024-08-07

**Recommendation:** Accept as is

**Comment:**

Two expert reviewers have thoroughly checked the paper. While they pointed out some weaknesses (e.g., the need for prior knowledge of certain problem parameters), they appreciated the author response which have cleared up some of their concerns. The reviewers eventually all agreed that the paper lives up to the TMLR standards and can be accepted for publication. I concur with this recommendation.

**Audience:**

The reviewers found the paper to be interesting, and pointed at several parallel works that considered very similar problems. This clearly indicates that the problem studied in the paper and the results should be of interest for a good part of the TMLR readership.

**Claims And Evidence:**

The paper supports all of its claims by rigorous proofs, whose correctness was checked by the reviewers to a reasonable extent.